# Diurnal and day-to-day characteristics of ambient particle mass size distributions from HR-ToF-AMS measurements at an urban site and a suburban site in Hong Kong

Berto Paul Lee[1], Hao Wang[2], and Chak Keung Chan[1,2*]

[1]School of Energy and Environment, City University of Hong Kong, Hong Kong, China

[2]Division of Environment, Hong Kong University of Science and Technology, Hong Kong, China

*Correspondence to:* Chak Keung Chan (chak.k.chan@cityu.edu.hk)

**Abstract.** Mass concentration based particle size distributions measured by a high-resolution aerosol mass spectrometer were systematically analyzed to assess long and short-term temporal characteristics of ambient particle size distributions sampled at a typical urban environment close to emission sources and a suburban coastal site representing a regional and local pollution receptor location in Hong Kong. Measured distributions were bimodal and deconvoluted into submodes which were analyzed for day-to-day variations and diurnal variations.

Traffic and cooking emissions at the urban site contributed substantially to particle mass in both modes, while notable decreases in mass median diameters were limited to the morning rush hour. Inorganic particle components displayed varying diurnal behavior, including nocturnal nitrate formation and daytime photochemical formation evident in both modes. Suburban particle size distributions exhibited notable seasonal disparities with differing influence of local formation, particularly in spring and summer, and transport which dominated in the fall season leading to notably higher sulfate and organic accumulation mode particle concentrations. Variations in particle mixing state were evaluated by comparison of inter-species mass median diameter trends at both measurement sites. Internal mixing was prevalent in the accumulation mode in spring at the urban site, while greater frequency of time periods with external mixing of particle populations comprising different fractions of organic constituents was observed in summer. At the suburban site, sulfate and nitrate in the accumulation mode more frequently exhibited differing particle size distributions in all seasons signifying a greater extent of external mixing.

At the urban site, periods of greater submicron inorganic mass concentrations were more likely to be caused by increases in both Aitken and accumulation mode particle mass in summer, while at the suburban receptor location organic and nitrate Aitken mode particle mass contributed more regularly to higher total submicron species mass concentrations in most seasons (spring, summer and winter).

## 1. Introduction

Apart from mass and chemical composition, the size distribution of fine particles represents a vital physical property with important implications for human health and environmental effects of ambient aerosols (Seinfeld and Pandis, 2006). Particle size relates directly to the aerodynamic properties which govern the penetration and deposition of particles in the airways and lungs (Davidson et al., 2005) as well as the scattering and absorption of light which affect the radiative properties and hence ambient visibility (Ahlquist and Charlson, 1967;Bohren and Huffman, 1983;Charlson et al., 1991;Schwartz, 1996;Seinfeld and Pandis, 2006). Hygroscopic growth in response to changes in ambient humidity can alter particle light scattering properties (Seinfeld and Pandis, 2006;Köhler, 1936) and activation of condensation nuclei particles into cloud droplets depend on atmospheric conditions, chemical composition, mixing state as well as the size and morphology of particles (Abbatt et al., 2005;Kerminen et al., 2012;Meng et al., 2014;Westervelt et al., 2013).

Studies into the size distribution of ambient particulate matter in Hong Kong have been largely based on size-segregated filter samples (Yao et al., 2007a;Zheng et al., 2008;Zhuang et al., 1999;Huang et al., 2014;Bian et al., 2014) and measurements by electrostatic classifier instruments (Cheung et al., 2015;Yao et al., 2007b) and were hence either limited in size resolution (offline filter samples) or chemical resolution (total particle count by classification). Most measurements in Hong Kong were conducted in suburban environments. Inorganic ammonium and sulfate were mainly found in fine mode particles in condensation and droplet mode size ranges, while nitrate had strong coarse mode contributions (Zhuang et al., 1999). Seasonal differences were evident in solvent-extractable organics and trace metals which were mainly found in $PM_{0.5}$ particles in the wet season and winter whereas in fall a shift to larger particles (0.5–2.5 μm fraction) in fall indicated a possibly stronger influence of aged particle components in the transition period of the Asian monsoon (Zheng et al., 2008). Size distributions acquired by a fast mobility particle sizer at the suburban HKUST supersite were investigated more recently to study the formation and accumulation of ultrafine particles under different air flow regimes. Particle number concentration enhancements during the day were attributed to secondary formation, while evening and nighttime peaks were thought to be related to transport of aged aerosols from upwind locations. Nucleation mode particle peaks were often observed in fall and related to regional pollution influence (Cheung et al., 2015). New particle formation events at the same site occurred as single and two-stage growth processes with organics and sulfuric acid contributing mainly to first stage growth in the daytime while nighttime second stage growth was attributed to ammonium nitrate and organics. Particle size growth into the diameter range of cloud condensation nuclei (CCN) was typically only achieved with the second growth stage (Man et al., 2015).

Investigations into particle size distributions in urban areas of Hong Kong are even scarcer. Yao et al. (Yao et al., 2007b) studied the properties and behavior of particles in vehicle plumes and reported a competing process between ambient background particles and fresh soot particles in the condensation of gaseous precursors and a dependency on temperature with bimodal volume size distributions observed at lower ambient temperatures and unimodal distributions in the lower accumulation size range at higher ambient temperatures.

The Aerodyne aerosol mass spectrometer (Canagaratna et al., 2007) is widely used to determine the chemical composition of major organic and inorganic components of non-refractory submicron particulate matter ($NR-PM_1$). In contrast to most traditional aerosol sizing instruments, the AMS is capable of resolving main chemical constituents

within size distributions through analysis of particle flight times and particle ensemble mass spectra (Canagaratna et al., 2007;Jayne et al., 2000;Jimenez et al., 2003;Rupakheti et al., 2005) and thus yields valuable additional information on differences in composition of submicron particles with the gross of particle mass in the Aitken mode range ($D_p$ ~ 10-100nm) and the accumulation mode range ($D_p$ ~ 100-1000nm) covered by the AMS. Thus far most studies employing ambient size distribution data from aerosol mass spectrometer measurements investigated longer time period averages, i.e. campaign averages (Salcedo et al., 2006;Sun et al., 2009;Aiken et al., 2009;Huang et al., 2010;Takegawa et al., 2009;Saarikoski et al., 2012;Li et al., 2015) or specific time periods of interest (Elser et al., 2016;Lee et al., 2013). Mohr et al. separated organic particle mass size distributions by periods of dominant influence of different PMF-resolved organic aerosol factors to study the properties of mass size distributions in relation to organic aerosol composition (Mohr et al., 2012). The 3D-factorization technique is an extension of traditional AMS PMF analysis on organic aerosol allowing to obtain estimates on the size distributions of organic aerosol factors, however under the assumption that factor size distributions remain invariant over the measurement period (Ulbrich et al., 2012).

The temporal evolution of species-specific size distributions, are mostly discussed qualitatively (Drewnick et al., 2005) and only few studies have evaluated temporal trends in mass size distributions in greater detail. Particle nucleation and subsequent growth events were investigated in Pittsburgh using size data from an AMS and two SMPS as well as various gaseous pollutant instruments and meteorological information. The AMS mass size distributions were evaluated quantitatively using the time series of binned particle concentrations generated from the grouping of raw data into wider size bins to represent different stages in the particle growth process. (Zhang et al., 2004). The same method was employed to evaluate contributions of ultrafine mode and accumulation mode particles to total organic particle mass (Zhang et al., 2005) by summation of size bins in the range of 30-100 nm and 100-1000nm. The authors also explored diurnal changes in size distributions of particle species by averaging over 3h periods in the morning (6–9 am) and afternoon (1–4 pm). Sun et al. present a qualitative discussion of diurnal variations in the mass size distributions of the m/z 44, m/z 57 and derived $C_4H_9^+$ ion signals from measurements at an urban site in New York (Sun et al., 2011). Similarly, Setyan et al. examined diurnal changes in the mass size distributions of organics and sulfate qualitatively and used binned concentrations (40–120, 120–200, and 200–800) nm in their quantitative analysis to study the evolution of particle chemistry in new particle formation and growth events (Setyan et al., 2012).

In this work, we introduce a systematic approach of assessing temporal variations in AMS mass-based particle size distributions from hourly diurnal variations to 24h-average based day-to-day trends to utilize two key instrumental advantages, i.e. species segregation and high time resolution, to obtain a more detailed and quantitative understanding of the variabilities in ambient particle mass size distributions and to provide an additional dimension to standard AMS data analysis techniques. In this context, we present a detailed discussion of particle size data from HR-ToF-AMS measurements during two field campaigns in Hong Kong in both urban and suburban environments. We aim to evaluate characteristic recurrent changes in size distribution as well as longer term trends in different seasons by analyzing day-to-day variations and diurnal variations of size distributions of submicron organics, sulfate, and nitrate particle mass. The two contrasting sites represent a typical urban source environment (inner-city, roadside station)

close to primary emission sources and a suburban location (coastal, HKUST supersite) which is largely a downwind
receptor of varying amounts of local urban, regional and long-range transported pollutants (Li et al., 2015;Huang et
al., 2014).
**2.   Methodology**
**2.1. Field campaigns**
Sampling of ambient submicron non-refractory particulate matter ($NR-PM_1$) was carried out using an Aerodyne HR-
ToF-AMS at the HKUST air quality supersite covering four seasons between May 2011 and February 2012 (spring:
2011-05, summer: 2011-09, fall: 2011-11&12, winter: 2012-02). The HKUST supersite is located on the campus of
the Hong Kong University of Science and Technology (22°20'N, 114°16'E), on the east coast of Hong Kong in a
suburban area with few primary emission sources in the immediate vicinity. Sampled air was drawn from the rooftop
of a pump house building at an approximate height of 25m above ground level.  For detailed descriptions of the
experimental setup, operating conditions, data treatment, and overall species composition we refer the reader to
previous publications (Lee et al., 2013;Li et al., 2015;Li et al., 2013). A further sampling campaign took place between
spring 2013 (2013-03 to 2013-05) and summer 2013 (2013-05 to 2013-07) at an inner-city urban location in the
densely populated and built-up Kowloon peninsula. Measurements were conducted next to the roadside air quality
monitoring station (AQMS) operated by the Environmental Protection Department (EPD) of the HKSAR Government
in the Mong Kok (MK) district on a pedestrian crossing at a major road junction. Sampled air was drawn from a height
of 3m above ground level. A comprehensive analysis of trends in species concentration and composition identified in
this urban campaign has been presented previously (Lee et al., 2015). In both campaigns, particles were sampled
through a $PM_{2.5}$ cyclone at a flow rate of 16.67 L/min into a sampling port from which 0.08 L/min was drawn by the
AMS and the remainder drawn by co-sampling instruments and an auxiliary pump. Sample air for the AMS passed
through a 1m long diffusion dryer (BMI, San Francisco CA, USA) filled with silica gel to remove bulk gas- and
particle-phase water. Additional data from various collocated instruments including meteorological data (wind,
temperature, relative humidity, solar irradiation), volatile organic compounds (VOCs) and standard trace gases such
as $NO_x$, $SO_2$, and $O_3$ were available.
AMS data were treated according to general AMS data treatment principles (DeCarlo et al., 2006;Jimenez et al., 2003)
with standard software packages (SQUIRREL, PIKA). Analysis of the unit-mass resolution mass spectra yielded non-
refractory submicron particle species concentrations of major inorganic constituents (SO4, NO3, NH4, Chl) and total
organics at a base time resolution of 10 min. Positive Matrix Factorization (PMF) was used to deconvolute high-
resolution organic mass spectra acquired at 10 min time resolution following recommended PMF guidelines for AMS
data (Zhang et al., 2011) with the AMS PMF analysis toolkit (Ulbrich et al., 2009). At the urban Mong Kok site, six
organic aerosol (OA) factors were identified encompassing three secondary organic aerosol (SOA) and three primary
organic aerosol (POA) factors of which one was attributed to traffic emissions and two to cooking activities (Lee et
al., 2015). Similarly, four factors were obtained from analysis of the urban HKUST site dataset with two SOA factors
and two POA factors, related to traffic and cooking respectively (Li et al., 2015). Further details on the treatment of
AMS size distribution data from both sampling campaigns are provided in the following section.
**2.2. Data acquisition and treatment**
In both campaigns, mass concentration based size distributions in terms of vacuum-aerodynamic particle diameter
($dM/d\log D_{va}$) were established by joint acquisition of particle time-of-flight (PToF) measurements and unit mass
resolution mass spectra (V-mode) with alternation between modes every 20s for 30 cycles amounting to 5 min of total
sampling time. High-resolution mass spectra were acquired for the following 5 min, and thus the overall raw data time
resolution for each mode was equal to 10 min. The total particle mass measured in the PToF mode was normalized to
the V-mode mass concentration of the same time step. Daily size distributions were generated by averaging over 24h
periods (from 0:00 to 23:59). Hourly diurnal size distributions were reconstructed by grouping size distributions within
the same hour of the day and establishing representative size distributions based on average, median, 25th and 75th
percentile concentration values of each size bin (*referred to as size distribution sets hereinafter*).
At both sampling sites, the seasonally averaged AMS size distributions were bimodal (Lee et al., 2013;Lee et al.,
2015;Li et al., 2015) with similar distributions having been observed in other AMS field studies in various parts of the
world. (Zhang et al., 2014;Sun et al., 2011;Huang et al., 2011;Aiken et al., 2009;Zhang et al., 2005;Crippa et al.,
2013;Docherty et al., 2011;Mohr et al., 2012). Multimodality of size distributions is typical for environments where
different sources or formation processes of particles play a role and accordingly such distributions can also be
represented as sums of discrete lognormal distributions of the respective constituting submodes (John, 2011).
The measured bimodal size distributions in this work were deconvoluted by fitting two log-normal distributed modes,
including one closer to the Aitken size range (*mode diameter ~100nm*) and one in the accumulation size range (*mode*
*diameter ~500nm*) employing the Levenberg-Marquardt algorithm (Gill et al., 1981) as a non-linear least squares fit,
to evaluate differences in trends and formation or transformation processes in the two size regimes. An example of a
size distribution fit and associated parameters is depicted in Fig. S1 in the Supplement. Additional fit residual analyses
were carried out in cases where the Aitken mode only accounted for small parts of (<10%) of the total particle mass
and uncertainties in integrated mode particle mass from the peak fitting were examined for all size distributions.
Details are presented in the Supplement (Text S2). The smaller mode typically exhibited mode diameters in the range
of 100-200 nm ($D_{va}$) and is thus in the transition region between Aitken and lower accumulation mode. For a clearer
distinction from the larger mode which unambiguously belonged to the accumulation size range, we opt to refer to the
small mode as *Aitken mode* in this work. Mode diameter (*i.e.* mass median diameter, MMD), curve width (*i.e.*
geometric standard deviation, GSD) and curve area (*equivalent to particle mass concentration within the mode*) are
sufficient parameters to completely describe a lognormal distribution and these key variables are used in the following
analysis on trends in the fitted species-specific size distributions of organics, nitrate, and sulfate from both HR-AMS
sampling campaigns in Hong Kong. Particle diameters are discussed in terms of vacuum-aerodynamic diameter, with
detailed discussions on properties and relationships to other size metrics available elsewhere (DeCarlo et al.,
2004;Slowik et al., 2004). Further details on procedures of PToF data acquisition and size distribution averaging can
be found in the Supplement (Text S1, Text S2). The sequence of main data treatment and analysis steps is shown in
Fig. 1.
The transmission efficiency of the AMS aerodynamic lens is known to fall off below ~100nm and beyond ~550 nm
of vacuum-aerodynamic diameter (Liu et al., 2007;Takegawa et al., 2009;Zhang et al., 2004;Bahreini et al.,
2008;Williams et al., 2013;Knote et al., 2011) and may bias measured particle mass and mode diameters, particularly
in the accumulation mode towards lower values if significant particle mass fractions fall in the size region of $D_{va} >$
550 nm. In the Aitken mode range, the effect of limited lens transmission is expected to be less substantial as particle
volume (and hence particle mass) of Aitken mode particles are much smaller. We discuss the effects of lens
transmission briefly in section 3.4. Delayed vaporization of particle components, e.g. under high mass loadings, can
lead to small shifts towards larger mode diameters in AMS size distributions (Docherty et al., 2015) and enhanced
tails in the size distributions (Cross et al., 2009), which may lead to larger fit residuals at the trailing edges. Generally,
the discussion of size distributions in this work should be viewed in the context of the instrumental capabilities and
previously mentioned limitations of aerosol mass spectrometry. Therefore, the resolved Aitken and accumulation
modes in this work reflect the apparent Aitken and accumulation modes within AMS measurable particle mass size
distributions.

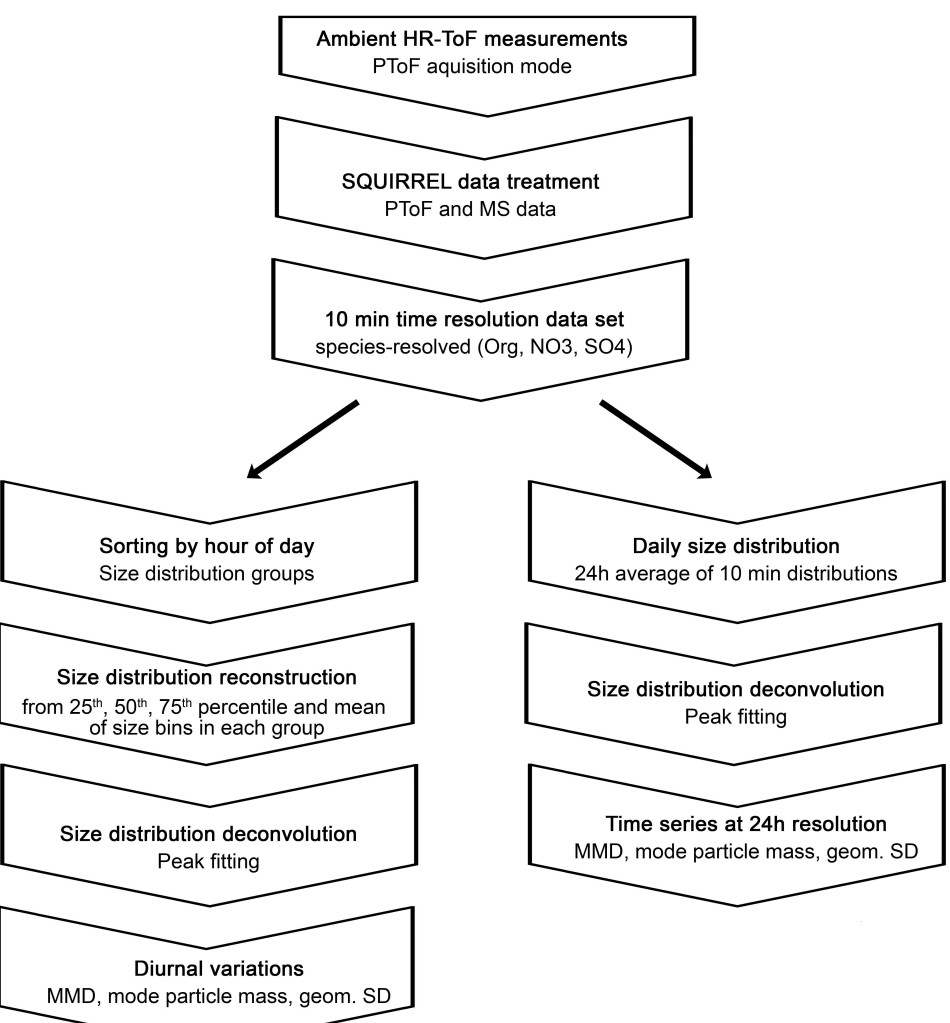

**Figure 1.** Flow chart of main data acquisition, data treatment and data analysis procedures

## 3.    Results and Discussion

### 3.1. Diurnal size distribution characteristics

Diurnal species variations are predominantly discussed in terms of total mass concentration up to the size cut of the sampling inlet or the instrumental capability, e.g. total species concentrations in NR-PM$_1$ for AMS-based studies. AMS mass-based size distributions can be utilized more systematically and complementary to standard AMS data analysis techniques by deconvoluting multimodal distributions into their constituting submodes and evaluating their variation and contribution to overall species concentration variations on a diurnal time scale. As previously mentioned, we examined size distributions reconstructed from the average, median, 25th and 75th percentile of hourly grouped size distributions, analogous to commonly reported AMS species diurnal variations, with quantitative analysis focusing on concentrations from the median dataset.

 **3.1.1.** **Urban roadside NR-PM₁**

The urban roadside measurements took place between March and July 2013 covering two seasons (Spring 2013:
*March to mid-May 2013;* Summer 2013: *mid-May to July 2013*) at a location dominated by the influence of primary
emission sources. Organics were the major particulate species in NR-PM₁ of which two-thirds were attributable to
traffic and cooking sources. Anthropogenic gas-phase species, including various VOCs, NOₓ, CO, and SO₂ were
continuously abundant as well (Lee et al., 2015;Sun et al., 2016). Particle size distributions at the urban site exhibited
discernible diurnal trends, with Fig. 2 depicting the variations in (mass median) diameters of the lognormal fitted
Aitken and accumulation modes, corresponding integrated peak areas representing the total mass accounted for by
particles in each mode, the geometric standard deviation signifying the spread across particle sizes as well as the total
submicron mass (NR-PM₁) diurnal variation for organics, sulfate, and nitrate based on AMS V-mode data. Individual
trends are discussed species-wise in the following.

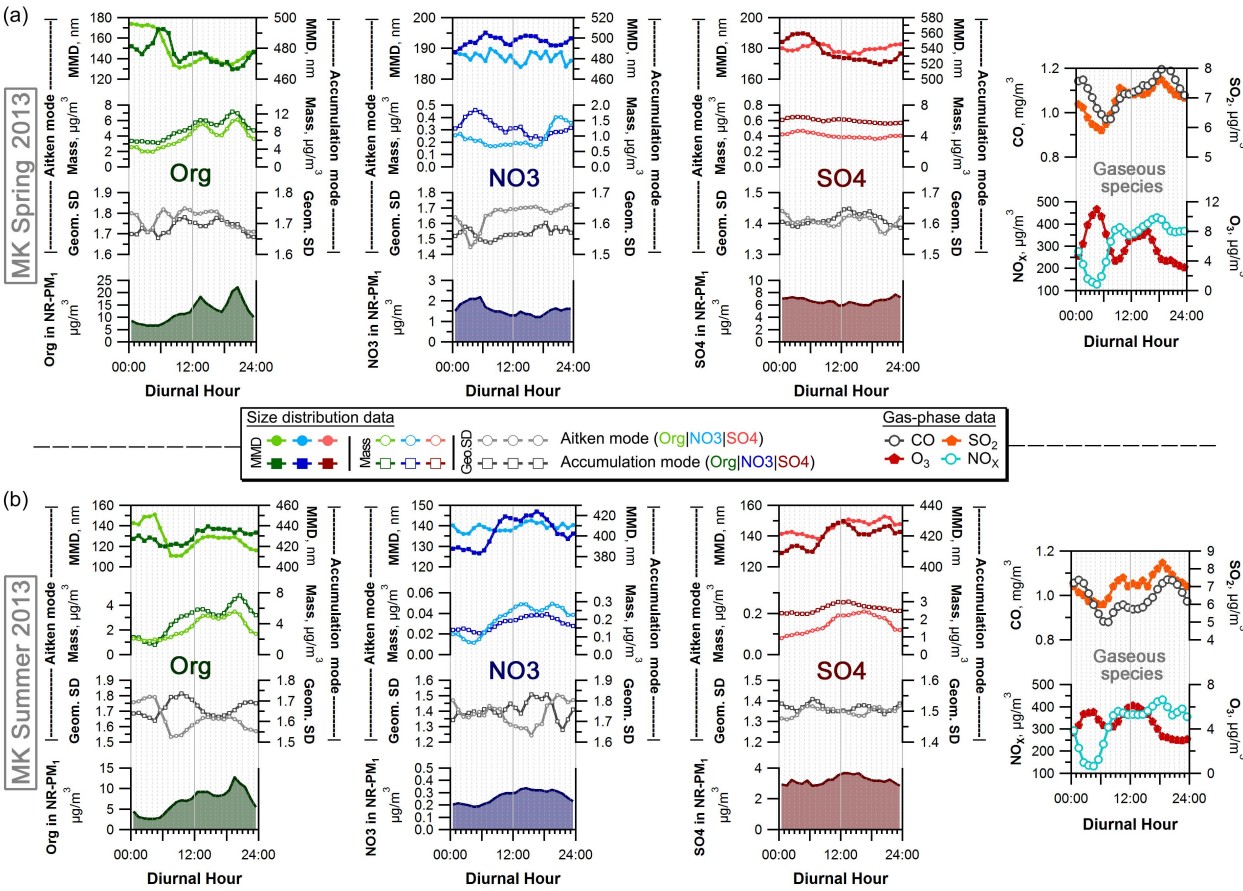


**Figure 2.** Diurnal variations of mode diameter (MMD), integrated mode mass concentration and width of the Aitken mode (*lighter*
*color*) and accumulation mode (*darker color*) from bimodal peak fits of the bin-median reconstructed size distributions at the urban
Mong Kok site and V-mode AMS species concentrations (*line with shaded background*) for organics, nitrate and sulfate (*left to*
*right*) in (a) spring 2013 and (b) summer 2013; The right-most panel depicts the median diurnal variations of relevant gas-phase
pollutants (O₃, CO, NOₓ, SO₂) measured at the adjacent Mong Kok Air Quality Monitoring Site (MK AQMS)
Organics
The diurnal variation of total Aitken and accumulation mode particle mass both largely followed the same trend as
total submicron organic mass (lower panels in Fig. 2 a, b) affirming that urban sources of organic particulate matter
contributed substantially to PM mass across the covered size region. Mass concentrations in both modes were smallest
during the night (*0:00 to 6:00*) and highest during lunch and dinner (*12:00 to 14:00, 19:00 to 21:00*), when the
influence of organic aerosol from cooking (COA - cooking organic aerosol) and from traffic (HOA – hydrocarbon-
like organic aerosol) were dominant (Lee et al., 2015). Trends in integrated mode particle mass and MMDs were
similar across all size distribution sets (Fig. S6 in the Supplement), confirming that they occurred persistently
throughout the measurement period and making diurnal timescale processes the dominant factor in determining size
characteristics of organic-containing particles at this urban roadside location.
Minimum Aitken mode particle mass concentrations (*median values*) amounted to 2.3 µg/m$^3$ in spring and 1.2 µg/m$^3$
in summer, accounting for 28-38% of total submicron particulate mass, and were typically reached between 03:00 and
04:00. These concentrations represent the estimated urban background mass of Aitken mode particles carried over
from the daytime and not removed by gravitational settling, coagulation or sweep-out as well as contributions from
nighttime activity such as traffic, which remains continuous in the inner-city districts at night albeit at much lower
vehicle numbers compared to the daytime.
Organic concentrations increased notably between 6:00 and 9:00 during the morning rush hour with traffic-related
constituents (*HOA – hydrocarbon-like organic aerosol*) accounting for the largest part (60% in spring, 40% in
summer) of this increase. In the Aitken mode, particle mass concentrations rose by 1.6 µg/m$^3$ (spring) and 0.8 µg/m$^3$
(summer) in the same time period. Assuming direct proportionality between the contribution of HOA to total
submicron organic mass increase and the increase of particle mass in each submode, 0.9 µg/m$^3$ (spring) and 0.3 µg/m$^3$
(summer) of particle mass were estimated as traffic-related organic components in the Aitken mode. Significant
changes were evident in the particle size metric (MMD) during the same time period, where a consistent decrease by
20-30% from about 170 nm (spring) or 160 nm (summer) to 130-140 nm (spring) or 120 nm (summer) was evident
with the concurrent increase in road traffic. This combined shift to smaller particle size and increase in total particle
mass denotes a strong increase in the total number concentrations of particles in the Aitken mode range by at least a
factor of 4-5 (assuming spherical particles and constant particle density) with significant additional contributions
expected from elemental carbon particles and smaller Aitken mode and nucleation mode particles below the range of
efficient particle transmission of the AMS inlet lens (Williams et al., 2013).
Beyond 10:00, changes in submicron organic mass concentrations were dominated by variations in cooking-related
organic aerosol (COA) components. During the main meal times (12:00 – 14:00 and 19:00 – 21:00) changes in organic
submicron mass were almost entirely (>80%) caused by COA in both seasons (Table C4 in the Supplement) and daily
maximum Aitken mode particle mass concentrations typically occur during these hours (5.5 – 6.2 µg/m$^3$ in spring, 3.1
– 3.5 µg/m$^3$ in summer) with higher concentrations during the dinner period. Analogous to HOA, considering
proportionality between COA fractional contribution and submode particle mass increase, primary cooking emissions
accounted for 1.7 - 1.8 µg/m$^3$ of organic particle mass in the Aitken mode. Dinner in summer represents a notable
exception, where the estimated cooking-related increase only amounted to 0.5 µg/m$^3$. This is mainly due to specific
local meteorological and geographical features owing to a greater frequency of easterly surface winds in the warmer
season and the geographical distribution of cooking sources predominantly to the east of the sampling site (Sun et al.,
2016;Lee et al., 2015), which led to considerably elevated Aitken mode mass concentration throughout the day
including the late afternoon period and a correspondingly smaller additional increase during the dinner time. The
aforementioned effect is particularly evident in the diurnal trend of the Aitken mode particle mass fraction among
total organic submicron mass (Fig. 3a) which displayed a broad bell-shape during the day in summer with nominal
increases of 9-10%, whereas in spring the variation follows a double peak behavior with nominal increases of 4-5%
during the meal times which emphasize the more intermittent behavior of cooking-related particle contributions in
spring. Cooking emissions did not lead to conspicuous changes in the size-related distribution metrics, i.e. there were
no obvious trends in particle diameters (MMDs) or distribution widths (GSDs) during the meal time periods (Fig. 2a,
b - *black lines in lower panels*).
In the accumulation mode, organic particle mass during the night hours (00:00 – 06:00) was 2.5 times larger in spring
(5.5 µg/m$^3$) than in summer (2.0 µg/m$^3$).  The mass concentration increase during the morning rush hour was larger
in summer ($\Delta M$=3.9 µg/m$^3$) than in spring ($\Delta M$=3.0 µg/m$^3$), which was mainly caused by daytime increases of SOA
components in summer (Lee et al., 2015), and consequently led to a lower fractional rush hour increase of traffic
related organic constituents. Estimated particle mass contributions of traffic emissions in the accumulation mode
amounted to 1.8 µg/m$^3$ in spring and 1.6 µg/m$^3$ in summer. In terms of particle size, the onset of the rush hour had
little conspicuous effects on the accumulation mode without clear trends in MMDs in both seasons. In summer the
shift to smaller MMDs was accompanied by a notable narrowing of the Aitken mode, whereas in spring Aitken mode
distribution widths remained largely stable throughout the day (Fig. 2a, b - *lower panels*).
Maximum accumulation mode particle concentrations during the meal hours reached 10.5 – 12.3 µg/m$^3$ in spring and
6.0 – 7.4 µg/m$^3$ in summer. Analogous to the Aitken mode, estimated cooking-related particle contributions in the
accumulation mode amounted to 2.0 µg/m$^3$ (spring) and 1.0 µg/m$^3$ (summer) during lunch, and to 2.7 µg/m$^3$ (spring)
and 2.4 µg/m$^3$ (summer) during dinner. Also, distribution widths (GSD) in the accumulation mode were not notably
affected by cooking emissions.  Seasonal differences were apparent in the mass median diameters of the accumulation
mode. In spring, mode diameters remained largely constant (+/- 10nm) apart from a subtle peak during the morning
rush hour, indicative of minor condensational growth of traffic-related primary organics or rapidly formed secondary
species. In summer, a consistent increase in particle size by 20nm (~5%) during the daytime points to particle growth
through secondary formation as a governing factor.
Aitken mode particles contributed larger fractions to the total increase in organic submicron particle mass during the
rush hour and mealtimes in spring (33-56%) than in summer (16-38%). These differences were presumably due to
seasonal meteorology and associated effects on the formation, accumulation, and dispersion of particles from primary
emission sources, as source strengths and characteristics of road traffic and commercial cooking are unlikely to vary
with seasons in the inner-city urban areas of Hong Kong. Ambient temperatures and solar irradiation differed
substantially with 7$^o$C higher average temperatures and three times higher integrated daily solar irradiation in summer
compared to spring (Fig. S10e-f in the Supplement). Lower overall ambient temperatures enhance condensation of
gas-phase emissions and particle nucleation and shift the gas-to-particle partitioning equilibrium of semi-volatile
constituents towards the particle-phase. We expect these volatility effects to be a main contributing factor, as sampling
took place in direct vicinity of the emission source, i.e. next to the road and thus potential impacts of physical effects
such as enhanced near-ground mixing and dispersion through thermally induced convection in summer are expected
to be of minor influence. Considering the previously discussed estimated traffic contributions during the rush hour,
the seasonal difference in mass concentration was much more pronounced in the Aitken mode (-67%, 0.6 µg/m$^3$) than
the accumulation mode (-12%, 0.2 µg/m$^3$), consistent with the expected stronger impact of reduced particle nucleation
and reduced condensation of semi-volatile exhaust components on fresher, smaller particles in the warmer season.
Comparing different size distribution sets (Fig. S6 in the Supplement), the average concentration set in summer yielded
notably larger resolved mass median diameters in both modes and greater Aitken mode mass compared to the median,
25[th], and 75[th] percentile concentration sets. This indicates a strong influence of extreme values (i.e. time periods with
both larger particle size and larger particle mass concentrations) and thus greater variability in size distributions in the
warmer season caused by specific high and low concentration events such as photochemical episodes and
precipitation, evident in the greater relative span of organic mass concentrations in summer (See Table C5 in the
Supplement: ratio of 10th and 90th percentile to median concentration in NR-PM$_1$). In spring, such events masked the
diurnal processes to a lesser extent and with consequently greater consistency across different size distribution sets.
Sulfate
Although variations of total submicron sulfate mass concentrations with time of day were generally subtle, distinct
trends were notable in MMDs and integrated mode mass concentrations in both Aitken and accumulation mode.
Generally, Aitken mode MMDs were 20% larger in spring (180nm) than in summer (150nm). While in spring
fluctuations in Aitken mode MMDs were small throughout the day within a narrow range of +/- 10nm and without
apparent regular features, the summertime diurnal variation exhibited a well-defined broad daytime peak with a shift
to ~15nm larger particle diameters. A matching trend was evident in the accumulation mode where MMDs increased
by ~20nm in summer. Conversely, in spring, a conspicuous nighttime peak in accumulation mode MMDs was
observed in the low traffic period between 01:00 and 07:00 which tracked closely with the diurnal variation of O$_3$
which peaked in the same period with the reduction of the NO$_x$ titration effects at low nighttime traffic volumes. While
particulate sulfate production during the day can be achieved through both homogeneous gas-phase oxidation of SO$_2$
by the OH radical as well as heterogeneous oxidation of SO$_2$ by dissolved H$_2$O$_2$ or O$_3$ (Seinfeld and Pandis, 2006),
nighttime production is limited to the non-photochemical heterogeneous pathway. The apparent increase in
accumulation mode particle size was also associated with an increase of integrated submode particle mass by ~0.7
µg/m$^3$ and thus suggests the possibility of heterogeneous SO$_2$ oxidation by residual ozone in the cooler and more
humid spring season as a local source of particulate sulfate. In the warmer and drier summer season, no corresponding
trend was apparent in either accumulation mode MMD or integrated mode concentration. The small magnitude of
additionally produced sulfate (< 1 µg/m$^3$) in spring renders the nighttime production a minor source of particulate
sulfate however and affirms that the bulk of the accumulation mode sulfate burden at the urban roadside still originated
from regional scale processes in both seasons. In summer, both modes exhibited notable increases in particle mass
concentration levels during the daylight hours by ~80% in the Aitken mode and by ~35% in the accumulation mode
compared to their respective nighttime "baseline" concentrations. Integrated over the whole day, the additional sulfate
burden above this baseline amounted to 0.4 µg/m$^3$ and 6 µg/m$^3$ and thereby accounted for 34% and 11% of the total
daily Aitken and accumulation mode particle mass respectively. This represents a rough estimation of possible local
photochemical contributions to the Aitken and accumulation size mode in summer at the urban roadside, excluding
possible physical effects, e.g. vertical mixing and advection or dilution laterally through the street canyon.
Enhancements in particle mass by photochemical contributions were more pronounced in the Aitken mode, with the
median fraction of Aitken mode particle mass among total AMS-measured particle mass increasing substantially from
its nighttime minimum at 4% to a maximum of 7% in the late afternoon in summer, while in spring the fraction
remained almost constant at 6% throughout the day (Fig. 3a).
Considering different size distribution sets (Fig. S6 in the Supplement), the 75$^{th}$ percentile size distributions and the
average size distributions displayed notable increases in Aitken mode particle mass during the nighttime by 20-50%
in spring. There was no corresponding trend in the accumulation mode, where changes in integrated mass
concentration remained consistently <10%. The skewing of the average and higher percentile data indicates the
influence of time periods with significantly elevated nighttime concentrations, likely related to events and atmospheric
conditions conducive to the extensive formation of Aitken mode sulfate particles. The accumulation mode showed no
notable changes in the average and 75$^{th}$ percentile data during the same time period, thus precluding physical processes
such as transport or lowering of the planetary boundary layer as likely influential factors for these observations.
Nitrate
Particulate nitrate mass concentrations in the Aitken and accumulation mode exhibited similar diurnal variations in
spring with lower daytime concentrations due to evaporation and higher nighttime concentrations where secondary
formation and gas-to-particle partitioning prevailed. Analogous to sulfate, the Aitken mode MMDs for nitrate showed
little change (<5%) throughout the day in both seasons. Aitken mode mass concentrations, however, exhibited a
twofold increase over the dinner hours accounting for approximately 0.9 µg/m$^3$ (~16%) of additional particle nitrate
mass per day. This may be due to the much higher abundance of small particles from cooking emissions providing
additional surface area to facilitate gas-to-particle partitioning of nitrate. Increased signal intensities of oxygenated
organic nitrogen ions (see Fig. S11 in the Supplement) have also been observed during dinner suggesting that organic
nitrate or other oxygenated nitrogen-containing organic species that produce nitrate fragments (Farmer et al., 2010)
may too have contributed to this observed concentration peak. Accumulation mode nitrate mass increased by almost
one-third in the low traffic period (01:00 – 07:00) compared to earlier night concentration levels (22:00 – 00:00)
accompanied by a slight increase in MMD by ~10nm in spring. This signifies notable nighttime nitrate production
through possibly nitric acid formation by ozone chemistry via the nitrate radical route under influence of organic
components or formation of N$_2$O$_5$ and subsequent hydrolysis during the night. Local nighttime nitrate production
effectively contributed ~3 µg/m$^3$ (~10%) to the total daily accumulation mode nitrate burden in spring.
Summertime nitrate production in Hong Kong has been mainly attributed to photochemical activity based on previous
measurements of inorganic gas- and particle-phase nitrogen species at the suburban HKUST site (Griffith et al., 2015).
Particulate nitrate mass concentrations at the urban Mong Kok site likewise exhibited clear daytime peaks, similar to

sulfate albeit at smaller magnitude with total integrated increases of ~0.3 µg/m³ and ~0.8 µg/m³ particulate nitrate per day in the Aitken and accumulation mode respectively. In the Aitken mode, particle mass remained elevated in the early night hours (~19:00 – 22:00), which was likely due to the previously mentioned cooking-related nitrate enhancement analogous to spring. The distribution of total submicron nitrate shifted slightly in favor of the Aitken mode in summer with ~18% of total submicron nitrate found in the Aitken mode compared to ~14% in spring. Comparing different size distribution sets (Fig. S6 in the Supplement), the average size distributions displayed notable disparity compared to the remaining sets in both seasons. In summer, integrated particle mass concentrations and MMDs from the average set exhibited consistently larger values than those from the 25th percentile, 75th percentile, and median sets indicating significant influence of time periods with high nitrate concentrations and larger nitrate-containing particles. In spring, the average data exhibited a decrease in MMD in the Aitken mode from night to day, implying prolonged periods of significantly smaller daytime Aitken mode particles.

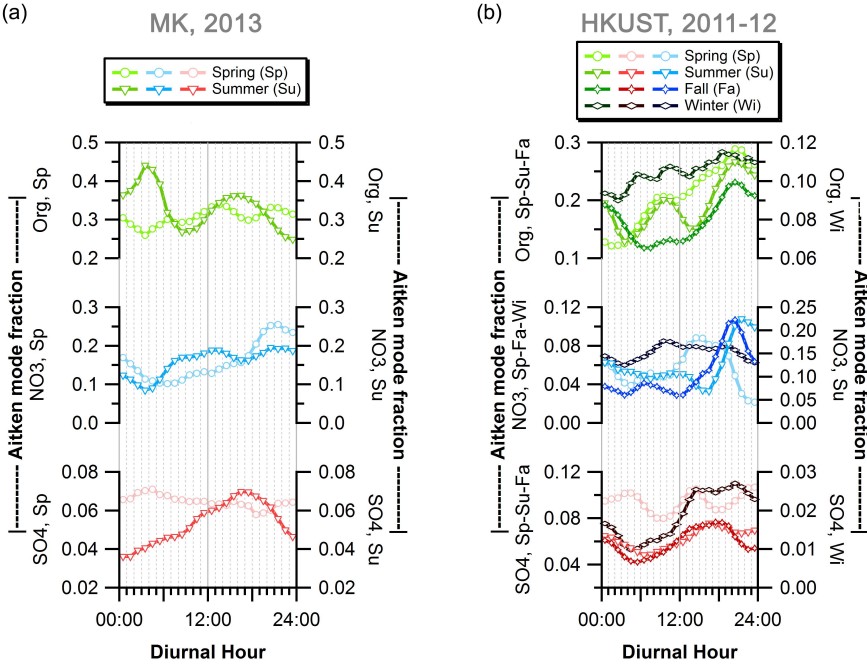

**Figure 3.** Diurnal variation of the fraction of Aitken mode particle mass among total submicron species mass for organics (*top*), nitrate (*middle*) and sulfate (*bottom*) at (a) the urban Mong Kong site, and (b) the suburban HKUST supersite in different seasons; *based on concentrations from bin-median size distributions, seasons denoted by marker color and type of marker symbol*

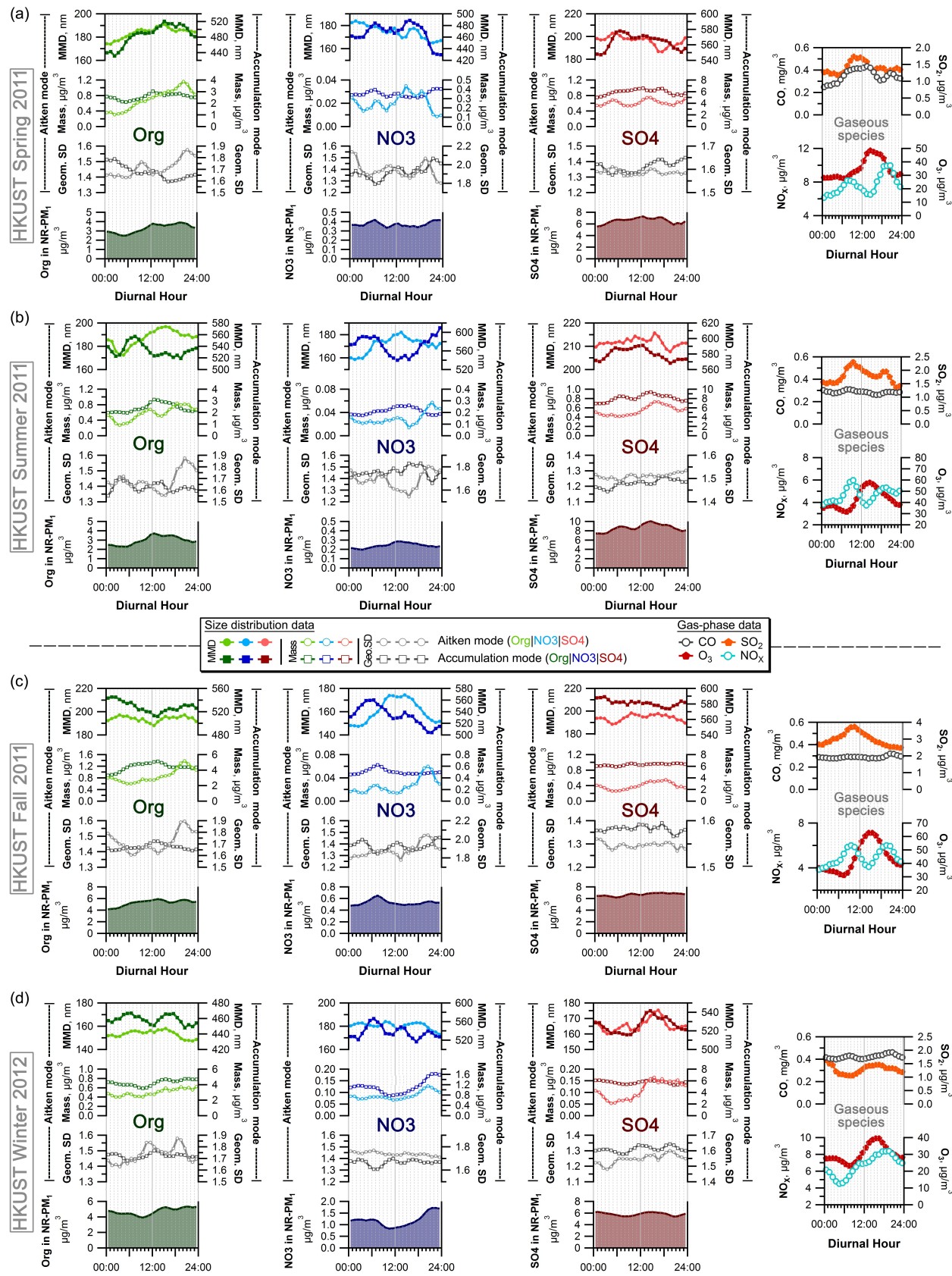

**Figure 4.** Diurnal variations of mode diameter (MMD), integrated mode mass concentration and width of the Aitken mode (*lighter*
*color*) and accumulation mode (*darker color*) from bimodal peak fits of the bin-median reconstructed size distributions at the
suburban HKUST site and V-mode AMS species concentrations (line with shaded background) for organics, nitrate and sulfate
(left to right) in (a) spring 2011, (b) summer 2011, (c) fall 2011 and (d) winter 2012; The right-most panel depicts the median
diurnal variations of relevant gas-phase pollutants ($O_3$, CO, $NO_x$, $SO_2$) measured at the same site.
**3.1.2.    Suburban coastal NR-PM$_1$**
The suburban HKUST site as a downwind receptor of urban and regional pollution was generally dominated by sulfate
and oxygenated secondary organic aerosol (SOA) components and much lower fractions of primary organic
constituents, which combined typically made up less than a quarter of total organics (Li et al., 2015). Trends in the
species segregated particle size distributions are discussed analogously to section 3.1.1., with Fig. 4 illustrating the
diurnal trends of the fitting parameters (MMD, integrated mode mass, geometric standard deviation) for organics,
sulfate, and nitrate at the suburban HKUST site.
Organics
There were significant seasonal differences with larger fractions (Fig. 3b) and concentrations (Fig. 5c) of Aitken mode
mass in total organic submicron particle mass in spring and summer compared to fall and winter, indicating greater
influence of closer-ranged formation sources in the warmer season. Springtime integrated Aitken mode mass
concentrations (~0.8 µg/m$^3$) were twice as high as those in winter (~0.4 µg/m$^3$). In the accumulation mode, highest
particle mass loadings were observed in fall (5 µg/m$^3$) and lowest loadings in spring (3 µg/m$^3$) following the frequency
pattern of continental air mass influence (Fig. S12 in the Supplement) in each season indicating continental transport
of particulate mass or gas-phase precursors. Lowest mass concentrations in the Aitken mode typically occurred in the
night hours (00:00 – 05:00) in a range of 0.3 – 0.5 µg/m$^3$ in spring, summer, and winter, while in fall mass loadings
of 0.7 - 0.8 µg/m$^3$ were reached. Diurnal changes were least pronounced in winter with largely constant integrated
Aitken mode particle concentrations. In the remaining seasons, varying degrees of daytime changes were apparent
with a general increase around 06:00, likely owing to citybound commuter traffic from surrounding roads to the west
of the sampling site at 1-2km of lateral distance. This also led to a modest increase in particle polydispersity with a
discernible widening of the Aitken mode size distributions (*black solid line, lowest panels in* Fig. 3). Daily maxima
in spring, summer and fall were reached in the early evening (~21:00) with marked differences in absolute mass
concentrations depending on the respective season, from a summer time low of 0.8 µg/m$^3$ to a fall season high of 1.4
µg/m$^3$. Mass median diameters in the Aitken mode were smaller in the night hours and displayed subtle increments
during the day in the range of 10-20 nm reaching their maximum typically in the late afternoon, except for the fall
season when mass median diameters displayed very little variation with time of day.
Total particle mass in the accumulation mode in spring and summer reached minima during the night hours (2 µg/m$^3$)
and maxima (3 µg/m$^3$) around noon, remaining stable in the daylight hours thereafter.  MMDs increased notably from
440nm at night to 510nm during the day in spring, while in summer a morning rise by ~30nm from 530nm to 560nm
was obvious between 06:00 and 10:00 and coincided with the morning rush hour and the associated early morning
peak of NO$_x$ and an otherwise stable mode diameter of 530nm for the rest of the day. In fall, the increase in
accumulation mode organic mass occurred much earlier, starting in the dark hours at 04:00, with a corresponding
trend also evident for nitrate but absent for sulfate, indicating a common source of these organic and nitrate enriched
particles. Nighttime MMDs for organics were generally larger (540nm) and decreased to a minimum of 510nm in the
early afternoon accompanied by a slight widening of the distribution.  In winter, mass concentrations decreased
appreciably in the early morning hours and started to increase only beyond 10:00. In the colder seasons (fall, winter),
a similar concentration pattern was also observed for gas-phase SO$_2$ which is considered as a largely regional pollutant
with few distinct local sources (Yuan et al., 2013), indicating that changes in boundary layer and mixing with regional
background were likely the more dominant processes in winter.

Sulfate
Aitken mode sulfate mass concentrations peaked in the afternoon from spring throughout fall with maximum
concentrations reached progressively later in the afternoon (14:00 in spring; 16:00 in fall). Nominal concentrations
were highest in spring and summer (0.5-0.6 µg/m$^3$), slightly lower in fall (0.4 µg/m$^3$) and reached the lowest levels in
winter (0.1 µg/m$^3$). In addition to the afternoon peak, a conspicuous early morning peak of similar magnitude was
evident in spring between 02:00 and 06:00. A greater proportion of southerly winds was evident in said time period
compared to the overall seasonal wind frequency distribution (Fig. S13a in the Supplement) and may indicate transport
of sulfate from marine sources in the southern parts of Hong Kong.  Diurnal variations in MMDs and GSDs were
generally small and without obvious regular trends. Nominal mass median diameters were significantly lower in winter
(~170nm) compared to spring and fall (~190nm) and summer (~210nm).
Trends in accumulation mode particle mass were more pronounced. In spring, a shallow concentration valley during
the late evening and night hours (20:00 to 03:00) with minimum concentrations of 5 µg/m$^3$ was apparent, while
daytime concentrations stayed largely invariant at 6 µg/m$^3$. The MMDs followed a similar variation with a minimum
mode diameter around 550nm in the early hours of the day and slightly larger daytime MMDs around 570nm. Nominal
concentrations were larger in summer with a nighttime valley concentration of 7 µg/m$^3$ and a well-pronounced broad
day peak with a maximum of 9.5 µg/m$^3$ in the early afternoon (14:00-15:00). A prior additional morning peak occurred
between 04:00 and 10:00 with particle mass concentrations reaching 8.5 µg/m$^3$ related to a consistent north-easterly
morning wind pattern (Fig. S13b in the Supplement) and likely associated with transport from north-easterly coastal
regions or nighttime fisheries related maritime traffic. The diurnal trend in mass median diameter was similar to that
in spring with a night minimum of 570nm and day maximum of 590nm.
In fall, accumulation mode characteristics showed no significant diurnal variability, with a largely stable integrated
particle mass of 6 µg/m$^3$ and only subtle MMD changes (585nm at night; 575nm during the day). In winter, two
concentration dips with reductions by ~0.5 µg/m$^3$ between 06:00 and 10:00 and between 18:00 and 22:00 were evident,
while MMDs increased during the day between 10:00 and 15:00 from 520nm, peaking at a size of 540nm.

Nitrate
Nitrate particle mass in the Aitken mode was generally small from spring throughout fall amounting to 0.01 - 0.06
µg/m³. Winter time concentrations were larger in a range of 0.06 - 0.08 µg/m³ during the day and 0.10 - 0.12 µg/m³ in
the late evening hours. The latter evening peak centered around 21:00 was evident in most seasons (except spring)
and accounted for 12-23% (0.1-0.25 µg/m³) of total daily Aitken mode nitrate mass burden. Similar to the urban
roadside location, these nighttime nitrate peaks coincided with the peak period of organic cooking aerosol
concentrations (Fig. S14 in the Supplement), which were however significantly smaller at the suburban measurement
site and mainly attributed to the operation of an on-campus student canteen (Li et al., 2015). Trends in mass median
diameters varied between seasons with no discernible trend in winter, a subtle decreasing trend with time of day in
spring and broad daytime diameter increases in summer and fall. Solar irradiation in these two seasons was
comparatively high (Fig. S10b-c in the Supplement) indicating that photochemical nitrate production in the Aitken
mode may have led to this observed growth in particle size.
Integrated particle mass concentrations in the accumulation mode only exhibited subtle variations from spring
throughout fall, with essentially constant diurnal concentrations in spring, a subtle daytime peak in summer which
accounted for ~ 15% of total daily accumulation mode nitrate (corresponding to 0.7 µg/m³) and a conspicuous morning
peak between 04:00 and 10:00 in fall accounting for ~ 5% of total daily accumulation mode nitrate (corresponding to
0.5 µg/m³). Clearer seasonal differences were evident in the trends of MMDs. In spring, MMDs decreased appreciably
over the late evening hours (21:00-0:00) with a concurrent widening of the size distribution (increase in GSD). In
summer, accumulation mode diameters decreased during the day by ~40nm with a similar trend in accumulation mode
organics. Winter time MMDs exhibited a more complex pattern with larger mode diameters in the early hours (04:00
– 10:00) and during the noon-time, and a late-afternoon dip leading to larger spread of intra-day mode diameters
ranging from 510nm to 570nm.
In comparison to the urban roadside measurements, diurnal particle size characteristics and mass concentrations in the
Aitken and accumulation mode were much more variable for all investigated species at the suburban HKUST site,
indicating that longer time scale processes and irregular events (transport patterns, local meteorology) were probably
more important in governing particle size distribution characteristics than diurnal processes.
**3.2. Day-to-day size distributions and seasonal averages**
To evaluate the evolution of particle size distributions within seasons, average species-specific size distributions were
generated by averaging raw distributions over 24h periods (between 0:00 and 23:59). There was clear long-term
variability in both resolved Aitken and accumulation mode MMDs and integrated submode particle mass
concentrations for all species (Fig. S15-16 in the Supplement) and overall seasonal differences which have been briefly
addressed in the discussion of the diurnal size distribution variations between seasons. Figure 5 depicts the overall
average values for all daily fitted MMDs and integrated particle mass concentrations in both the Aitken and
accumulation mode at the suburban HKUST and urban MK sites.

### 3.2.1. Seasonal trends

For the MK roadside station, particle mode diameters were generally larger in spring than in summer for all three investigated species, but with clear differences in the magnitude of changes among individual species. In the Aitken mode, organics and sulfate displayed a moderate decrease in mode diameter from spring to summer by 7-8% each, while nitrate saw a more significant decrease by 25% from spring to summer. In contrast, accumulation mode MMDs for organics exhibited only a subtle decrease by 5% and more substantial decreases for sulfate and nitrate by 20-22% each. Total Aitken mode particle mass decreases varied strongly: -15% for organics, -36% for sulfate and -67% for nitrate. In the accumulation mode, organics and sulfate exhibited similar relative decreases by 40-46%, while nitrate particle mass reduced drastically by 85%.

At the suburban HKUST site, Aitken mode MMDs of nitrate and organics decreased with the progression of seasons from spring to winter with highest mode diameters observed in spring and summer and appreciable decreases in winter by -9% for nitrate and -25% for organics compared to the warmer seasons. Sulfate displayed a similar winter time decrease in MMD (-15%) and an increase of similar magnitude in the summer season (+13%) compared to spring and fall. Variations in sulfate and organic accumulation mode diameters were minor between spring and fall, while wintertime MMDs were 7-12% lower. Nitrate exhibited an overall higher variability in mass median diameters in the accumulation mode in spring (larger standard deviation) and with on average 10% lower MMDs compared to other seasons. In line with the reduction in Aitken mode MMDs in winter, the integrated Aitken mode particle mass decreased as well, by -16% for organics and almost -75% for sulfate, whereas nitrate contributions remained largely stable throughout the seasons. Organic accumulation mode particle mass was significantly higher in the fall and winter season by factors of 1.6 – 2. Diurnal variations in the degree of oxygenation were least pronounced in these seasons (Li et al., 2015) suggesting that influence of transport in autumn and winter likely dominated over local formation, thus exerting greater effects on particle mass in the larger size mode. Particulate nitrate concentrations were generally low in the accumulation mode from spring through fall, but increased sharply in winter by factors of 3 – 4. Sulfate accumulation mode mass concentrations remained more stable but saw significant summer time enhancements by ~30% likely due to photochemical activity which also led to high concentrations of Ox and a higher degree of oxygenation of organic aerosol among the four seasons (Li et al., 2015).

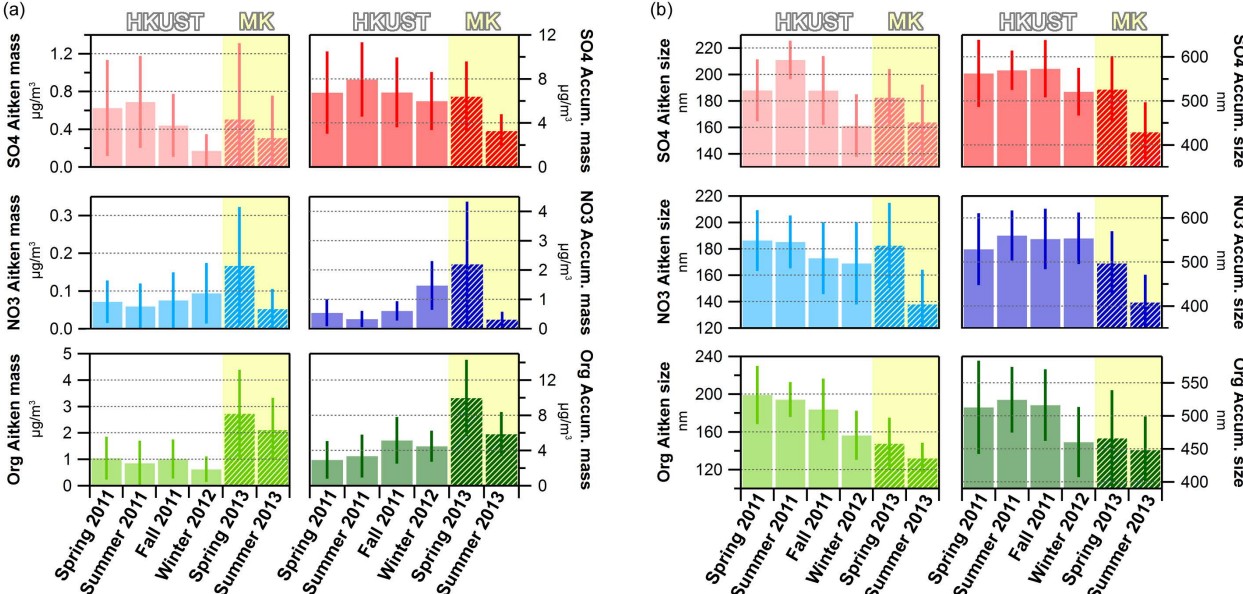

**Figure 5.** Average and standard deviation of daily fit values of Aitken and accumulation mode particle mass and mass median diameters at the suburban HKUST site (*solid bars*) and urban MK site (*hashed bars*). The integrated particle mass is depicted in (a) for the Aitken mode (*left panels*) was well as the accumulation mode (*right panels*) for sulfate, nitrate, and organics respectively. The mass median diameter is depicted in (b) for the Aitken mode (*left panels*) as well as the accumulation mode (*right panels*) for sulfate, nitrate and organics respectively.

Large particles contribute more to particle volume and hence particle mass. Correspondingly, the total submicron concentration of a given species is typically governed by changes in the accumulation mode particle mass and accordingly observed correlation values between integrated accumulation mode particle mass and individual NR-PM$_1$ species mass concentrations were generally high ($R_{pr}$>0.90) at both measurement sites (Fig. S17 in the Supplement). This applied to both measurement sites regardless of the season. Aitken mode trends were less akin. At the urban roadside station, neither sulfate nor nitrate particle mass in the Aitken mode notably correlated with the respective total submicron species mass concentration in spring (all $R_{pr}$≤0.20), whereas in summer correlations were more significant with $R_{pr}$=0.51 for sulfate and $R_{pr}$=0.80 for nitrate. This signifies that periods of greater species mass concentrations were more likely to be caused by increases in both Aitken and accumulation mode particle mass indicating that particle formation and growth affecting smaller particles was more likely to occur in the warmer season. For organics, Aitken mode particle mass and submicron species mass correlated only weakly ($R_{pr}$ = 0.26 in spring and $R_{pr}$ = 0.38 in summer), i.e. each organic particle submode was governed by largely different dominant sources or formation processes in both seasons at the roadside.

At the suburban background site, Aitken mode particle mass for sulfate showed little correlation with total submicron sulfate concentration ($R_{pr}$≤0.10) apart from the spring season ($R_{pr}$=0.36) where more frequent wet and foggy conditions may have facilitated sulfate formation in both size modes. For organics and nitrate significantly larger correlation coefficients of submode particle mass to total species concentration ($0.5 \leq R_{pr} \leq 0.7$) were observed in most seasons (spring, summer, winter) indicating significant influence of local or regional formation processes on

organic and nitrate Aitken mode particulate mass at the suburban receptor location. In the fall season, much weaker
correlations ($0.2 \le R_{pr} \le 0.4$) were likely caused by the dominance of continental air mass influence (Fig. S12c in the
Supplement) and greater influence of aged accumulation mode particles on total submicron nitrate mass
concentrations.

### 3.2.2.    Inferred changes in mixing state

Shifts in mixing state of ambient particles can be inferred from the inter-species analysis of mass median diameters.
Close nominal agreement (i.e. diameter ratios close to 1) infer that different species were distributed similarly across
the particle size range which thus most likely represents a largely internally mixed particle population, while the spread
of data (correlation coefficient) indicates the temporal homogeneity or divergence of resolved mode diameters. A
hypothetically perfectly internally mixed particle population over the whole sampling period would, therefore, yield
MMD ratios and Pearson's R values of 1 between species, while larger or smaller values are indicative of a greater
frequency of heterogeneous (i.e. more externally mixed) particle populations (Fig. 6).
At the urban Mong Kok site, changes in accumulation mode mass median diameters for nitrate and sulfate followed
similar trends ($R_{pr}$ = 0.88-0.89) and with diameter ratios close to 1 (0.94–0.95) Similarly, fitted accumulation mode
diameters of organic constituents predominantly followed that of sulfate in spring nominally (diameter ratio 0.88) and
temporally ($R_{pr}$ = 0.80). The nominal agreement of organic and sulfate accumulation mode diameters persisted
(diameter ratio 1.03) overall in summer, however, there was significantly more temporal divergence ($R_{pr}$ = 0.65)
indicating a greater frequency of time periods with external mixing of particle populations comprising different
fractions of organic constituents.
External mixing is more prevalent for freshly formed smaller particles which have typically undergone less
condensational growth, coagulation or aqueous-phase reactions.  Indeed, the correlation coefficients of both nitrate
and organic Aitken mode MMDs with respect to sulfate were notably lower (0.50 and 0.62) indicating frequent periods
of particle populations with different species prevailing in different size regions within the Aitken mode.
Sulfate and nitrate were still more likely to occur internally mixed in the Aitken mode in spring with similar diameters
(nitrate to sulfate MMD ratio = 1.00), while organic Aitken mode MMDs were consistently lower, indicating greater
fractions of organic dominated particles towards the lower end and more inorganic dominated particles towards the
upper end of the fitted Aitken mode.
In summer, both nitrate and organic MMDs tended to be lower than those of sulfate (diameter ratios of 0.79 – 0.83)
but similar to each other, thus implying a shift to externally mixed populations of more nitrate and organic enhanced
and internally mixed smaller Aitken mode particles and sulfate dominated larger Aitken mode particles.
At the suburban HKUST site, accumulation mode MMDs of both nitrate and organics were generally quite similar to
those of sulfate with diameter ratios of 0.88 – 1.06. Compared to the urban site, correlation coefficients of nitrate and
sulfate were consistently lower (0.54 – 0.67) indicating a much greater frequency of time periods where sulfate and
nitrate dominated particles in the accumulation exhibited significantly different particle size distributions.
In winter, organic MMDs were consistently lower than those of sulfate and nitrate indicating a greater proportion of
externally mixed particle populations with organics enriched particles in the lower accumulation size range and
inorganic dominated particles in the larger accumulation size range. The least variability in particle size was observed
in the summer season where MMDs in both Aitken and accumulation mode displayed variations in relatively narrow
ranges between 200-250nm and 500-700nm, whereas in the remaining seasons time periods with particle populations
of lower MMD were more frequent, extending to MMDs as low as 100nm in the Aitken mode and 300nm in the
accumulation mode.
In the Aitken mode, mass median diameters overall were quite similar across species, with diameter ratios of organic
and nitrate distributions to those of sulfate in the range of 0.87 – 1.06, indicating that they generally covered a similar
size range. The temporal agreement was highly variable with correlation coefficients ($R_{pr}$) spanning from 0.21 to 0.75
indicating that Aitken mode particle populations at the suburban site were generally more diverse and likely influenced
by a greater range of particle formation and growth mechanisms compared to the urban Mong Kok site.

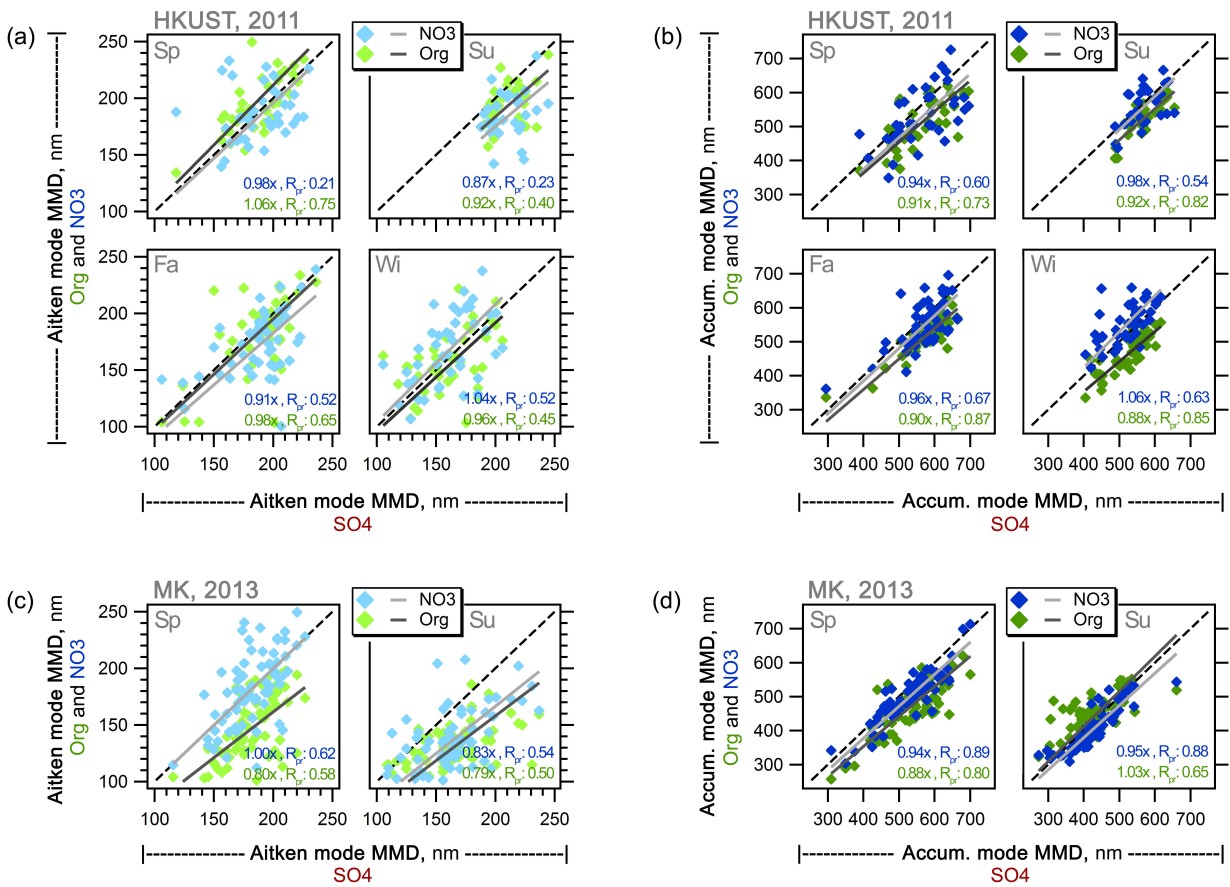


**Figure 6.** Scatter plots of fitted mass median diameters of organics and nitrate vs. sulfate for (a) the Aitken mode and (b) the
accumulation mode at the HKUST suburban site, and (c) the Aitken mode and (d) the accumulation mode at the urban Mong Kok
site

## 3.3. Comparison to previous studies

Particle size distribution studies in Hong Kong are generally scarce and have focused on either size segregated filter samples (MOUDI) for general ambient measurements or electrostatic classification in particle formation and particle growth studies (Guo et al., 2012;Cheung et al., 2015) . The latter studies focus on specific and narrow time periods and lack general discussions on ambient particle size distributions.

Two ambient studies were undertaken at the suburban coastal HKUST site using size-segregated samples from a ten-stage MOUDI sampler and offline chromatographic analysis. Inorganic constituents (NH4, NO3, SO4) in fine particles (i.e. $D_p$<1.8 µm) were shown to follow bimodal distributions with mode diameters in the range of 0.14-0.21 µm and 0.46-0.58 µm in samples collected in the winter season, while the main mode was observed in the coarse region (4-6 µm) for all three species (Zhuang et al., 1999). A subsequent year-long observational study also reported bimodal fine particle distributions with mode diameters of 0.1-0.3 µm and 0.7–0.9 µm and 1-2 additional modes in the coarse region (Bian et al., 2014), however, the main mode in the size distributions of sulfate, ammonium, potassium and oxalate was observed in the droplet mode (0.7 – 0.9 µm) in this study. Vehicle exhaust plumes sampled on-road from a Mobile Real-time Air Monitoring Platform (MAP) across Hong Kong's road network exhibited three distinct particle volume size distributions: a unimodal distribution with an accumulation mode at 0.2 µm and two bimodal distributions with a minor mode at 0.2 µm and the dominant mode at 0.5 or 0.7 µm (Yao et al., 2007b).

The bimodality in the fine particle range across these studies is consistent with the AMS-based results in this work. Nominally, the accumulation mode diameters from filter based studies and the chase studies are larger than those from AMS measurements where maximum mode diameters occurred at $D_{va}$ ~ 700nm, corresponding to $D_a$ ~ 470 (assuming $D_{va}$ ~ $D_a$ * density; particle density ~ 1.5 g/cm$^3$). Direct comparability is however limited due to fundamental differences in sizing techniques (MOUDI: atmospheric pressure; AMS: near-vacuum), sampling times (MOUDI: 24h samples, scattered time line; AMS: minute raw resolution averaged to hourly or daily, continuous time line), measurement uncertainties (MOUDI: sampling artifacts such as vapor adsorption and desorption; AMS: inlet lens transmission) and aerosol pretreatment (none for MOUDI with potential impacts on particle size in high humidity (>80%) conditions (Fang et al., 1991); AMS: removal of water prior to introduction to instrument).

## 3.4. Influence of AMS lens transmission

The quantitative measurement of particle components in the AMS is dependent on three major factors which may lead to particle loss prior to detection (Huffman et al., 2005). Irregularly shaped particles deviating from the flight path in the vacuum chamber may miss the vaporizer. Particles bouncing off the vaporizer surface will not be vaporized and hence may not be detected. Lastly, the aerodynamic lens which is part of the instrument's inlet system does not transmit particles uniformly across all particle diameters. Small particles are lost due to insufficient focusing or diffusion and large particle impact the lens apertures (Liu et al., 2007;Williams et al., 2013). Being a function of particle size, the latter factor affects both total AMS quantifiable particle mass (NR-PM$_1$) and measured mass size distributions. Transmission curves determined for the standard lens, which is fitted in most AMS instruments, can

vary but typically show efficient (i.e. close to 100%) transmission in the range of 100-550nm (Knote et al., 2011)
falling of significantly at either edge.
We examined the potential impact of lens transmission on the AMS mass size distributions on a number of 24h size
distributions from the fall season HKUST dataset covering both efficient and reduced lens transmission size regimes
(accumulation mode diameters between 400 and 600nm). Panels a-c in Fig. S18 in the Supplement depict original and
lens-transmission corrected 24h mass size distributions for organic, nitrate and sulfate, assuming the transmission
function (subpanel d) reported by Liu et al., 2007. Impacts were generally larger in the accumulation mode range with
evident shifts to larger mode diameters and larger mode mass concentrations observed in all size distributions. In the
small diameter range, enhanced shoulders can occur which may however be artifacts due to the larger uncertainties
(low signal to noise ratio at small particle mass), i.e. greater noisiness at the leading end of AMS size distributions.
For a quantitative comparison, bimodal fitting parameters from the corrected distributions were plotted against those
from, the original distributions in Fig. S18 in the Supplement (subpanels e-g refer to the Aitken mode and subpanels
h-j to the accumulation mode). Leading edge shoulders in the corrected size distributions were not considered in the
fitting. Changes in the Aitken mode mass median diameters were minor (on average ~3%), while the integrated mode
particle mass increased moderately (~48%). In the accumulation mode, mass median diameters increased by ~28%
and integrated mode particle mass doubled. The distribution widths (geometric standard deviations) exhibited little
change in both modes (increases of 2-3%).
Fitting results will therefore vary depending on whether AMS size distribution and concentration data are corrected
for lens transmission. While explicit lens transmission corrections can improve the accuracy of quantification of AMS
species concentration and size distribution measurements, few ambient studies explicitly use lens transmission
corrections based on individual experimental determinations or literature values e.g. (Quinn et al., 2006;Cross et al.,
2007). Lens transmission curves can vary between instruments (Fast et al., 2009) and are inherently difficult to
determine accurately experimentally. As discussed previously, scaling of size distributions by lens transmission curves
may introduce artifacts in noisier size distributions (e.g. low end of the Aitken mode, low concentration periods, short
term size distribution averages). Trailing edges from slow vaporization (e.g. under high particle mass loadings) may
be exacerbated and inflate mass concentrations at the upper size cut range of the AMS. The majority of ambient
studies employs a combined correction factor (collection efficiency, CE) considered to be the joint product of the
previously mentioned transmission efficiencies related to particle bounce, beam broadening and lens transmission
(Middlebrook et al., 2012) derived from aerosol composition and by comparison to collocated speciation or particle
sizing instruments. As the AMS lens transmission curve could not be determined in this study and to avoid additional
uncertainties from the application of non-instrument specific lens transmission values, we followed the CE correction
method in the analysis of the size distribution data in this study. The reported values of resolved mode diameters and
integrated mode should therefore be regarded as lower bound estimates in the context of the instrumental limitations
affecting ambient AMS measurements.

### 4. Conclusion

A detailed analysis of AMS mass-based particle size distributions of sulfate, nitrate, and organics in submicron particulate matter measured at two contrasting locations in Hong Kong during two field campaigns has been undertaken. Deconvolution of size distributions into Aitken and accumulation submodes was accomplished by log-normal peak fitting and trends in particle size (mass median diameters), dispersity (geometric standard deviation) and overall particle mass (integrated mode area) were discussed on a diurnal time scale and on a daily basis to evaluate longer-term changes in size distribution characteristics. At the urban roadside location, clear diurnal influences of primary particle and gas-phase species were evident affecting both inorganic and organic component size distributions. Traffic and cooking contributed an estimated $0.3 - 0.9\,\mu g/m^3$ and $0.5 - 1.8\,\mu g/m^3$ of organic component particle mass in the Aitken mode, and $1.6 - 1.8\,\mu g/m^3$ and $1.0 - 2.7\,\mu g/m^3$ respectively in the accumulation mode with concentrations level varying with seasons. Notable changes in Aitken mode mass median diameters of organics were limited to the morning rush hour. Daytime particle concentration maxima of sulfate and nitrate in summer indicated substantial influence of photochemical processes, which also led to increments in mass median diameters in the accumulation mode thus inferring associated particle growth. Nocturnal nitrate formation was apparent in the accumulation mode in spring concurring with the nighttime peak of ozone at the roadside, while in the Aitken mode nitrate particle concentrations were significantly elevated during the dinner hours. Organics-related size distributions were mostly governed by intra-day changes at the urban site with very similar trends across different size distribution sets (i.e. concentration regimes), while disparities in diurnal variations among different size distribution sets were evident for nitrate and sulfate, particularly affecting the average sets, indicating stronger influence of irregular external factors which were not associated with diurnal time scale processes.

Suburban particle size distributions exhibited variable diurnal characteristics, suggesting that irregular processes such as transport and seasonal meteorological conditions were the more dominant processes influencing particle size characteristics. Aitken mode particle mass of organics was significantly larger in spring and summer indicating greater influence of more local formation sources in the warm season. In the accumulation mode, organic particle mass concentrations were highest in fall and lowest in spring, following the frequency pattern of continental air mass influence. For sulfate, Aitken mode mass concentrations mass concentrations peaked in the afternoon from spring throughout fall with highest nominal concentrations in spring and summer and lowest levels in winter, while accumulation mode particle mass was highest in summer and fall and lowest in winter, similar to the trend observed among organic constituents.

Nitrate particle mass in the Aitken mode was generally small in most seasons $(0.01 - 0.06\,\mu g/m^3)$, except winter where daytime concentrations reached $\sim 0.1\,\mu g/m^3$. In both modes, changes in mass median diameters varied temporally and in magnitude with seasons, indicating a stronger influence of specific meteorological conditions on the properties of nitrate-containing particles at the suburban site. At the urban site, periods of greater inorganic species mass concentrations were more likely to be caused by increases in both Aitken and accumulation mode particle mass in summer, indicating that particle formation and growth affecting smaller particles was more likely to occur in the warmer season. At the suburban receptor location, significant correlation of submode particle mass to total species concentration $(0.5 \leq R_{pr} \leq 0.7)$ was observed for organics and nitrate in most seasons (spring, summer, winter)

suggesting notable influence of local or regional formation processes on organic and nitrate Aitken mode particulate mass. Variations in particle mixing state were examined by evaluation of inter-species mass median diameter trends at both measurement sites. In the accumulation mode at the urban site, internal mixing appeared to be prevalent in spring, while greater frequency of time periods with external mixing of particle populations comprising different fractions of organic constituents was observed in summer. External mixing was predominant in the Aitken mode at the urban location in both seasons. At the suburban site, sulfate and nitrate in the accumulation mode more frequently exhibited differing particle size distributions in all seasons signifying a greater extent of external mixing. In winter, external mixing of more organics enriched particles in the lower accumulation size range was evident.

**Data availability**

The data is available upon request. To obtain the data, please contact Chak K. Chan (chak.k.chan@cityu.edu.hk).

**Competing interests**

The authors declare that they have no conflict of interest.

**Acknowledgements**

This work was supported by the Environmental Conservation Fund of Hong Kong (project number ECWW09EG04).
Chak K. Chan gratefully acknowledges the startup fund of the City University of Hong Kong.

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
