# Peer review of "Diurnal and day-to-day characteristics of ambient particle mass"

_Atmospheric Chemistry and Physics, 2017_

## Referee Comment (RC1) · Anonymous Referee #1 · 19 Jun 2017

This manuscript presented a detailed analysis of a large data set of size-resolved particle composition measured by HR-AMS in Hong Kong. Both long-term trends and diurnal variations of the mass size distributions of submicron organic material, sulfate, and nitrate are discussed on the basis of previous understanding about the sources. Variations in the particle mixing state are also evaluated. This is perhaps the first study that looked at long-term AMS mass size distributions systematically, which potentially may serve as a good example of utilizing such data to derive better understanding of the sources and the atmospheric processing of submicron particles. The current

manuscript has however not yet arrived there. My main suggestions are (1) to jus-
tify the possible bias of the deconvolution of the mass size distributions ((Bian et al.,
2014)maybe exclude some data; see my comment #1) and (2) to make a clear differ-
ence on which results are novel and which ones have already been published from
previous analysis. Also, some figures contain too much information and hence are dif-
ficult to read. I therefore think a major revision or a resubmission is needed before this
paper being accepted as a publication on ACP.

Specific comments:

(1) Regarding the analysis method (Section 2.2):

a) As shown in Figure D1, Aitken-mode peak often occurred in the left tail of the
Accumulation-mode peak. If the heights of the two peak differ a lot (for example, one
as only a few percent of the other one), it is easy to overfit the small peak, which may
cause large uncertainties in quantifying the small peak. It is unclear to me how such
overfitting is controlled in this study. Are any of the data points in Figures 1, 2, and 5
subject to this possibility?

b) I disagree that the transmission efficiency of the AMS lens unlikely affects the pre-
sented analysis (Line 122-126). Because the particle velocity calibration only spans for
a certain range, extrapolation of fit may lead overestimation of the mass size distribu-
tions in small sizes. Slow vaporization and bounce may lead overestimation at Dva > 1
$\mu$m (Ref.: http://cires1.colorado.edu/jimenez-group/wiki/index.php/AMSUsrMtgs, Best
Practices: IE and Velocity Calibrations - Ed Fortner & John Jayne). More importantly,
the transmission efficiency for standard AMS lens drops at $\sim$ Dva> 400 nm or < 100
nm, and is below 20% for Dva > 1 $\mu$m or < 60 nm (Liu et al., 2007). The transmission
therefore skews the size distributions. Zhang et al. (2005) showed that in Pittsburgh
when the AMS suggests an accumulation mode at 500 nm, MOUDI shows a peak at
900 nm Dva. In this case, the fitting to AMS distributions might miss the main mode.
Bian et al. (2014) showed that sulfate and nitrate etc. indeed occurred in mode size

much greater than 500 nm Dva in Hong Kong. Given all the reasons, for urban area that has larger accumulation mode, I think the parameters (GSD, MMD, and integrated area) from fitting to the right peak (e.g., in Figure D1) cannot represent the actual accumulation-mode distributions. Ideally, the size distributions can be corrected for transmission efficiency (at least for the right side). But it is very difficult to obtain the transmission efficiency for a specific AMS with standard vaporizer. I suggest the authors to justify their accumulation-mode analysis by additional data (e.g., from SMPS or MOUDI) or improved algorithm. Otherwise, it may not be meaningful to discuss the accumulation-mode changes.

(2) Figures 1 and 2: The discussions in page 5-11 are difficult to follow by reading those figures. For example, diurnal profiles for four gaseous pollutants have no size dependence. Showing them twice with the two particle modes is very confusing. Similarly, the shaded diurnal profiles for total submicron mass of different species made the figures difficult to read. I suggest to move those into a separate figure.

(3) The numbering of section 3.1, 3.2, and 3.2.1 seems wrong.

(4) Line 157: "median values" - it is better to clarify in the captions of Figures 1 and 2 what are the medians.

(5) Line 161: What is "residual traffic"?

(6) Line 168: The abbreviations only need to be defined when the full terms first appear. Same in figure captions.

(7) Line 191: What is "residual organic particle mass"?

(8) Line 186 and Line 215: Figures do not appear in order.

(9) Line 217-219: The smaller fraction of Aitken-mode to the total increase may be caused by a greater accumulation mode contribution. In the summer, we expect to have more SOA in general (stronger emissions of the precursors and stronger oxidation), which also may lead increased organic submicron particle mass.

(10) Line 244-247: The matching of ozone and sulfate is not enough to prove that the nighttime sulfate peak is contributed by heterogeneous SO2 oxidation by ozone. Are there any other evidence to support (PMF aqueous-processing factor, results from other studies, and so on)?

(11) Line 273-274: While the median MMD seem showing little change, the mean and 25th\75th percentiles show significant diurnal variations (Figure D2). Why? Also, although in Line 103-105, there is a bit information about the diurnal distributions. Figure D2 would confuse readers a lot by the ranking of the values (meaning that medians were not located between 25th and 75th). It is important to clarify what the median, mean, and 25th\75th stand for? I mean not the median values of MMD values but the MMD from a reconstructed distribution, right?

(12) Overall the discussion in Section 3 only focused on what were seen from this study. Do the interpretations agree or disagree with what are known from other studies (other than AMS). For example, for mixing state, are the findings here consistent with the understanding from single particle analysis? The paper needs to show which results are novel and which ones have already been published from previous analysis in terms of understanding the sources and atmospheric processing of submicron particles.

Technical remarks: Line 107: Extra period after "the world". Line 157: Add hyphen between "Aitken" and "mode" when used as adjective. Similarly for "accumulation-mode particle concentrations" and so on.

Reference:

Bian, Q., Huang, X. H. H., and Yu, J. Z.: One-year observations of size distribution characteristics of major aerosol constituents at a coastal receptor site in Hong Kong - Part 1: Inorganic ions and oxalate, Atmos. Chem. Phys., 14, 9013-9027, 10.5194/acp-14-9013-2014, 2014.

Liu, P. S. K., Deng, R., Smith, K. A., Williams, L. R., Jayne, J. T., Canagaratna, M. R.,

[Figure]

Moore, K., Onasch, T. B., Worsnop, D. R., and Deshler, T.: Transmission efficiency of an aerodynamic focusing lens system: Comparison of model calculations and laboratory measurements for the Aerodyne Aerosol Mass Spectrometer, Aerosol Sci. Technol., 41, 721-733, 10.1080/02786820701422278, 2007.

Zhang, Q., Canagaratna, M. R., Jayne, J. T., Worsnop, D. R., and Jimenez, J. L.: Time- and size-resolved chemical composition of submicron particles in Pittsburgh: Implications for aerosol sources and processes, J. Geophys. Res., 110, D07s09, 10.1029/2004jd004649, 2005.

---

## Referee Comment (RC2) · Anonymous Referee #2 · 20 Jun 2017

This manuscript reports a systematic study on the long term chemically-resolved size distribution data measured by a high-resolution AMS from an urban and a suburban location in Hong Kong. Measured size distributions of individual species were fitted using a bimodal lognormal model and the derived mode sizes and submode concentrations were analyzed for seasonal and diurnal variations. Based on these results, the authors discussed the influences of different sources on aerosol sizes, differences between urban and suburban aerosols, and variations in aerosol mixing states.

The work reported in this ms is technically sound and interesting and the synthesis of

long-term AMS size distribution data is a novel undertaking. However, the assumption that all aerosol size distributions are bimodal appears to be overly simplified and somewhat arbitrary. Urban particles, in particular, are contributed by various primary and secondary sources and particle from different sources tend to have different size distributions. Although I could see the benefit of simplifying the complexity by using a bimodal assumption, it would be helpful that the authors elaborate a bit more on the justification for this treatment and provide more details on how well the biomodal log-normal model perform in fitting the observation data. Maybe a more systematic evaluation of the quality of fit for the size distribution data is more appropriate than one example (Fig. D1). Also, I would like mention that more sophisticated methods, such as the 3-dimentional factor analysis reported in Ulbrich et al. (2012), maybe useful to explore the number of modes. I also notice that the naming of the size modes in this work is a bit confusing. Aitken mode refers to particles smaller than 100 nm in diameter. However, according to Fig. 1 and 2, the mode diameters for the so-called "Aitken mode" determined through bimodal log-normal fitting are all above 100 nm, some even reaching 200 nm. Additionally, the discussions on diurnal variations of aerosol size mode focus very much on the impacts of emissions sources and physical and chemical processes. However, changes of air masses due to wind shifts or upwind impacts could also be important and should be evaluated.

Following are detailed comments:

The numbering of the sections does not seem logic. For example, according to content, 3.2.1 is parallel to 3.2.

Line 11 - 12: this sentence is difficult to understand, consider to revise.

Line 63, the AMS lens transmission is close to zero for particles smaller than 30-35 nm, so it is not precise to say the Aitken mode particles (10-100 nm) are covered by AMS.

Line 65 states that the AMS particle size data from ambient measurements are rarely

investigated in depth. This is not true. A number of studies, including a few from more than 10 years ago, analyzed the size-resolved composition data from AMS quite extensively and utilized the information to elucidate aerosol sources, new particle formation and growth mechanisms, and other atmospheric processes. Several references (not the complete list) are provided at the end of this comment in the reference section. In addition, Ulbrich et al. (2014) reported a comprehensive study on the size-resolved mass spectral data from an ambient study using 3-D factorization models. Considering that this manuscript focuses on AMS size distribution data, I'd like to recommend that the authors provide a background review on previous works in the introduction. Additionally, I notice that citations are sometime missing when findings from the authors' own research group are mentioned. This could cause confusion when the results from this work alone are sufficient to support the claim. A thorough check for in-text citations is recommended.

Line 91, what's the RH at the exit of the dryer?

Line 130 - 132, the sentence "Utilizing ..." is vague, consider to revise.

Line 160, "sweep-out" by what, rain?

Line 161, what's residual traffic?

Line 167 – 169, clarify what the decreases correspond to.

Line 170- 173, are there SMPS measurements to support the increase of particle number concentrations?

Line 172, where does the cutoff size of 50 nm come from?

Line 174 – 190, how important was COA in Aitken mode around noon time? What about contributions from secondary aerosol formation and other primary sources such as HOA? Summertime SOA and SIA formation tend to be higher and can influence particles in all size modes. I am not sure change of cooking behavior was the only reason for the different diurnal shapes between spring and summer.

Line 189 – 190, this sentence is somewhat confusing

Line 213-215, this sentence is confusing. Please clarify.

Line 215-216, what does "nucleation of gas-phase emissions" mean?

Line 222-223, "reduce nucleation… of more volatile exhaust component on fresher, smaller particles…"? Did nucleation ever occur with the volatile component in the atmosphere?

Line 244-247, this discussion seems somewhat speculative. Are there data to support the nighttime heterogeneous oxidation of SO2 by O3 in Hongkong during spring time? Has this issue been investigated in previous publication(s)? Did wind direction or air mass origin play a role in the observed size mode change?

Line 437, "particles containing different species were similar in size" is confusing.

Line 449, small particles are not just processed by condensational growth and coagulation. In the presence of high humidity, they can also go through aqueous-phase processing.

Fig. D6, can wind data be provided as well?

References:

Allan et al. (2003), Quantitative sampling using an Aerodyne Aerosol Mass Spectrometer. Part 2: Measurements of fine particulate chemical composition in two UK Cities, Journal of Geophysical Research-Atmospheres, 108(D3), 4091

Alfarra et al. (2004), Characterization of urban and regional organic aerosols in the lower Fraser Valley using two Aerodyne Aerosol Mass Spectrometers, Atmospheric Environment, 38, 5745–5758 Drewnick et al. (2004), Measurement of ambient aerosol composition during the PMTACS-NY 2001 using an Aerosol Mass Spectrometer. Part II: Chemically speciated mass distributions, Aerosol Science & Technology, 38(S1), 104-117.

Zhang et al. (2004), Insights into the chemistry of new particle formation and growth events in Pittsburgh based on Aerosol Mass Spectrometry, Environmental Science & Technology, 38(18), 4797-4809

Ulbrich et al. (2012), Three-dimensional factorization of size-resolved organic aerosol mass spectra from Mexico City, Atmospheric Measurement Techniques, 5, 195-224, doi:10.5194/amt-5-195-2012.

Setyan et al. (2014), Chemistry of new particle growth in mixed urban and biogenic emissions: insights from CARES, Atmos. Chem. Phys., 14, 6477-6494

Sun, Y et al. (2016), Primary and secondary aerosols in Beijing in winter: sources, variations and processes, Atmos. Chem. Phys., 2016(16), 8309-8329, doi:10.5194/acp-16-8309-2016.

---

## Author Response (AR1)

We thank the referee for his/her time to provide us with extensive and valuable input. Please find below our responses to the raised comments, questions and suggestions. In the following, raised **comments / suggestions are in red** and respective **responses in green**, while **alterations to the manuscript text are indicated in blue.**

**General Comment**

This manuscript presented a detailed analysis of a large data set of size-resolved particle composition measured by HR-AMS in Hong Kong. Both long-term trends and diurnal variations of the mass size distributions of submicron organic material, sulfate, and nitrate are discussed on the basis of previous understanding about the sources. Variations in the particle mixing state are also evaluated. This is perhaps the first study that looked at long-term AMS mass size distributions systematically, which potentially may serve as a good example of utilizing such data to derive better understanding of the sources and the atmospheric processing of submicron particles. The current manuscript has however not yet arrived there. My main suggestions are (1) to justify the possible bias of the deconvolution of the mass size distributions (Bian et al., 2014) maybe exclude some data; see my comment #1) and (2) to make a clear difference on which results are novel and which ones have already been published from previous analysis. Also, some figures contain too much information and hence are difficult to read. I therefore think a major revision or a resubmission is needed before this paper being accepted as a publication on ACP.

We provide a discussion with respect to (1) and (2) further below in the specific comment section. In terms of figure complexity, we understand that the provided information density per figure is high. We are thus providing a revised set of figures. The main data presented in this work have not been published before, but are based on further in-depth analysis of ambient datasets which have been discussed previously.

**Specific Comments**

**Comment**      (1) As shown in Figure D1, Aitken-mode peak often occurred in the left tail of the Accumulation-mode peak. If the heights of the two peak differ a lot (for example, one as only a few percent of the other one), it is easy to overfit the small peak, which may cause large uncertainties in quantifying the small peak. It is unclear to me how such overfitting is controlled in this study. Are any of the data points in Figures 1, 2, and 5 subject to this possibility?

**Response**     In the vast majority of analyzed size distributions the bimodal characteristics were very obvious, and unimodal fitting of such distributions would lead to large positive residuals in the lower size region, justifying the fitting of a second peak.

In extreme cases, as the reviewer notes, the difference in peak height (and peak area) between the two fitted modes can be large. When the ratio of Aitken to accumulation mode was very small (<10%), we evaluated both uni- and bimodal peak fits for which we depict one example such an extreme case below (Figure R1). In this case, from the day-to-day size distribution set (suburban HKUST site, fall, SO4), the peak area ratio of Aitken to accumulation mode was ~2%. The lognormal peak fittings are presented in subpanel (a) for the unimodal case and (b) for the bimodal case. Panels (c) and (d) show histograms of the residuals from both cases. Panel (e) shows the cumulative probability distribution for the residuals from both fittings and the calculated D value from the Kolmogorov-Smirnov Test at 95% CI. It is obvious that even in such an extreme case, the bimodal fitting resolve the raw distribution more accurately with a quasi-normal residual distribution and cumulative probability density function. In contrast, the unimodal fitting exhibits a considerably skewed residual distribution that is tail-heavy towards larger positive concentration values and centered towards negative residuals. We obtained similar results for all borderline cases in our measurements and thus generally opted for bimodal deconvolution. We are therefore confident that overfitting of the small peak was unlikely to be a major issue in this work, but we agree that this question is indeed relevant and we will include a more detailed discussion in the methodology sections of the main text and supplementary material for reference in the revised manuscript. The concerned discussion is appended at the bottom of this document.

With regard to the peak fit process itself, Igor's multipeak fitting tool uses the Levenberg-Marquardt algorithm to adjust the fit parameters with the goal to minimize the sum of squares of the deviations as an iterative process starting from the provided initial guesses. The standard deviations of the fit parameters provide an estimate of the variability of parameters between the final fit solution and the surrounding solution space with similar, but slightly larger residuals. We provide a discussion of these analyses in the revised manuscript and Supporting Material.

| | |
|---|---|
| **Alteration** | Revision and addition to Section B of the Supporting Material and associated Tables (C1-C3) and Figures (D2-D5). The relevant sections are appended at the bottom of this document. |
| **Comment** | (2) I disagree that the transmission efficiency of the AMS lens unlikely affects the presented analysis (Line 122-126). Because the particle velocity calibration only spans for a certain range, extrapolation of fit may lead overestimation of the mass size distributions in small sizes. Slow vaporization and bounce may lead overestimation at $D_{va} > 1$ µm (Ref.: http://cires1.colorado.edu/jimenez-group/wiki/index.php / AMSUsrMtgs, Best Practices: IE and Velocity Calibrations - Ed Fortner & John Jayne). More importantly, the transmission efficiency for standard AMS lens drops at Dva> 400 nm or < 100 nm, and is below 20% for Dva > 1 µm or < 60 nm (Liu et al., 2007). The transmission therefore skews the size distributions. Zhang et al. (2005) showed that in Pittsburgh when the AMS suggests an accumulation mode at 500 nm, MOUDI shows a peak at 900 nm Dva. In this case, the fitting to AMS distributions might miss the main mode. Bian et al. (2014) showed that sulfate and nitrate etc. indeed occurred in mode size much greater than 500 nm Dva in Hong Kong. Given all the reasons, for urban area that has larger accumulation mode, I think the parameters (GSD, MMD, and integrated area) from fitting to the right peak (e.g., in Figure D1) cannot represent the actual accumulation-mode distributions. Ideally, the size distributions can be corrected for transmission efficiency (at least for the right side). But it is very difficult to obtain the transmission efficiency for a specific AMS with standard vaporizer. I suggest the authors to justify their accumulation-mode analysis by additional data (e.g., from SMPS or MOUDI) or improved algorithm. Otherwise, it may not be meaningful to discuss the accumulation-mode changes. |
| **Response** | We agree with the reviewer that lens transmission is an important issue in AMS-related work in general and represents a key instrumental limitation, which affects the vast majority of AMS publications as lens transmission corrections are not commonly performed. |
| | The main aim of our work however is to present an analysis method that can provide an additional dimension to standard AMS data analysis techniques, given that bi- or multimodal size distributions from AMS measurements have been reported frequently in the literature. In our study, we focus on discussing the trends of mode diameters and mode particle mass (peak area) to provide additional, complementing information to preceding studies that only utilized standard mass concentration based analyses. |
| | Lens transmission efficiencies have been reported to vary between instruments (even at the same pressure level) and a standard lens as used in this study is expected to transmit efficiently (~100%) between 90nm and 700nm $D_{va}$, thereafter decreasing to ~0.3 at 1000nm (Williams et al., 2013). The largest observed mode diameters in the accumulation mode in our work at either sampling location were ~700nm ($D_{va}$). The referenced size distribution work (Bian et al., 2014) relies on MOUDI samples and showed mode diameters ~800-900nm of aerodynamic diameter $D_a$, which if we assume $D_a \sim D_p$ and a particle density of ~ 1.5 g/cm$^3$ is beyond the transmission capability of the AMS (with $D_{va} \sim 1.4$ µm) in the droplet mode range. We note that comparability of results from MOUDI studies and AMS is limited given the different sizing techniques, sampling times (minute resolution vs. daily samples) and more importantly aerosol pretreatment i.e. "as-is" for MOUDI vs. removal of water prior to AMS measurements, which can influence particle size in high humidity (>80%) conditions (Fang et al., 1991). |
| | While we agree that additional particle sizing instrumentation for inter-instrumental comparison are useful, the measurements presented in this study were conducted with a limited set of available instruments and did unfortunately not encompass complementary particle size distribution measurements (by either electrostatic classification or MOUDI samplers). |
| | We revised the statements in line 122-126 and further stress the aims and scope of our work as well as the definition of the Aitken and accumulation mode as representing the apparent Aitken- and accumulation-mode contributions to AMS-measurable particle mass (i.e. within the capabilities and limitations of AMS as an ambient analytical instrument). We still view that the presented work is useful to the growing AMS community to offer additional dimensions in the analysis of AMS size distribution data. |
| **Alteration** | The transmission efficiency of the AMS aerodynamic lens is known to fall off beyond ~0.7 µm of vacuum-aerodynamic diameter (Liu et al., 2007;Takegawa et al., 2009;Zhang et al., |

2004;Bahreini et al., 2008;Williams et al., 2013) and may bias measured particle mass and mode diameters in the accumulation mode towards lower values if significant particle mass fractions fall in the size region of $D_{va} > 0.7$ µm. Resolved MMDs at either sampling location were typically within the efficient upper transmission limit in this work.

The discussion of size distributions in this work should be viewed in the context of the instrumental capabilities and limitations of aerosol mass spectrometry, i.e. resolved Aitken and accumulation modes in this work are understood to represent the apparent Aitken and accumulation modes within AMS measurable particle mass size distributions.

**Comment**    Figures 1 and 2: The discussions in page 5-11 are difficult to follow by reading those figures. For example, diurnal profiles for four gaseous pollutants have no size dependence. Showing them twice with the two particle modes is very confusing. Similarly, the shaded diurnal profiles for total submicron mass of different species made the figures difficult to read. I suggest to move those into a separate figure.

**Response**    Gas data are shown in duplicate (noted in the figure caption) to enable direct eye-guided comparison of concentration trends in both panels. We have revised the figures for more intuitive readability.

**Alteration**    Figures 1 and 2 (now Figures 2 and 3 in the revised manuscript) have been replotted.

**Comment**    The numbering of section 3.1, 3.2, and 3.2.1 seems wrong.

**Response**    The numbering of sections in chapter 3 is erroneous. Part 3.2. is in fact 3.1.1., while 3.2. is 3.1.2. This also affects section numbers thereafter, and we provide a corrected chapter numbering in the revised manuscript.

**Comment**    Line 157: "median values" - it is better to clarify in the captions of Figures 1 and 2 what are the medians.

**Response**    We state more clearly in the text and figure captions that median refers to values from the bin-median size distribution in the revised manuscript.

**Comment**    Line 161: What is "residual traffic"?

**Response**    We employ the term "residual traffic" in analogy to "background" concentration levels, as traffic in the central inner-city districts remains continuous at night albeit at much lower vehicle number compared to the daytime, whereas in more remote areas or smaller cities traffic at night is typically intermittent. We will change the wording to avoid confusion.

**Alteration**    […] as well as contributions from nighttime activity such as traffic, which remains continuous in the inner-city districts at night albeit at much lower vehicle numbers compared to the daytime.

**Comment**    Line 168: The abbreviations only need to be defined when the full terms first appear. Same in figure captions.

**Response**    We will remove duplicate definitions in the revised manuscript.

**Comment**    Line 191: What is "residual organic particle mass"?

**Response**    This term has been used in a context similar to the discussion of "residual traffic", and refers to any background organic aerosol contributions as well as local contributions that are not immediately removed by settling or sweep-out. We agree that this may be confusing.

**Alteration**    In the accumulation mode, organic particle mass during the night hours (00:00 – 06:00) was 2.5 times larger in spring (5.5 µg m$^{-3}$) than in summer (2.0 µg m$^{-3}$).

**Comment**    Line 186 and Line 215: Figures do not appear in order.

**Response**    In this manuscript, Figures have been arranged to enable the reader to compare the results from the two different sites. The manuscript text also follows this general structure. Given the size of the plots, some subfigures had to be grouped into separate plots and therefore may not appear strictly in numerical order in the manuscript text.

**Comment**    Line 217-219: The smaller fraction of Aitken-mode to the total increase may be caused by a greater accumulation mode contribution. In the summer, we expect to have more SOA in general (stronger emissions of the precursors and stronger oxidation), which also may lead increased organic submicron particle mass

**Response** We agree with the reviewer that SOA influence may be a viable explanation for the increased accumulation mass contribution, however, our measurement data do not support this.
Changes in SOA concentrations during the daylight hours were small in both seasons (Lee et al., 2015). We also note that accumulation mode particle mass increases were smaller in summer than spring, i.e. even with the likelihood of stronger oxidation conditions in summer the corresponding SOA formation did not seem to lead to significant enhancements in measured organic submicron particle mass.

**Comment** Line 244-247: The matching of ozone and sulfate is not enough to prove that the nighttime sulfate peak is contributed by heterogeneous $SO_2$ oxidation by ozone. Are there any other evidence?

**Response** The limited amount of additional measurement data beyond gas-phase standard criteria pollutant data prevents a deeper analysis of this remarkable observation in this work.
We believe that the current wording in the manuscript clarifies that the observed trends are indications ("this points to…") and can serve as impetus for further future study.

**Comment** Line 273-274: While the median MMD seem showing little change, the mean and 25thn75th percentiles show significant diurnal variations (Figure D2). Why? Also, although in Line 103-105, there is a bit information about the diurnal distributions. Figure D2 would confuse readers a lot by the ranking of the values (meaning that medians were not located between 25th and 75th). It is important to clarify what the median, mean, and 25thn75th stand for? I mean not the median values of MMD values but the MMD from a reconstructed distribution, right?

**Response** The reviewer's interpretation of the different size distribution groups is correct. We have added an additional Figure 1 in the main text to illustrate the origin of these size distributions to better guide the reader in the discussions following later on. The Figure is appended at the bottom of this document.
In Figure D2, left lower panel for nitrate, there is indeed a notable diurnal variation in the mean set MMD. The 25th and 75th percentile distributions exhibit certain fluctuations, which are however minor and within a narrow range. We further take note of this observation in Lines 293-298, where the differences in distribution sets are discussed. We believe that this difference between the percentile (25th PC, median, 75thPC) sets and the mean set is an effect of averaging same-hour concentration values from different days to yield the diurnal average values. As different days may experience different concentration levels (longer-term fluctuations, e.g. transport, photochemically active periods etc.) and may be distributed disproportionately across certain hours of the day, this results in "skewed" average size distributions. As noted in the manuscript, we therefore chose to utilize median data in this study for the interpretation of diurnal variations.

**Alteration** Addition of Figure 1 to show the sequence of main data treatment and analysis steps.

**Comment** Overall the discussion in Section 3 only focused on what were seen from this study. Do the interpretations agree or disagree with what are known from other studies (other than AMS). For example, for mixing state, are the findings here consistent with the understanding from single particle analysis? The paper needs to show which results are novel and which ones have already been published from previous analysis in terms of understanding the sources and atmospheric processing of submicron particles.

**Response** Size distribution studies in Hong Kong are generally rare and are either not chemically resolved (SMPS, FMPS) or rely on MOUDI sampling.
Comparability is limited, given different sizing techniques (mobility vs. aerodynamic / vacuum-aerodynamic), sampling times (real-time or near-real time vs. 24h to 48h) and aerosol pre-treatment (removal of water for AMS measurements vs. "as-is" for MOUDI and SMPS typically). To our knowledge, single particle analysis from ambient measurements in Hong Kong analyzing particle mixing state have not been reported yet. We have added a section on particle size distribution measurements undertaken in Hong Kong in the revised manuscript.

With respect to novelty, as noted in the introduction, the main focus of this paper is to demonstrate a systematic method of utilizing AMS size distributions and to provide chemically resolved particle mass size distributions on finer temporal scales. Detailed chemically resolved diurnal size distribution variations and longer term daily size distribution measurements from ambient AMS sampling campaigns are scarce, as are detailed size distribution studies focusing on the Hong Kong and Pearl River Delta Region.

**Alteration**     Addition of Chapter 3.3 in the main text. The relevant part is appended to the bottom of this document.

**Comment**     Technical remarks: Line 107: Extra period after "the world". Line 157: Add hyphen between "Aitken" and "mode" when used as adjective. Similarly for "accumulation mode particle concentrations" and so on.

**Response**     We have taken these technical remarks into account in our manuscript revision.

**Changes in sections of main manuscript and Supporting Material**
**Main manuscript**

[Figure]

**Figure 1.** Flow chart of main data acquisition, data treatment and data analysis procedures

**[…]**

**3.3 Comparison to previous studies**

Particle size distribution studies in Hong Kong are generally scarce and have focused on either size segregated filter samples (MOUDI) for general ambient measurements or electrostatic classification in particle formation and particle growth studies (Guo et al., 2012;Cheung et al., 2015) . The latter studies focus on specific and narrow time periods and lack general discussions on ambient particle size distributions.

Two ambient studies were undertaken at the suburban coastal HKUST site using size-segregated samples from a ten-stage MOUDI sampler and offline chromatographic analysis. Inorganic constituents (NH4, NO3, SO4) in fine particles (i.e. $D_p$<1.8 µm) were shown to follow bimodal distributions with mode diameters in the range of 0.14-0.21 µm and 0.46-0.58 µm in samples collected in the winter season, while the main mode was observed in the coarse region (4-6 µm) for all three species (Zhuang et al., 1999). A subsequent year-long observational study also reported bimodal fine particle distributions with mode diameters of 0.1-0.3 µm and 0.7–0.9 µm and 1-2 additional modes in the coarse region (Bian et al., 2014), however, the main mode in the size distributions of sulfate, ammonium, potassium and oxalate was observed in the droplet mode (0.7 – 0.9 µm) in this study. Vehicle exhaust plumes sampled on-road from a Mobile Real-time Air Monitoring Platform (MAP) across Hong Kong's road network exhibited three distinct particle volume size distributions: a unimodal distribution with an accumulation mode at 0.2 µm and two bimodal distributions with a minor mode at 0.2 µm and the dominant mode at 0.5 or 0.7 µm (Yao et al., 2007).

The bimodality in the fine particle range across these studies is consistent with the AMS-based results in this work. Nominally, the accumulation mode diameters from filter based studies and the chase studies are larger than those from AMS measurements where maximum mode diameters occurred at $D_{va}$ ~ 700nm, corresponding to $D_a$ ~ 470 (assuming $D_{va}$ ~ $D_a$ * density; particle density ~ 1.5 g/cm$^3$). Direct comparability is however limited due to fundamental differences in sizing techniques (MOUDI: atmospheric pressure; AMS: near-vacuum), sampling times (MOUDI: 24h samples, scattered time line; AMS: minute raw resolution averaged to hourly or daily, continuous time line), measurement uncertainties (MOUDI: sampling artifacts such as vapor adsorption and desorption; AMS: inlet lens transmission) and aerosol pretreatment (none for MOUDI with potential impacts on particle size in high humidity (>80%) conditions (Fang et al., 1991); AMS: removal of water prior to introduction to instrument).

**Supporting Material**

Lognormal peaks were fitted to each 24h and hour-of-day AMS mass size distribution respectively employing the *Multipeak Fit V2* algorithm in *Igor Pro (Wavemetrics)* using a simple vertical offset as the baseline and initial guesses on peak position, height, and width based on visual inspection of the raw size distribution. The multipeak fitting tool employs the Levenberg-Marquardt algorithm (Gill et al., 1981) as a non-linear least squares fit and iteratively adjusts the initial fit parameter guesses until a convergent solution with minimized residuals is achieved. In sporadic cases, the fitted solution led to excessive deviations from the initial guesses with greatly shifted peak locations and large fluctuations in peak width. In such cases, results from the peak fits of immediately adjacent size distributions (i.e. previous and next distributions in the sequence) were evaluated and used to adjust the fitting process by fixing either the location (*primary*) or the width of the peak (*secondary*) to the average value of the two adjacent fitted distributions.

For the diurnal size distributions, measurement data from time periods with large differences in species concentration levels were pooled together. The averaging of mass (or volume) based size distribution involves different uncertainties for each size bin due to the cubic relationship between particle mass (or volume) and particle diameter and the corresponding improvement in signal-to-noise ratio with increasing particle size. To establish reliable diurnal trends, we adopt an approach similar to the analysis of conventional species concentration diurnal trends by evaluating size distributions reconstructed from the average, median, $25^{th}$ and $75^{th}$ percentile of each size bin. Similar diurnal trends in the fitting parameters across these different size distributions would confirm that changes were indeed recurrent daily while divergent trends would indicate that irregular processes (e.g. episodic events) were more significant in determining size distribution characteristics. Since episodic pollution events and clean periods (e.g. prolonged precipitation) were not removed from the dataset, the quantitative analysis focuses on trends observed in the median dataset to minimize skewing effects of high and low concentration periods.

Uncertainties can arise from the peak fitting process itself. While the bimodality of the size distributions was obvious in most cases (i.e. a main mode with a shoulder towards smaller particle sizes, e.g. Figure D1), accumulation mode particle mass can occasionally dominate the mass size distribution and diminish the Aitken mode. To achieve confidence in the appropriateness of the bimodal fitting we evaluated both unimodal and bimodal peak fits whenever the Aitken to accumulation mode peak ratio was <10% and we depict a representative example below (Figure D2a. b). The distribution of the fit residuals (Figure D2c, d) was examined and cumulative probability distributions of the fit residuals compared by the Kolmogorov-Smirnov test (Figure D2e) to assess whether fit residuals were significantly different at 95% confidence level (CL). It is evident that the bimodal fit performs better at resolving the raw size distribution in the smaller size region and overall yields a more normal residual distribution. The Kolmogorov-Smirnov test confirms that the residual distributions are statistically different ($D>D_{critical}$ at 95% CL). We tested all borderline cases using the outlined procedure. In this study, bimodal fits yielded unanimously better results in all cases for both diurnal and day-to-day size distributions and all investigated species, i.e. the Aitken mode always remained clearly distinguishable from the accumulation mode.

While the peak fitting algorithm yields a unique individual solution with a set of parameters for which resulting residuals (*difference of fitted and original distribution*) are minimized, the surrounding solution space provides a potentially infinite number of similar solutions with slightly larger residuals. The standard deviations of the fit parameters can provide an estimate of the variability of the peak parameters between the final fit solution and the surrounding solution space. We evaluated the uncertainty in peak area (i.e. integrated mode particle mass) which represents the combined uncertainty of the peak position, width and height (which altogether directly determine the peak area) for all fitted size distributions in this work.

Figure D3 depicts the standard deviation of resolved peak area (i.e. integrated mode particle mass concentration) nominally and relative to the peak area for the diurnal size distributions of NO3 at the urban Mong Kok site in summer 2013 and Tables C1-C2 summarize the values of percent standard deviations for all species at both measurement sites respectively. The median datasets, which were used for quantitative discussion for the diurnal size distribution analysis, exhibited particle mass uncertainties of 14-48% in the Aitken mode and 1-12% in the accumulation mode at the suburban HKUST site, and 7-44% in the Aitken mode and 1-6% in the accumulation mode at the urban MK site. Figure D4 depicts the 75[th] percentile-bin diurnal variation of NO3 (which displayed the largest uncertainties in Figure D3) with the corresponding peak area variability, and shows that the interpretation of the diurnal variation would remain largely unaffected from the incurred uncertainties.

For the day-to-day 24h size distributions a corresponding analysis was undertaken, with Figure D5 depicting the size distributions of NO3 at the HKUST site for all covered seasons exemplarily, and Table C3 summarizes the values of percent standard deviations for all species at both measurement sites respectively. Peak fit uncertainties typically increase with decreasing integrated peak area and can exceed the values of the peak area in the Aitken mode in a small number of cases (e.g. Figure D5c,e – ratios >1). Quantification of the Aitken mode may not be possible at high levels of confidence in these isolated cases. They were retained in the dataset due to their low frequency of occurrence and to enable a complete discussion over the full concentration range without biasing towards larger concentration (i.e. fitted peak areas) values.

**Table C1.** Percentiles of relative standard deviation (*rows; corresponding to the box-whiskers plot in Figure D3e,f*) in percent from lognormal peak fits (bimodal deconvolution) for the resolved Aitken mode (a) and accumulation mode (b) particle concentration for diurnal size distributions at the HKUST supersite (2011/12), *columns* describe the data set, i.e. reconstructed size distributions from the 25th percentile, median, 75th percentile and mean of the size bins
*Sp=Spring, Su=Summer, Fa=Fall, Wi=Winter*

(a)

| Aitken mode % SD | | 25th PC Distr. | | | | Median Distr. | | | | 75th PC Distr. | | | | Mean Distr. | | | | Range |
|---|---|---|---|---|---|---|---|---|---|---|---|---|---|---|---|---|---|---|
| | | Sp | Su | Fa | Wi | Sp | Su | Fa | Wi | Sp | Su | Fa | Wi | Sp | Su | Fa | Wi | |
| **NO3** (UST) | PC-90 | 67 | 76 | 39 | 85 | 76 | 44 | 54 | 55 | 80 | 75 | 56 | 39 | 36 | 77 | 44 | 42 | 36-85 |
| | PC-75 | 52 | 42 | 28 | 57 | 66 | 32 | 39 | 40 | 61 | 59 | 44 | 29 | 22 | 38 | 35 | 34 | 22-66 |
| | **PC-50** | **36** | **33** | **22** | **46** | **44** | **26** | **31** | **22** | **40** | **43** | **34** | **23** | **18** | **28** | **24** | **30** | **18-46** |
| | PC-25 | 29 | 25 | 16 | 28 | 26 | 14 | 24 | 18 | 25 | 29 | 22 | 17 | 14 | 19 | 20 | 25 | 14-29 |
| | PC-10 | 21 | 20 | 15 | 24 | 19 | 12 | 19 | 15 | 17 | 19 | 17 | 10 | 13 | 13 | 15 | 21 | 10-24 |
| **SO4** (UST) | PC-90 | 38 | 38 | 42 | 74 | 19 | 36 | 39 | 43 | 22 | 40 | 40 | 42 | 19 | 36 | 37 | 81 | 19-81 |
| | PC-75 | 35 | 33 | 38 | 55 | 18 | 28 | 33 | 32 | 18 | 30 | 35 | 64 | 17 | 30 | 30 | 66 | 17-66 |
| | **PC-50** | **28** | **30** | **33** | **32** | **16** | **26** | **27** | **25** | **14** | **24** | **27** | **27** | **14** | **24** | **26** | **48** | **14-48** |
| | PC-25 | 21 | 25 | 26 | 24 | 13 | 21 | 22 | 21 | 11 | 18 | 24 | 22 | 12 | 21 | 23 | 40 | 11-40 |
| | PC-10 | 17 | 23 | 24 | 19 | 11 | 20 | 20 | 13 | 10 | 14 | 20 | 19 | 10 | 19 | 22 | 35 | 10-35 |
| **Org** (UST) | PC-90 | 52 | 23 | 41 | 44 | 42 | 28 | 32 | 27 | 47 | 48 | 45 | 53 | 46 | 29 | 24 | 32 | 23-53 |
| | PC-75 | 41 | 18 | 26 | 28 | 30 | 22 | 27 | 22 | 26 | 39 | 35 | 44 | 36 | 23 | 21 | 25 | 18-44 |
| | **PC-50** | **26** | **14** | **17** | **21** | **19** | **18** | **21** | **17** | **20** | **32** | **26** | **35** | **23** | **18** | **19** | **21** | **14-35** |
| | PC-25 | 16 | 11 | 13 | 18 | 17 | 16 | 19 | 15 | 18 | 28 | 20 | 29 | 18 | 14 | 16 | 17 | 11-29 |
| | PC-10 | 9 | 9 | 10 | 11 | 14 | 9 | 15 | 12 | 15 | 20 | 17 | 21 | 17 | 11 | 14 | 16 | 9-21 |

(b)

| Accum. mode % SD | | 25th PC Distr. | | | | Median Distr. | | | | 75th PC Distr. | | | | Mean Distr. | | | | Range |
|---|---|---|---|---|---|---|---|---|---|---|---|---|---|---|---|---|---|---|
| | | Sp | Su | Fa | Wi | Sp | Su | Fa | Wi | Sp | Su | Fa | Wi | Sp | Su | Fa | Wi | |
| **NO3** (UST) | PC-90 | 8 | 5 | 8 | 4 | 7 | 5 | 6 | 3 | 4 | 9 | 6 | 3 | 3 | 4 | 5 | 1 | 1-9 |
| | PC-75 | 6 | 4 | 7 | 3 | 3 | 4 | 5 | 2 | 4 | 7 | 5 | 2 | 2 | 3 | 5 | 1 | 1-7 |
| | **PC-50** | **4** | **3** | **5** | **2** | **3** | **3** | **5** | **2** | **3** | **4** | **5** | **2** | **2** | **3** | **4** | **1** | **1-5** |
| | PC-25 | 3 | 3 | 4 | 2 | 2 | 3 | 4 | 1 | 3 | 3 | 4 | 2 | 2 | 2 | 4 | 1 | 1-4 |
| | PC-10 | 2 | 2 | 4 | 1 | 2 | 2 | 4 | 1 | 2 | 2 | 3 | 1 | 2 | 2 | 3 | 1 | 1-4 |
| **SO4** (UST) | PC-90 | 2 | 3 | 3 | 5 | 2 | 2 | 2 | 3 | 3 | 2 | 3 | 4 | 2 | 2 | 2 | 2 | 2-5 |
| | PC-75 | 2 | 2 | 2 | 2 | 2 | 2 | 2 | 2 | 2 | 2 | 2 | 3 | 1 | 2 | 2 | 2 | 1-3 |
| | **PC-50** | **2** | **2** | **2** | **1** | **2** | **2** | **2** | **2** | **2** | **2** | **2** | **2** | **1** | **2** | **2** | **1** | **1-2** |
| | PC-25 | 2 | 2 | 2 | 1 | 1 | 2 | 2 | 1 | 1 | 2 | 2 | 2 | 1 | 2 | 2 | 1 | 1-2 |
| | PC-10 | 1 | 2 | 1 | 1 | 1 | 2 | 2 | 1 | 1 | 1 | 2 | 1 | 1 | 2 | 2 | 1 | 1-2 |
| **Org** (UST) | PC-90 | 29 | 12 | 16 | 9 | 18 | 8 | 9 | 5 | 10 | 6 | 7 | 3 | 18 | 7 | 5 | 5 | 3-29 |
| | PC-75 | 18 | 9 | 11 | 7 | 10 | 5 | 5 | 3 | 6 | 4 | 5 | 2 | 11 | 5 | 4 | 3 | 2-18 |
| | **PC-50** | **12** | **7** | **5** | **5** | **6** | **4** | **4** | **3** | **4** | **3** | **4** | **2** | **7** | **3** | **3** | **3** | **2-12** |
| | PC-25 | 7 | 4 | 3 | 3 | 5 | 3 | 3 | 2 | 3 | 2 | 3 | 2 | 5 | 2 | 2 | 2 | 2-7 |
| | PC-10 | 4 | 4 | 1 | 2 | 4 | 2 | 2 | 1 | 2 | 2 | 2 | 1 | 2 | 1 | 2 | 2 | 1-4 |

**Table C2.** Percentiles of relative standard deviation (*rows; corresponding to the box-whiskers plot in Figure D3e,f*) in percent from lognormal peak fits (bimodal deconvolution) for the resolved Aitken mode (a) and accumulation mode (b) particle concentration for diurnal size distributions at the urban MK site (2013), *columns describe the data set, i.e. reconstructed size distributions from the 25th percentile, median, 75th percentile and mean of the size bins Sp=Spring, Su=Summer*

(a)

| Aitken mode
% SD | | 25th PC Distr. | | Median Distr. | | 75th PC Distr. | | Mean Distr. | | Range |
|---|---|---|---|---|---|---|---|---|---|---|
| | | Sp | Su | Sp | Su | Sp | Su | Sp | Su | |
| NO3 (MK) | PC-90 | 38 | 46 | 40 | 34 | 40 | 41 | 22 | 26 | 22-46 |
| | PC-75 | 27 | 34 | 26 | 28 | 34 | 35 | 20 | 16 | 16-35 |
| | **PC-50** | **15** | **24** | **23** | **21** | **25** | **28** | **18** | **14** | **14-28** |
| | PC-25 | 9 | 20 | 19 | 17 | 18 | 22 | 15 | 12 | 9-22 |
| | PC-10 | 6 | 19 | 16 | 14 | 15 | 20 | 13 | 11 | 6-20 |
| SO4 (MK) | PC-90 | 63 | 46 | 35 | 38 | 24 | 31 | 23 | 21 | 21-63 |
| | PC-75 | 50 | 36 | 30 | 36 | 21 | 28 | 21 | 20 | 20-50 |
| | **PC-50** | **44** | **33** | **28** | **24** | **20** | **23** | **19** | **18** | **18-44** |
| | PC-25 | 37 | 27 | 24 | 21 | 17 | 20 | 15 | 17 | 15-37 |
| | PC-10 | 33 | 25 | 21 | 19 | 15 | 18 | 15 | 16 | 15-33 |
| Org (MK) | PC-90 | 22 | 12 | 22 | 19 | 30 | 21 | 15 | 14 | 12-30 |
| | PC-75 | 16 | 10 | 12 | 12 | 18 | 11 | 12 | 8 | 8-18 |
| | **PC-50** | **10** | **8** | **10** | **9** | **10** | **9** | **8** | **7** | **7-10** |
| | PC-25 | 8 | 7 | 8 | 7 | 7 | 8 | 6 | 6 | 6-8 |
| | PC-10 | 7 | 6 | 7 | 6 | 6 | 6 | 5 | 5 | 5-7 |

(b)

| Accum. mode
% SD | | 25th PC Distr. | | Median Distr. | | 75th PC Distr. | | Mean Distr. | | Range |
|---|---|---|---|---|---|---|---|---|---|---|
| | | Sp | Su | Sp | Su | Sp | Su | Sp | Su | |
| NO3 (MK) | PC-90 | 9 | 9 | 6 | 7 | 3 | 5 | 2 | 5 | 2-9 |
| | PC-75 | 6 | 8 | 4 | 5 | 3 | 4 | 2 | 4 | 2-8 |
| | **PC-50** | **4** | **6** | **3** | **4** | **2** | **3** | **2** | **3** | **2-6** |
| | PC-25 | 2 | 5 | 2 | 4 | 2 | 3 | 1 | 3 | 1-5 |
| | PC-10 | 2 | 4 | 1 | 3 | 1 | 3 | 1 | 2 | 1-4 |
| SO4 (MK) | PC-90 | 4 | 3 | 2 | 3 | 2 | 5 | 2 | 3 | 2-5 |
| | PC-75 | 3 | 2 | 2 | 3 | 2 | 4 | 2 | 2 | 2-4 |
| | **PC-50** | **3** | **2** | **2** | **2** | **1** | **4** | **2** | **2** | **1-4** |
| | PC-25 | 2 | 2 | 2 | 2 | 1 | 3 | 1 | 2 | 1-3 |
| | PC-10 | 2 | 2 | 1 | 2 | 1 | 3 | 1 | 2 | 1-3 |
| Org (MK) | PC-90 | 9 | 8 | 8 | 7 | 9 | 5 | 6 | 6 | 5-9 |
| | PC-75 | 6 | 7 | 5 | 5 | 5 | 4 | 5 | 4 | 4-7 |
| | **PC-50** | **5** | **5** | **4** | **4** | **3** | **4** | **3** | **4** | **3-5** |
| | PC-25 | 4 | 4 | 3 | 3 | 3 | 3 | 3 | 3 | 3-4 |
| | PC-10 | 2 | 3 | 3 | 3 | 5 | 2 | 2 | 2 | 2-3 |

**Table C3.** Percentiles of percent standard deviation (*rows; corresponding to the box-whiskers plot in Figure D5e,f*) from lognormal peak fits (bimodal deconvolution) for the resolved Aitken mode and accumulation mode for 24h day-to-day size distributions at the suburban HKUST site (a) and the urban MK site (b) for all investigated species, *columns* describe the uncertainties in terms of quartiles of resolved peak area, where Q1 refers to the lowest 25% and Q4 the highest 25% of resolved peak area (see also Figure D4)

(a)

| HKUST '11/12 % SD | | NO3 | | | | | SO4 | | | | | Org | | | | |
|---|---|---|---|---|---|---|---|---|---|---|---|---|---|---|---|---|
| | | Q1 | Q2 | Q3 | Q4 | Range | Q1 | Q2 | Q3 | Q4 | Range | Q1 | Q2 | Q3 | Q4 | Range |
| Aitken mode | PC-90 | 95 | 64 | 50 | 27 | 27-95 | 97 | 78 | 49 | 30 | 30-97 | 62 | 37 | 28 | 26 | 26-62 |
| | PC-75 | 58 | 47 | 37 | 24 | 24-58 | 91 | 57 | 36 | 20 | 20-91 | 41 | 24 | 21 | 22 | 21-41 |
| | **PC-50** | **47** | **35** | **25** | **17** | **17-47** | **60** | **39** | **29** | **16** | **16-60** | **30** | **17** | **14** | **17** | **14-30** |
| | PC-25 | 30 | 25 | 20 | 13 | 13-30 | 32 | 26 | 21 | 13 | 13-32 | 25 | 14 | 10 | 12 | 10-25 |
| | PC-10 | 20 | 16 | 15 | 8 | 8-20 | 24 | 18 | 13 | 11 | 11-24 | 16 | 11 | 6 | 7 | 6-16 |
| Accum. mode | PC-90 | 31 | 9 | 7 | 4 | 4-31 | 4 | 3 | 3 | 3 | 3-4 | 18 | 11 | 9 | 7 | 7-18 |
| | PC-75 | 13 | 6 | 5 | 3 | 3-13 | 3 | 3 | 3 | 2 | 2-3 | 9 | 7 | 6 | 5 | 5-9 |
| | **PC-50** | **8** | **4** | **3** | **2** | **2-8** | **2** | **2** | **2** | **2** | **~2** | **6** | **4** | **5** | **3** | **3-6** |
| | PC-25 | 3 | 3 | 2 | 2 | 2-3 | 2 | 2 | 2 | 2 | ~2 | 4 | 3 | 3 | 2 | 2-4 |
| | PC-10 | 1 | 2 | 2 | 1 | 1-2 | 2 | 2 | 2 | 1 | 1-2 | 3 | 2 | 2 | 2 | 2-3 |

(b)

| MK '13 % SD | | NO3 | | | | | SO4 | | | | | Org | | | | |
|---|---|---|---|---|---|---|---|---|---|---|---|---|---|---|---|---|
| | | Q1 | Q2 | Q3 | Q4 | Range | Q1 | Q2 | Q3 | Q4 | Range | Q1 | Q2 | Q3 | Q4 | Range |
| Aitken mode | PC-90 | 62 | 52 | 47 | 30 | 30-62 | 94 | 50 | 29 | 24 | 24-94 | 23 | 17 | 13 | 16 | 13-23 |
| | PC-75 | 41 | 42 | 30 | 25 | 25-42 | 47 | 35 | 22 | 16 | 16-47 | 16 | 14 | 11 | 12 | 11-16 |
| | **PC-50** | **28** | **34** | **21** | **18** | **18-34** | **33** | **24** | **17** | **12** | **12-33** | **11** | **10** | **6** | **8** | **6-11** |
| | PC-25 | 21 | 22 | 19 | 12 | 12-22 | 26 | 17 | 13 | 9 | 9-26 | 8 | 7 | 5 | 6 | 5-8 |
| | PC-10 | 6 | 10 | 13 | 8 | 6-13 | 19 | 14 | 10 | 5 | 5-19 | 6 | 5 | 3 | 3 | 3-6 |
| Accum. mode | PC-90 | 22 | 21 | 6 | 6 | 6-22 | 7 | 6 | 6 | 4 | 4-7 | 13 | 8 | 9 | 8 | 8-13 |
| | PC-75 | 17 | 10 | 5 | 3 | 3-17 | 4 | 4 | 4 | 2 | 2-4 | 8 | 7 | 6 | 4 | 4-8 |
| | **PC-50** | **10** | **6** | **4** | **2** | **2-10** | **3** | **2** | **2** | **1** | **1-3** | **6** | **5** | **4** | **2** | **2-6** |
| | PC-25 | 6 | 3 | 2 | 2 | 2-6 | 2 | 2 | 1 | 1 | 1-2 | 4 | 4 | 3 | 2 | 2-4 |
| | PC-10 | 3 | 1 | 2 | 1 | 1-3 | 1 | 1 | 1 | 1 | ~1 | 2 | 3 | 2 | 1 | 1-3 |

[Figure]

**Figure D1.** Example of a log-normal peak fit *(Multipeak Fit V2, Igor Pro, Wavemetrics, Levenberg-Marquardt algorithm)* of an AMS organics size distribution

[Figure]

**Figure D2.** 24h average size distribution of sulfate (12/12/2011, suburban HKUST site) with (a) unimodal and (b) bimodal logN peak fitting applied; histograms of residuals from the unimodal (c) and bimodal (d) distributions with Gaussian fit (green); and cumulative probability density functions of uni- and bimodal fit residuals (e) with Kolmogorov-Smirnov D metric values at 95% confidence level

[Figure]

**Figure D3.** Standard deviation of peak area as a function of mode peak area (a,b), histogram of relative standard deviation i.e. the ratio of standard deviation to mode peak area (c,d) where the last bin also contains all values beyond the last bin range, and relative standard deviation as a function of mode peak area (e,f) for the fitted Aitken and accumulation mode with binned box-whiskers plot ($25^{th}$ to $75^{th}$ PC box with horizontal median line and $10^{th}$ to $90^{th}$ PC whiskers where bins refer to quartiles of peak area from lowest Q1 to highest Q4); data for diurnal size distributions of NO3 at the urban Mong Kok site in summer 2013.

[Figure]

**Figure D4.** Plot of 75th percentile-bin diurnal variation with peak area fit variability and relative standard deviation (corresponding to green data and second to last box in Figure D3e,f)

[Figure]

**Figure D5.** Standard deviation of peak area as a function of mode peak area (a,b), histogram of relative standard deviation i.e. the ratio of standard deviation to mode peak area (c,d) where the last bin also contains all values beyond the last bin range, and relative standard deviation as a function of mode peak area (e,f) for the fitted Aitken and accumulation mode with binned box-whiskers plot (25th to 75th PC box with horizontal median line and 10th to 90th PC whiskers where bins refer to quartiles of peak area from lowest Q1 to highest Q4); data for day-to-day size distributions of NO3 at the HKUST site including all seasons.

**References**

[revised manuscript text omitted]

We thank the referee for his/her time to provide us with extensive and valuable input. Please find below our responses to the raised comments, questions and suggestions. In the following, raised **comments / suggestions are in red** and respective **responses in green**, while **alterations to the manuscript text are indicated in blue**.

**General Comment**

This manuscript reports a systematic study on the long term chemically-resolved size distribution data measured by a high-resolution AMS from an urban and a suburban location in Hong Kong. Measured size distributions of individual species were fitted using a bimodal lognormal model and the derived mode sizes and submode concentrations were analyzed for seasonal and diurnal variations. Based on these results, the authors discussed the influences of different sources on aerosol sizes, differences between urban and suburban aerosols, and variations in aerosol mixing states. The work reported in this ms is technically sound and interesting and the synthesis of long-term AMS size distribution data is a novel undertaking. However, the assumption that all aerosol size distributions are bimodal appears to be overly simplified and somewhat arbitrary. Urban particles, in particular, are contributed by various primary and secondary sources and particle from different sources tend to have different size distributions. Although I could see the benefit of simplifying the complexity by using a bimodal assumption, it would be helpful that the authors elaborate a bit more on the justification for this treatment and provide more details on how well the biomodal log-normal model perform in fitting the observation data. Maybe a more systematic evaluation of the quality of fit for the size distribution data is more appropriate than one example (Fig. D1). Also, I would like mention that more sophisticated methods, such as the 3-dimentional factor analysis reported in Ulbrich et al. (2012), maybe useful to explore the number of modes. I also notice that the naming of the size modes in this work is a bit confusing. Aitken mode refers to particles smaller than 100 nm in diameter. However, according to Fig. 1 and 2, the mode diameters for the so-called "Aitken mode" determined through bimodal log-normal fitting are all above 100 nm, some even reaching 200 nm. Additionally, the discussions on diurnal variations of aerosol size mode focus very much on the impacts of emissions sources and physical and chemical processes. However, changes of air masses due to wind shifts or upwind impacts could also be important and should be evaluated.

We will present a brief review of AMS-related size distribution work as well as more details for the considerations and procedures involved in the peak fitting in the revised manuscript and Supporting Material. While we agree that especially in more complex urban environments, multimodality in size distributions is likely we can only clearly distinguish two modes in the mass size distributions from our field measurements. The impacts from coarser particles ($>PM_1$) are usually visible as enhanced tails at the upper boundary of the mass size distributions, however, the influence of decreasing lens transmission and possible impacts of longer evaporation times of larger particles (Canagaratna et al., 2004) render further examination unfeasible. Otherwise, there were no clear indications of possible further submodes (e.g. shoulders, peaks, etc.) and fitting of additional modes did not appear warranted, leading to our approach of a bimodal deconvolution.

The naming conventions have been chosen as to capture the overall character of the modes. Bearing in mind that presented particle diameters are in terms of vacuum-aerodynamic diameter, which is related to the more commonly used mobility diameter by approximately a factor of particle density, the resolved mode diameters correspond to mobility diameters around 100-150nm, thus close to the Aitken mode.

We recognize that our Aitken particle mode is in the borderline region between Aitken and accumulation mode and we will clarify the definition more clearly in the introductory and methodology part in the revised manuscript.

For the day-to-day size distributions, we examined wind frequency data and data from backtrajectory analysis (see Figures D12-13 in the Supporting Material). The diurnal analysis is split by seasons and thus encompasses 1-2 months of measurement data. Surface wind patterns at both measurement sites were rather complex and subject to irregular processes (e.g. street canyon effects, land-sea-breeze, monsoon circulation). It is unlikely that air mass changes or wind shifts occurred at regular diurnal time scales at both measurement sites and are therefore not discussed in the context of the diurnal size distributions.

The smaller mode typically exhibited mode diameters in the range of 100-200 nm ($d_{va}$) and is thus in the transition region between Aitken and lower accumulation mode. For a clearer distinction from the larger mode which unambiguously belonged to the accumulation size range, we opt to refer to the small mode as *Aitken mode* in this work.

Associated further changes:
- Addition of Chapter 3.3 to the manuscript, discussing previous studies.
- Revision and addition to Section B of the Supporting Material.

The relevant changes are appended at the bottom of this document

**Specific Comments**

**Comment**      The numbering of the sections does not seem logic. For example, according to content, 3.2.1 is parallel to 3.2.

**Response**      The numbering of sections in chapter 3 is erroneous. Part 3.2. should in fact be 3.1.1., while 3.2. should be 3.1.2. This also affects section numbers thereafter, and we provide a corrected chapter numbering in the revised manuscript.

**Comment**      Line 11 - 12: this sentence is difficult to understand, consider to revise.

**Response**      We have revised the concerned sentence.

**Alteration**      The size distributions displayed bimodal characteristics and were deconvoluted into submodes which were analyzed for diurnal trends and longer term day-to-day variations.

**Comment**      Line 63, the AMS lens transmission is close to zero for particles smaller than 30-35 nm, so it is not precise to say the Aitken mode particles (10-100 nm) are covered by AMS.

**Response**      We agree that the AMS inlet lens is not capable of capturing particles in the smallest Aitken mode range which is relevant in terms of particle number concentration. Particle volume and thus particle mass is however dominated by large particles in each mode, and thus the AMS would still be able to capture most of the Aitken mode (from a particle mass perspective), which is mentioned in Line 63 ("particle mass").

**Comment**      Line 65 states that the AMS particle size data from ambient measurements are rarely investigated in depth. This is not true. A number of studies, including a few from more than 10 years ago, analyzed the size-resolved composition data from AMS quite extensively and utilized the information to elucidate aerosol sources, new particle formation and growth mechanisms, and other atmospheric processes. Several references (not the complete list) are provided at the end of this comment in the reference section. In addition, Ulbrich et al. (2014) reported a comprehensive study on the size-resolved mass spectral data from an ambient study using 3-D factorization models. Considering that this manuscript focuses on AMS size distribution data, I'd like to recommend that the authors provide a background review on previous works in the introduction. Additionally, I notice that citations are sometime missing when findings from the authors' own research group are mentioned. This could cause confusion when the results from this work alone are sufficient to support the claim. A thorough check for in-text citations is recommended.

**Response**      We seek to clarify the sentence concerned (Line 65). We aim to stress that AMS particle size distribution on finer time scales are in fact rarely investigated. The majority of studies examined overall average size distributions i.e. covering the whole sampling campaign, specific episodic event or limited time periods of interest.

We will rephrase this sentence and incorporate a short section reviewing these studies as recommended by the reviewer.

**Alteration**      Thus far most studies employing ambient size distribution data from aerosol mass spectrometer measurements investigated longer time period averages, i.e. campaign averages (Salcedo et al., 2006;Sun et al., 2009;Aiken et al., 2009;Huang et al., 2010;Takegawa et al., 2009;Saarikoski et al., 2012;Li et al., 2015) or specific time periods of interest (Elser et al., 2016;Lee et al., 2013). Mohr et al. separated organic particle mass size distributions by periods of dominant influence of different PMF-resolved organic aerosol factors to study the properties of mass size distributions in relation to organic aerosol composition (Mohr et al., 2012). The 3D-factorization technique is an extension of traditional AMS PMF analysis on organic aerosol allowing estimates on the size distributions of organic aerosol factors, however under the assumption that factor size distributions remain invariant over the measurement period (Ulbrich et al., 2012).

The temporal evolution of species-specific size distributions, are mostly discussed qualitatively (Drewnick et al., 2005) and only few studies have evaluated temporal trends in mass size distributions in greater detail. Particle nucleation and subsequent growth events were investigated in Pittsburgh using size data from an AMS and two SMPS as well as various gaseous pollutant instruments and meteorological information. The AMS mass size distributions were evaluated quantitatively using the time series of binned particle concentrations generated from the grouping of raw data into wider size bins to represent different stages in the particle growth process. (Zhang et al., 2004). The same method was employed to evaluate contributions of ultrafine mode and accumulation mode particles to total organic particle mass (Zhang et al., 2005) by summation of size bins in the range of 30-100 nm and 100-1000nm. The authors also explored diurnal changes in size distributions of particle species by averaging over 3h periods in the morning (6–9 am) and afternoon (1–4 pm). Sun et al. present a qualitative discussion of diurnal variations in the mass size distributions of the m/z 44, m/z 57 and derived $C_4H_9^+$ ion signals from measurements at an urban site in New York (Sun et al., 2011). Similarly, Setyan et al. examined diurnal changes in the mass size distributions of organics and sulfate qualitatively and used binned concentrations (40–120, 120–200, and 200–800) nm in their quantitative analysis to study the evolution of particle chemistry in new particle formation and growth events (Setyan et al., 2012).

| | |
|---|---|
| **Comment** | Line 91, what's the RH at the exit of the dryer? |
| **Response** | The relative humidity at the diffusion drier outlet was consistently in a range of 20-30% and tested periodically in both campaigns. |

| | |
|---|---|
| **Comment** | Line 130 - 132, the sentence "Utilizing ..." is vague, consider to revise |
| **Response** | We have revised the sentence. |
| **Alteration** | AMS mass-based size distributions can be utilized more systematically and complementary to standard AMS data analysis techniques by deconvoluting multimodal distributions into their constituting submodes and evaluating their variation and contribution to overall species concentration variations on a diurnal time scale. |

| | |
|---|---|
| **Comment** | Line 160, "sweep-out" by what, rain? |
| **Response** | "Sweep out" refers to removal by circulation (e.g. surface wind, traffic induced turbulence), which is particularly important for urban street canyons where periodic changes in circulation (heating, traffic patterns) exist. Washout by rain is another possibility, but is more dependent on specific meteorological conditions, i.e. occurs on irregular basis. |

| | |
|---|---|
| **Comment** | Line 161, what's residual traffic? |
| **Response** | We employ the term "residual traffic" in similarity to "background" concentration levels, as traffic in the central inner-city districts remains continuous albeit at much lower vehicle number compared to the daytime, whereas in more remote areas or smaller cities traffic at night is typically intermittent. We will change the wording to avoid confusion. |
| | […] as well as contributions from nighttime activity such as traffic, which remains continuous in the inner-city districts at night albeit at much lower vehicle numbers compared to the daytime. |

| | |
|---|---|
| **Comment** | Line 167 – 169, clarify what the decreases correspond to. |
| **Response** | We have modified the sentence for clarification. |
| **Alteration** | Significant changes were evident in the particle size metric (MMD) during the same time period, where a consistent decrease by 20-30% from about 170 nm (spring) or 160 nm (summer) to 130-140 nm (spring) or 120 nm (summer) was evident with the concurrent increase in road traffic. |

| | |
|---|---|
| **Comment** | Line 170- 173, are there SMPS measurements to support the increase of particle number concentrations? |
| **Response** | The measurements presented in this study were conducted with a limited set of available instruments and did unfortunately not encompass complementary particle size distribution measurements (by either electrostatic classification or MOUDI samplers). |

| | |
|---|---|
| **Comment** | Line 172, where does the cutoff size of 50 nm come from? |
| **Response** | We here refer to the limitations in AMS lens transmission (see references in Lines 123-124) at the lower size end. The current wording may be confusing and we will revise the sentence to clarify. |

**Alteration** […] from elemental carbon particles and smaller Aitken mode and nucleation mode particles below the range of efficient particle transmission of the AMS inlet lens.

**Comment** Line 174 – 190, how important was COA in Aitken mode around noon time? What about contributions from secondary aerosol formation and other primary sources such as HOA? Summertime SOA and SIA formation tend to be higher and can influence particles in all size modes. I am not sure change of cooking behavior was the only reason for the different diurnal shapes between spring and summer.

**Response** As discussed in the captioned section, changes in organic aerosol (as resolved by PMF) during the meal hours were almost exclusively due to changes in COA. Total HOA concentrations beyond the morning rush hour (6:00 – 9:00) remained stable throughout the day and only fell off in the late night hours. Changes in SOA concentrations during the daylight hours were small in both seasons (Lee et al., 2015) and concentration-wise much smaller than daytime COA concentrations. Significant influences from SOA and HOA are thus not likely. Similarly, changes in cooking behavior were not evident, but as mentioned in the captioned section, we consider advection/transport due to differences in wind patterns in spring and summer more likely to be responsible for the observed trend.

**Comment** Line 189 – 190, this sentence is somewhat confusing

**Response** We have revised this sentence for clarification.

**Alteration** Cooking emissions did not lead to conspicuous changes in the size-related distribution metrics, i.e. there were no obvious trends in particle diameters (MMDs) or distribution widths (GSDs) during the meal time periods.

**Comment** Line 213-215, this sentence is confusing. Please clarify.

**Response** This sentence is in support of the previous statement and underlines that meteorological parameters differed substantially between the two seasons. We note that the mentioning of the data source (temperature measurements were taken directly at the roadside station – solar irradiation data were only available from the farther away HKUST site) is not directly relevant to the discussion and may confuse the reader at this point. As the origin of the data is clearly mentioned in the caption of the related Figure, we will omit the information in-text in the revised manuscript.

**Alteration** Ambient temperatures and solar irradiation differed substantially with 7oC higher average temperatures and three times higher integrated daily solar irradiation in summer compared to spring (Figure D6e,f in the Supporting Material).

**Comment** Line 215-216, what does "nucleation of gas-phase emissions" mean?

**Response** The wording is erroneous and has been corrected.

**Alteration** Lower overall ambient temperatures enhance condensation of gas-phase emissions and particle nucleation and shift the gas-to-particle partitioning equilibrium of semi-volatile constituents […]

**Comment** Line 222-223, "reduce nucleation: : : of more volatile exhaust component on fresher, smaller particles: : :"? Did nucleation ever occur with the volatile component in the atmosphere?

**Response** "Reduce" refers to both particle nucleation and, separately, the condensation of volatile exhaust components. We revise this sentence for clarification.
The section in question discusses traffic-exhaust related components from the measurements at the urban roadside location. The site was impacted strongly by vehicle emissions and thus was affected by both nucleation and gas-to-particle conversion from components originating from vehicle exhaust. Both processes are inversely related to temperature. Vehicle exhaust upon discharge from the tailpipe into the ambient atmosphere is rapidly diluted and cooled leading to both gas-to-particle conversion of (semi-)volatile species, homogenous and heterogeneous nucleation and condensation or adsorption on pre-existing particles (Kittelson et al., 2006a;Kittelson et al., 2006b) in the immediate vicinity of the exhaust pipe. The extent of these processes would depend on various parameters including engine type, engine load, species distribution, as well as ambient conditions, e.g. temperature as discussed here.

**Alteration**  […] consistent with the expected stronger impact of reduced particle nucleation and reduced condensation of semi-volatile exhaust components on fresher, smaller particles in the warmer season.

**Comment**  Line 244-247, this discussion seems somewhat speculative. Are there data to support the nighttime heterogeneous oxidation of SO2 by O3 in Hongkong during spring time? Has this issue been investigated in previous publication(s)? Did wind direction or air mass origin play a role in the observed size mode change?

**Response**  Neither wind direction nor air mass origin had any diurnal features that would explain this regularly recurring (i.e. diurnal) observation.
It must be noted that this effect is likely due to the roadside character of the measurement site, leading to ozone peaking during the nighttime (as noted in the manuscript due to NOx titration during the day), and was thus not evident at the suburban measurement site. We are not aware of any detailed investigations into the diurnal concentration characteristics of inorganic components at roadside measurement sites in Hong Kong. The limited amount of additional measurement data beyond gas-phase standard criteria pollutant data unfortunately prevents a deeper analysis in this work.

**Comment**  Line 437, "particles containing different species were similar in size" is confusing.

**Response**  We have revised the sentence for clarification.

**Alteration**  Close nominal agreement (i.e. diameter ratios close to 1) infer that different species were distributed similarly across the particle size range which thus most likely represents a largely internally mixed particle population, while the spread of data (correlation coefficient) indicates the temporal homogeneity or divergence of resolved mode diameters.

**Comment**  Line 449, small particles are not just processed by condensational growth and coagulation. In the presence of high humidity, they can also go through aqueous-phase processing.

**Response**  We have revised the statement to include the possibility of aqueous-phase processing.

**Alteration**  External mixing is more prevalent for freshly formed smaller particles which have typically undergone less atmospheric processing, such as condensational growth, coagulation or aqueous-phase reactions.

**Comment**  Fig. D6, can wind data be provided as well?

**Response**  Wind patterns at both measurement sites are strongly influenced by irregular processes (street canyon effects, land-sea-breeze, monsoon winds) which typically change from day-to-day. Including diurnal wind data in Figure D6 may thus not be meaningful.

**Changes in sections of main manuscript and Supporting Material**
**Main manuscript**

**3.3 Comparison to previous studies**

Particle size distribution studies in Hong Kong are generally scarce and have focused on either size segregated filter samples (MOUDI) for general ambient measurements or electrostatic classification in particle formation and particle growth studies (Guo et al., 2012;Cheung et al., 2015) . The latter studies focus on specific and narrow time periods and lack general discussions on ambient particle size distributions.

Two ambient studies were undertaken at the suburban coastal HKUST site using size-segregated samples from a ten-stage MOUDI sampler and offline chromatographic analysis. Inorganic constituents (NH4, NO3, SO4) in fine particles (i.e. $D_p < 1.8$ µm) were shown to follow bimodal distributions with mode diameters in the range of 0.14-0.21 µm and 0.46-0.58 µm in samples collected in the winter season, while the main mode was observed in the coarse region (4-6 µm) for all three species (Zhuang et al., 1999). A subsequent year-long observational study also reported bimodal fine particle distributions with mode diameters of 0.1-0.3 µm and 0.7–0.9 µm and 1-2 additional modes in the coarse region (Bian et al., 2014), however, the main mode in the size distributions of sulfate, ammonium, potassium and oxalate was observed in the droplet mode (0.7 – 0.9 µm) in this study. Vehicle exhaust plumes sampled on-road from a Mobile Real-time Air Monitoring Platform (MAP) across Hong Kong's road network exhibited three distinct particle volume size distributions: a unimodal distribution with an accumulation mode at 0.2 µm and two bimodal distributions with a minor mode at 0.2 µm and the dominant mode at 0.5 or 0.7 µm (Yao et al., 2007).

The bimodality in the fine particle range across these studies is consistent with the AMS-based results in this work. Nominally, the accumulation mode diameters from filter based studies and the chase studies are larger than those from AMS measurements where maximum mode diameters occurred at $d_{va}$ ~ 700nm, corresponding to $d_a$ ~ 470 (assuming $d_{va}$ ~ $d_a$ * density; particle density ~ 1.5 g/cm$^3$). Direct comparability is however limited due to fundamental differences in sizing techniques (MOUDI: atmospheric pressure; AMS: near-vacuum), sampling times (MOUDI: 24h samples, scattered time line; AMS: minute raw resolution averaged to hourly or daily, continuous time line), measurement uncertainties (MOUDI: sampling artifacts such as vapor adsorption and desorption; AMS: inlet lens transmission) and aerosol pretreatment (none for MOUDI with potential impacts on particle size in high humidity (>80%) conditions (Fang et al., 1991); AMS: removal of water prior to introduction to instrument).

**Supporting Material**

Lognormal peaks were fitted to each 24h and hour-of-day AMS mass size distribution respectively employing the *Multipeak Fit V2* algorithm in *Igor Pro (Wavemetrics)* using a simple vertical offset as the baseline and initial guesses on peak position, height, and width based on visual inspection of the raw size distribution. The multipeak fitting tool employs the Levenberg-Marquardt algorithm (Gill et al., 1981) as a non-linear least squares fit and iteratively adjusts the initial fit parameter guesses until a convergent solution with minimized residuals is achieved. In sporadic cases, the fitted solution led to excessive deviations from the initial guesses with greatly shifted peak locations and large fluctuations in peak width. In such cases, results from the peak fits of immediately adjacent size distributions (i.e. previous and next distributions in the sequence) were evaluated and used to adjust the fitting process by fixing either the location (*primary*) or the width of the peak (*secondary*) to the average value of the two adjacent fitted distributions.

For the diurnal size distributions, measurement data from time periods with large differences in species concentration levels were pooled together. The averaging of mass (or volume) based size distribution involves different uncertainties for each size bin due to the cubic relationship between particle mass (or volume) and particle diameter and the corresponding improvement in signal-to-noise ratio with increasing particle size. To establish reliable diurnal trends we adopt an approach similar to the analysis of conventional species concentration diurnal trends by evaluating size distributions reconstructed from the average, median, 25th and 75th percentile of each size bin. Similar diurnal trends in the fitting parameters across these different size distributions would confirm that changes were indeed recurrent daily while divergent trends would indicate that irregular processes (e.g. episodic events) were more significant in determining size distribution characteristics. Since episodic pollution events and clean periods (e.g. prolonged precipitation) were not removed from the dataset, the quantitative analysis focuses on trends observed in the median dataset to minimize skewing effects of high and low concentration periods.

Uncertainties can arise from the peak fitting process itself. While the bimodality of the size distributions was obvious in most cases (i.e. a main mode with a shoulder towards smaller particle sizes, e.g. Figure D1), accumulation mode particle mass can occasionally dominate the mass size distribution and diminish the Aitken mode. To achieve confidence in the appropriateness of the bimodal fitting we evaluated both unimodal and bimodal peak fits whenever the Aitken to accumulation mode peak ratio was <10% and we depict a representative example below (Figure D2a. b). The distribution of the fit residuals (Figure D2c, d)  was examined and cumulative probability distributions of the fit residuals compared by the Kolmogorov-Smirnov test (Figure D2e) to assess whether fit residuals were significantly different at 95% confidence level (CL). It is evident that the bimodal fit performs better at resolving the raw size distribution in the smaller size region and overall yields a more normal residual distribution. The Kolmogorov-Smirnov test confirms that the residual distributions are statistically different ($D>D_{critical}$ at 95% CL). We tested all borderline cases using the outlined procedure. In this study, bimodal fits yielded unanimously better results in all cases for both diurnal and day-to-day size distributions and all investigated species, i.e. the Aitken mode always remained clearly distinguishable from the accumulation mode.

While the peak fitting algorithm yields a unique individual solution with a set of parameters for which resulting residuals (*difference of fitted and original distribution*) are minimized, the surrounding solution space provides a potentially infinite number of similar solutions with slightly larger residuals. The standard deviations of the fit parameters can provide an estimate of the variability of the peak parameters between the final fit solution and the surrounding solution space. We evaluated the uncertainty in peak area (i.e. integrated mode particle mass) which represents the combined uncertainty of the peak position, width and height (which altogether directly determine the peak area) for all fitted size distributions in this work.

Figure D3 depicts the standard deviation of resolved peak area (i.e. integrated mode particle mass concentration) nominally and relative to the peak area for the diurnal size distributions of NO3 at the urban Mong Kok site in summer 2013 and Tables C1-C2 summarize the values of percent standard deviations for all species at both measurement sites respectively. The median datasets, which were used for quantitative discussion for the diurnal size distribution analysis, exhibited particle mass uncertainties of 14-48% in the Aitken mode and 1-12% in the accumulation mode at the suburban HKUST site, and 7-44% in the Aitken mode and 1-6% in the accumulation mode at the urban MK site. Figure D4 depicts the 75[th] percentile-bin diurnal variation of NO3 (which displayed the largest uncertainties in Figure D3) with the corresponding peak area variability, and shows that the interpretation of the diurnal variation would remain largely unaffected from the incurred uncertainties.

For the day-to-day 24h size distributions a corresponding analysis was undertaken, with Figure D5 depicting the size distributions of NO3 at the HKUST site for all covered seasons exemplarily, and Table C3 summarizes the values of percent standard deviations for all species at both measurement sites respectively. Peak fit uncertainties typically increase with decreasing integrated peak area and can exceed the values of the peak area in the Aitken mode in a small number of cases (e.g. Figure D5c,e – ratios >1). Quantification of the Aitken mode may not be possible at high levels of confidence in these isolated cases. They were retained in the dataset due to their low frequency of occurrence and to enable a complete discussion over the full concentration range without biasing towards larger concentration (i.e. fitted peak areas) values.

**Table C1.** Percentiles of relative standard deviation (*rows; corresponding to the box-whiskers plot in Figure D3e,f*) in percent from lognormal peak fits (bimodal deconvolution) for the resolved Aitken mode (a) and accumulation mode (b) particle concentration for diurnal size distributions at the HKUST supersite (2011/12), *columns* describe the data set, i.e. reconstructed size distributions from the 25th percentile, median, 75th percentile and mean of the size bins

*Sp=Spring, Su=Summer, Fa=Fall, Wi=Winter*

(a)

| Aitken mode % SD | | 25th PC Distr. | | | | Median Distr. | | | | 75th PC Distr. | | | | Mean Distr. | | | | Range |
|---|---|---|---|---|---|---|---|---|---|---|---|---|---|---|---|---|---|---|
| | | Sp | Su | Fa | Wi | Sp | Su | Fa | Wi | Sp | Su | Fa | Wi | Sp | Su | Fa | Wi | |
| **NO3** (UST) | PC-90 | 67 | 76 | 39 | 85 | 76 | 44 | 54 | 55 | 80 | 75 | 56 | 39 | 36 | 77 | 44 | 42 | 36-85 |
| | PC-75 | 52 | 42 | 28 | 57 | 66 | 32 | 39 | 40 | 61 | 59 | 44 | 29 | 22 | 38 | 35 | 34 | 22-66 |
| | **PC-50** | **36** | **33** | **22** | **46** | **44** | **26** | **31** | **22** | **40** | **43** | **34** | **23** | **18** | **28** | **24** | **30** | **18-46** |
| | PC-25 | 29 | 25 | 16 | 28 | 26 | 14 | 24 | 18 | 25 | 29 | 22 | 17 | 14 | 19 | 20 | 25 | 14-29 |
| | PC-10 | 21 | 20 | 15 | 24 | 19 | 12 | 19 | 15 | 17 | 19 | 17 | 10 | 13 | 13 | 15 | 21 | 10-24 |
| **SO4** (UST) | PC-90 | 38 | 38 | 42 | 74 | 19 | 36 | 39 | 43 | 22 | 40 | 40 | 42 | 19 | 36 | 37 | 81 | 19-81 |
| | PC-75 | 35 | 33 | 38 | 55 | 18 | 28 | 33 | 32 | 18 | 30 | 35 | 64 | 17 | 30 | 30 | 66 | 17-66 |
| | **PC-50** | **28** | **30** | **33** | **32** | **16** | **26** | **27** | **25** | **14** | **24** | **27** | **27** | **14** | **24** | **26** | **48** | **14-48** |
| | PC-25 | 21 | 25 | 26 | 24 | 13 | 21 | 22 | 21 | 11 | 18 | 24 | 22 | 12 | 21 | 23 | 40 | 11-40 |
| | PC-10 | 17 | 23 | 24 | 19 | 11 | 20 | 20 | 13 | 10 | 14 | 20 | 19 | 10 | 19 | 22 | 35 | 10-35 |
| **Org** (UST) | PC-90 | 52 | 23 | 41 | 44 | 42 | 28 | 32 | 27 | 47 | 48 | 45 | 53 | 46 | 29 | 24 | 32 | 23-53 |
| | PC-75 | 41 | 18 | 26 | 28 | 30 | 22 | 27 | 22 | 26 | 39 | 35 | 44 | 36 | 23 | 21 | 25 | 18-44 |
| | **PC-50** | **26** | **14** | **17** | **21** | **19** | **18** | **21** | **17** | **20** | **32** | **26** | **35** | **23** | **18** | **19** | **21** | **14-35** |
| | PC-25 | 16 | 11 | 13 | 18 | 17 | 16 | 19 | 15 | 18 | 28 | 20 | 29 | 18 | 14 | 16 | 17 | 11-29 |
| | PC-10 | 9 | 9 | 10 | 11 | 14 | 9 | 15 | 12 | 15 | 20 | 17 | 21 | 17 | 11 | 14 | 16 | 9-21 |

(b)

| Accum. mode % SD | | 25th PC Distr. | | | | Median Distr. | | | | 75th PC Distr. | | | | Mean Distr. | | | | Range |
|---|---|---|---|---|---|---|---|---|---|---|---|---|---|---|---|---|---|---|
| | | Sp | Su | Fa | Wi | Sp | Su | Fa | Wi | Sp | Su | Fa | Wi | Sp | Su | Fa | Wi | |
| **NO3** (UST) | PC-90 | 8 | 5 | 8 | 4 | 7 | 5 | 6 | 3 | 4 | 9 | 6 | 3 | 3 | 4 | 5 | 1 | 1-9 |
| | PC-75 | 6 | 4 | 7 | 3 | 3 | 4 | 5 | 2 | 4 | 7 | 5 | 2 | 2 | 3 | 5 | 1 | 1-7 |
| | **PC-50** | **4** | **3** | **5** | **2** | **3** | **3** | **5** | **2** | **3** | **4** | **5** | **2** | **2** | **3** | **4** | **1** | **1-5** |
| | PC-25 | 3 | 3 | 4 | 2 | 2 | 3 | 4 | 1 | 3 | 3 | 4 | 2 | 2 | 2 | 4 | 1 | 1-4 |
| | PC-10 | 2 | 2 | 4 | 1 | 2 | 2 | 4 | 1 | 2 | 2 | 3 | 1 | 2 | 2 | 3 | 1 | 1-4 |
| **SO4** (UST) | PC-90 | 2 | 3 | 3 | 5 | 2 | 2 | 2 | 3 | 3 | 2 | 3 | 4 | 2 | 2 | 2 | 2 | 2-5 |
| | PC-75 | 2 | 2 | 2 | 2 | 2 | 2 | 2 | 2 | 2 | 2 | 2 | 3 | 1 | 2 | 2 | 2 | 1-3 |
| | **PC-50** | **2** | **2** | **2** | **1** | **2** | **2** | **2** | **2** | **2** | **2** | **2** | **2** | **1** | **2** | **2** | **1** | **1-2** |
| | PC-25 | 2 | 2 | 2 | 1 | 1 | 2 | 2 | 1 | 1 | 2 | 2 | 2 | 1 | 2 | 2 | 1 | 1-2 |
| | PC-10 | 1 | 2 | 1 | 1 | 1 | 2 | 2 | 1 | 1 | 1 | 2 | 1 | 1 | 2 | 2 | 1 | 1-2 |
| **Org** (UST) | PC-90 | 29 | 12 | 16 | 9 | 18 | 8 | 9 | 5 | 10 | 6 | 7 | 3 | 18 | 7 | 5 | 5 | 3-29 |
| | PC-75 | 18 | 9 | 11 | 7 | 10 | 5 | 5 | 3 | 6 | 4 | 5 | 2 | 11 | 5 | 4 | 3 | 2-18 |
| | **PC-50** | **12** | **7** | **5** | **5** | **6** | **4** | **4** | **3** | **4** | **3** | **4** | **2** | **7** | **3** | **3** | **3** | **2-12** |
| | PC-25 | 7 | 4 | 3 | 3 | 5 | 3 | 3 | 2 | 3 | 2 | 3 | 2 | 5 | 2 | 2 | 2 | 2-7 |
| | PC-10 | 4 | 4 | 1 | 2 | 4 | 2 | 2 | 1 | 2 | 2 | 2 | 1 | 2 | 1 | 2 | 2 | 1-4 |

**Table C2.** Percentiles of relative standard deviation (*rows; corresponding to the box-whiskers plot in Figure D3e,f*) in percent from lognormal peak fits (bimodal deconvolution) for the resolved Aitken mode (a) and accumulation mode (b) particle concentration for diurnal size distributions at the urban MK site (2013), *columns* describe the data set, i.e. reconstructed size distributions from the 25th percentile, median, 75th percentile and mean of the size bins *Sp=Spring, Su=Summer*

(a)

| Aitken mode
% SD | | 25th PC Distr. | | Median Distr. | | 75th PC Distr. | | Mean Distr. | | Range |
|---|---|---|---|---|---|---|---|---|---|---|
| | | Sp | Su | Sp | Su | Sp | Su | Sp | Su | |
| NO3
(MK) | PC-90 | 38 | 46 | 40 | 34 | 40 | 41 | 22 | 26 | 22-46 |
| | PC-75 | 27 | 34 | 26 | 28 | 34 | 35 | 20 | 16 | 16-35 |
| | **PC-50** | **15** | **24** | **23** | **21** | **25** | **28** | **18** | **14** | **14-28** |
| | PC-25 | 9 | 20 | 19 | 17 | 18 | 22 | 15 | 12 | 9-22 |
| | PC-10 | 6 | 19 | 16 | 14 | 15 | 20 | 13 | 11 | 6-20 |
| SO4
(MK) | PC-90 | 63 | 46 | 35 | 38 | 24 | 31 | 23 | 21 | 21-63 |
| | PC-75 | 50 | 36 | 30 | 36 | 21 | 28 | 21 | 20 | 20-50 |
| | **PC-50** | **44** | **33** | **28** | **24** | **20** | **23** | **19** | **18** | **18-44** |
| | PC-25 | 37 | 27 | 24 | 21 | 17 | 20 | 15 | 17 | 15-37 |
| | PC-10 | 33 | 25 | 21 | 19 | 15 | 18 | 15 | 16 | 15-33 |
| Org
(MK) | PC-90 | 22 | 12 | 22 | 19 | 30 | 21 | 15 | 14 | 12-30 |
| | PC-75 | 16 | 10 | 12 | 12 | 18 | 11 | 12 | 8 | 8-18 |
| | **PC-50** | **10** | **8** | **10** | **9** | **10** | **9** | **8** | **7** | **7-10** |
| | PC-25 | 8 | 7 | 8 | 7 | 7 | 8 | 6 | 6 | 6-8 |
| | PC-10 | 7 | 6 | 7 | 6 | 6 | 6 | 5 | 5 | 5-7 |

(b)

| Accum. mode
% SD | | 25th PC Distr. | | Median Distr. | | 75th PC Distr. | | Mean Distr. | | Range |
|---|---|---|---|---|---|---|---|---|---|---|
| | | Sp | Su | Sp | Su | Sp | Su | Sp | Su | |
| NO3
(MK) | PC-90 | 9 | 9 | 6 | 7 | 3 | 5 | 2 | 5 | 2-9 |
| | PC-75 | 6 | 8 | 4 | 5 | 3 | 4 | 2 | 4 | 2-8 |
| | **PC-50** | **4** | **6** | **3** | **4** | **2** | **3** | **2** | **3** | **2-6** |
| | PC-25 | 2 | 5 | 2 | 4 | 2 | 3 | 1 | 3 | 1-5 |
| | PC-10 | 2 | 4 | 1 | 3 | 1 | 3 | 1 | 2 | 1-4 |
| SO4
(MK) | PC-90 | 4 | 3 | 2 | 3 | 2 | 5 | 2 | 3 | 2-5 |
| | PC-75 | 3 | 2 | 2 | 3 | 2 | 4 | 2 | 2 | 2-4 |
| | **PC-50** | **3** | **2** | **2** | **2** | **1** | **4** | **2** | **2** | **1-4** |
| | PC-25 | 2 | 2 | 2 | 2 | 1 | 3 | 1 | 2 | 1-3 |
| | PC-10 | 2 | 2 | 1 | 2 | 1 | 3 | 1 | 2 | 1-3 |
| Org
(MK) | PC-90 | 9 | 8 | 8 | 7 | 9 | 5 | 6 | 6 | 5-9 |
| | PC-75 | 6 | 7 | 5 | 5 | 5 | 4 | 5 | 4 | 4-7 |
| | **PC-50** | **5** | **5** | **4** | **4** | **3** | **4** | **3** | **4** | **3-5** |
| | PC-25 | 4 | 4 | 3 | 3 | 3 | 3 | 3 | 3 | 3-4 |
| | PC-10 | 2 | 3 | 3 | 3 | 5 | 2 | 2 | 2 | 2-3 |

**Table C3.** Percentiles of percent standard deviation (*rows; corresponding to the box-whiskers plot in Figure D5e,f*) from lognormal peak fits (bimodal deconvolution) for the resolved Aitken mode and accumulation mode for 24h day-to-day size distributions at the suburban HKUST site (a) and the urban MK site (b) for all investigated species, *columns* describe the uncertainties in terms of quartiles of resolved peak area, where Q1 refers to the lowest 25% and Q4 the highest 25% of resolved peak area (see also Figure D4)

(a)

| HKUST '11/12 % SD | | NO3 | | | | | SO4 | | | | | Org | | | | |
|---|---|---|---|---|---|---|---|---|---|---|---|---|---|---|---|---|
| | | Q1 | Q2 | Q3 | Q4 | Range | Q1 | Q2 | Q3 | Q4 | Range | Q1 | Q2 | Q3 | Q4 | Range |
| **Aitken mode** | PC-90 | 95 | 64 | 50 | 27 | 27-95 | 97 | 78 | 49 | 30 | 30-97 | 62 | 37 | 28 | 26 | 26-62 |
| | PC-75 | 58 | 47 | 37 | 24 | 24-58 | 91 | 57 | 36 | 20 | 20-91 | 41 | 24 | 21 | 22 | 21-41 |
| | **PC-50** | **47** | **35** | **25** | **17** | **17-47** | **60** | **39** | **29** | **16** | **16-60** | **30** | **17** | **14** | **17** | **14-30** |
| | PC-25 | 30 | 25 | 20 | 13 | 13-30 | 32 | 26 | 21 | 13 | 13-32 | 25 | 14 | 10 | 12 | 10-25 |
| | PC-10 | 20 | 16 | 15 | 8 | 8-20 | 24 | 18 | 13 | 11 | 11-24 | 16 | 11 | 6 | 7 | 6-16 |
| **Accum. mode** | PC-90 | 31 | 9 | 7 | 4 | 4-31 | 4 | 3 | 3 | 3 | 3-4 | 18 | 11 | 9 | 7 | 7-18 |
| | PC-75 | 13 | 6 | 5 | 3 | 3-13 | 3 | 3 | 3 | 2 | 2-3 | 9 | 7 | 6 | 5 | 5-9 |
| | **PC-50** | **8** | **4** | **3** | **2** | **2-8** | **2** | **2** | **2** | **2** | **~2** | **6** | **4** | **5** | **3** | **3-6** |
| | PC-25 | 3 | 3 | 2 | 2 | 2-3 | 2 | 2 | 2 | 2 | ~2 | 4 | 3 | 3 | 2 | 2-4 |
| | PC-10 | 1 | 2 | 2 | 1 | 1-2 | 2 | 2 | 2 | 1 | 1-2 | 3 | 2 | 2 | 2 | 2-3 |

(b)

| MK '13 % SD | | NO3 | | | | | SO4 | | | | | Org | | | | |
|---|---|---|---|---|---|---|---|---|---|---|---|---|---|---|---|---|
| | | Q1 | Q2 | Q3 | Q4 | Range | Q1 | Q2 | Q3 | Q4 | Range | Q1 | Q2 | Q3 | Q4 | Range |
| **Aitken mode** | PC-90 | 62 | 52 | 47 | 30 | 30-62 | 94 | 50 | 29 | 24 | 24-94 | 23 | 17 | 13 | 16 | 13-23 |
| | PC-75 | 41 | 42 | 30 | 25 | 25-42 | 47 | 35 | 22 | 16 | 16-47 | 16 | 14 | 11 | 12 | 11-16 |
| | **PC-50** | **28** | **34** | **21** | **18** | **18-34** | **33** | **24** | **17** | **12** | **12-33** | **11** | **10** | **6** | **8** | **6-11** |
| | PC-25 | 21 | 22 | 19 | 12 | 12-22 | 26 | 17 | 13 | 9 | 9-26 | 8 | 7 | 5 | 6 | 5-8 |
| | PC-10 | 6 | 10 | 13 | 8 | 6-13 | 19 | 14 | 10 | 5 | 5-19 | 6 | 5 | 3 | 3 | 3-6 |
| **Accum. mode** | PC-90 | 22 | 21 | 6 | 6 | 6-22 | 7 | 6 | 6 | 4 | 4-7 | 13 | 8 | 9 | 8 | 8-13 |
| | PC-75 | 17 | 10 | 5 | 3 | 3-17 | 4 | 4 | 4 | 2 | 2-4 | 8 | 7 | 6 | 4 | 4-8 |
| | **PC-50** | **10** | **6** | **4** | **2** | **2-10** | **3** | **2** | **2** | **1** | **1-3** | **6** | **5** | **4** | **2** | **2-6** |
| | PC-25 | 6 | 3 | 2 | 2 | 2-6 | 2 | 2 | 1 | 1 | 1-2 | 4 | 4 | 3 | 2 | 2-4 |
| | PC-10 | 3 | 1 | 2 | 1 | 1-3 | 1 | 1 | 1 | 1 | ~1 | 2 | 3 | 2 | 1 | 1-3 |

[Figure]

**Figure D1.** Example of a log-normal peak fit *(Multipeak Fit V2, Igor Pro, Wavemetrics, Levenberg-Marquardt algorithm)* of an AMS organics size distribution

[Figure]

**Figure D2.** 24h average size distribution of sulfate (12/12/2011, suburban HKUST site) with (a) unimodal and (b) bimodal logN peak fitting applied; histograms of residuals from the unimodal (c) and bimodal (d) distributions with Gaussian fit (green); and cumulative probability density functions of uni- and bimodal fit residuals (e) with Kolmogorov-Smirnov D metric values at 95% confidence level

[Figure]

**Figure D3.** Standard deviation of peak area as a function of mode peak area (a,b), histogram of relative standard deviation i.e. the ratio of standard deviation to mode peak area (c,d) where the last bin also contains all values beyond the last bin range, and relative standard deviation as a function of mode peak area (e,f) for the fitted Aitken and accumulation mode with binned box-whiskers plot (25[th] to 75[th] PC box with horizontal median line and 10[th] to 90[th] PC whiskers where bins refer to quartiles of peak area from lowest Q1 to highest Q4); data for diurnal size distributions of NO3 at the urban Mong Kok site in summer 2013.

[Figure]

**Figure D4.** Plot of 75th percentile-bin diurnal variation with peak area fit variability and relative standard deviation (corresponding to green data and second to last box in Figure D3e,f)

[Figure]

**Figure D5.** Standard deviation of peak area as a function of mode peak area (a,b), histogram of relative standard deviation i.e. the ratio of standard deviation to mode peak area (c,d) where the last bin also contains all values beyond the last bin range, and relative standard deviation as a function of mode peak area (e,f) for the fitted Aitken and accumulation mode with binned box-whiskers plot ($25^{th}$ to $75^{th}$ PC box with horizontal median line and $10^{th}$ to $90^{th}$ PC whiskers where bins refer to quartiles of peak area from lowest Q1 to highest Q4); data for day-to-day size distributions of NO3 at the HKUST site including all seasons.

[revised manuscript text omitted]

The acceleration of a particle into the vacuum interior of the AMS is a function of its size (Jimenez et al., 2003) and thus particles of different size in the pulse ensemble travel at different velocities which are determined from particle flight times over a fixed flight path (i.e. the length of the chamber). Ions arriving at the detector are counted as a function of time between two subsequent particle packages passing through the particle chopper slit. Averaging over several chopper cycles yields a distribution of ions with respect to particle size and enables the measurement of mass concentrations of specific ions as a function of particle size, or of specific bulk species (Organics, SO4, NO3, NH4, Chl) by employing the fragmentation table as in the unit mass resolution acquisition mode (*V-mode*).

Primary logged data in PToF mode is the particle flight time from which velocity is directly inferred due to the fixed flight path length. Velocity can be related to particle diameter by calibration with a set of particles of known size. AMS sizing takes place in near vacuum conditions with particles subject to the free molecular flow regime rather than the transition regime, and consequently AMS size distributions are reported in terms of vacuum-aerodynamic diameter. The relationship between electrical mobility ($D_m$) and vacuum-aerodynamic diameter ($D_{va}$) is given by (Jayne et al., 2000):

$$D_{va} = D_m \times \delta_p \times S \qquad\qquad \text{Eq. A1}$$

where $\delta_p$ is the particle density and S a particle shape factor for non-spherical particles or those with internal voids. S has been experimentally determined for nitrate (S=0.8). For most other particles, where information on the shape factor are unknown, particles are assumed spherical (S=1). DeCarlo et al. provide a more fundamental discussion of the relationships of different particle diameters and their relation to particle density (DeCarlo et al., 2004;Slowik et al., 2004). Particle size calibrations for the AMS instrument in this study were carried out pre- and post-campaign with sets of monodisperse polystyrene latex particles (PSL, Duke Scientific, CA) in the range of 80nm to 800nm (at least 8 individual particle sizes per calibration). To compensate for the slow evaporation of PSL, the vaporizer temperature was temporarily increased to 800°C from its default value of 600°C for the duration of the calibration. As PSL ion flight traces (m/z 104) are typically broad, the particle flight time was chosen as the leading edge rather than apex of the signal trace, with an addition of half a chopper width time (*0.5*duty cycle/chopper frequency*) to account for chopper broadening. $D_{va}$ in nm and particle velocity v in m/s calculated from the fixed chamber length and measured flight times are related empirically by the following equation (Jayne et al., 2000):

$$v = v_l + \frac{v_g - v_l}{1 + \left(\frac{D_{va}}{D^*}\right)^b}$$

Eq. A2

where $v_l$ is the gas velocity inside the aerodynamic lens, $v_g$ the velocity of the gas as it leaves the lens, $D_{va}$ the vacuum aerodynamic diameter calculated from the PSL particle diameter and the PSL density of 1.05 g/cm$^3$ and a shape factor of 1. D* and b are empirical parameters without concrete physical meaning. As per calibration only particle velocity v and vacuum aerodynamic diameter $D_{va}$ are known. The remaining parameters are determined by a non-linear curve fit of v against $D_{va}$ using the relationship describe by Equation A2.

**B. Treatment of PToF data**

Standard AMS data treatment procedures were employed for the integration of the ion signals in SQUIRREL for unit-mass resolution data (*V-mode and PToF mode*) and PIKA for high resolution data (*W-mode*).

While in V- and W-mode, ions signals from particle constituents are inferred from the difference of blocked (gas-phase ion signals) and unblocked (sum of gas-phase and particle-phase ion signals) particle beam mass spectra, the baseline in the PToF mode for each m/z in the acquisition range is established by averaging of the ion signal in two defined time regions (DC markers) at the very beginning and very end of the chopper cycle. These correspond to velocities of particles beyond the upper and lower transmission capability of the instrument and can therefore represent background ion contributions (Allan et al., 2003).

In m/z channels where gas-phase species may interfere with the first DC marker region (e.g. m/z 15 or m/z 44), only the second DC marker is used for the PToF baseline. In this work, DC markers for all m/z ratios in the range of 12 – 150 were checked individually to determine the most appropriate selection of DC markers for each m/z channel. Mass concentrations per PToF mode run are calculated by application of the quantification principles of the unit-mass resolution mode (*V-mode*) with additional corrections for the chopper frequency and slit width to account for the lower number of particles passing into the flight chamber. As the chopping of the particle beam lowers overall sensitivity, the integrated particle mass concentration for each PToF run was normalized to the V-mode concentration of the same time step. To remove any remaining background biases, ambient size distributions in this study were background-corrected by subtraction of the size distributions acquired during particle-free time periods (ambient air sampled through HEPA-filter bypass).

For the evaluation of longer term trends in size distributions, the original AMS mass size distributions at 10min resolution were averaged to 24h distributions (from 00:00 to 23:59 each day) to evaluate progressive change in particle size distributions ("day-to-day") beyond diurnal influences. Regular reoccurring trends, i.e. diurnal variations, were in turn examined by grouping size distributions by their hour time stamp and reconstructing representative size distributions from the average, median, $25^{th}$ and $75^{th}$ percentile of each size bin for each hour of the day.

Lognormal peaks were fitted to each 24h and hour-of-day AMS mass size distribution respectively employing the *Multipeak Fit V2* algorithm in *Igor Pro (Wavemetrics)* using a simple vertical offset as the baseline and initial guesses on peak position, height, and width based on visual inspection of the raw size distribution. The multipeak fitting tool employs the Levenberg-Marquardt algorithm (Gill et al., 1981) as a non-linear least squares fit and iteratively adjusts the initial fit parameter guesses until a convergent solution with minimized residuals is achieved. In sporadic cases, the fitted solution led to excessive deviations from the initial guesses with greatly shifted peak locations and large fluctuations in peak width. In such cases, results from the peak fits of immediately adjacent size distributions (i.e. previous and next distributions in the sequence) were evaluated and used to adjust the fitting process by fixing either the location (*primary*) or the width of the peak (*secondary*) to the average value of the two adjacent fitted distributions. For the diurnal size distributions, measurement data from time periods with large differences in species concentration levels were pooled together. The averaging of mass (or volume) based size distribution involves different uncertainties for each size bin due to the cubic relationship between particle mass (or volume) and particle diameter and the corresponding improvement in signal-to-noise ratio with increasing particle size. To establish reliable diurnal trends we adopt an approach similar to the analysis of conventional species concentration diurnal trends by evaluating size distributions reconstructed from the average, median, $25^{th}$ and $75^{th}$ percentile of each size bin. Similar diurnal trends in the fitting parameters across these different size distributions would confirm that changes were indeed recurrent daily while divergent trends would indicate that irregular processes (e.g. episodic events) were more significant in determining size distribution characteristics. Since episodic pollution events and clean periods (e.g. prolonged precipitation) were not removed from the dataset, the quantitative analysis focuses on trends observed in the median dataset to minimize skewing effects of high and low concentration periods.

Uncertainties can arise from the peak fitting process itself. While the bimodality of the size distributions was obvious in most cases (i.e. a main mode with a shoulder towards smaller particle sizes, e.g. Figure D1), accumulation mode particle mass can occasionally dominate the mass size distribution and diminish the Aitken mode. To achieve confidence in the appropriateness of the bimodal fitting we evaluated both unimodal and bimodal peak fits whenever the Aitken to accumulation mode peak ratio was <10% and we depict a representative example below (Figure D2a.

b). The distribution of the fit residuals (Figure D2c, d) was examined and cumulative probability distributions of the fit residuals compared by the Kolmogorov-Smirnov test (Figure D2e) to assess whether fit residuals were significantly different at 95% confidence level (CL). It is evident that the bimodal fit performs better at resolving the raw size distribution in the smaller size region and overall yields a more normal residual distribution. The Kolmogorov-Smirnov test confirms that the residual distributions are statistically different ($D>D_{critical}$ at 95% CL). We tested all borderline cases using the outlined procedure. In this study, bimodal fits yielded unanimously better results in all cases for both diurnal and day-to-day size distributions and all investigated species, i.e. the Aitken mode always remained clearly distinguishable from the accumulation mode.

While the peak fitting algorithm yields a unique individual solution with a set of parameters for which resulting residuals (*difference of fitted and original distribution*) are minimized, the surrounding solution space provides a potentially infinite number of similar solutions with slightly larger residuals. The standard deviations of the fit parameters can provide an estimate of the variability of the peak parameters between the final fit solution and the surrounding solution space. We evaluated the uncertainty in peak area (i.e. integrated mode particle mass) which represents the combined uncertainty of the peak position, width and height (which altogether directly determine the peak area) for all fitted size distributions in this work.

Figure D3 depicts the standard deviation of resolved peak area (i.e. integrated mode particle mass concentration) nominally and relative to the peak area for the diurnal size distributions of NO3 at the urban Mong Kok site in summer 2013 and Tables C1-C2 summarize the values of percent standard deviations for all species at both measurement sites respectively. The median datasets, which were used for quantitative discussion for the diurnal size distribution analysis, exhibited particle mass uncertainties of 14-48% in the Aitken mode and 1-12% in the accumulation mode at the suburban HKUST site, and 7-44% in the Aitken mode and 1-6% in the accumulation mode at the urban MK site. Figure D4 depicts the 75th percentile-bin diurnal variation of NO3 (which displayed the largest uncertainties in Figure D3) with the corresponding peak area variability, and shows that the interpretation of the diurnal variation would remain largely unaffected from the incurred uncertainties.

For the day-to-day 24h size distributions a corresponding analysis was undertaken, with Figure D5 depicting the size distributions of NO3 at the HKUST site for all covered seasons exemplarily, and Table C3 summarizes the values of percent standard deviations for all species at both measurement sites respectively. Peak fit uncertainties typically increase with decreasing integrated peak area and can exceed the values of the peak area in the Aitken mode in a small number of cases (e.g. Figure D5c,e – ratios >1). Quantification of the Aitken mode may not be possible at high levels of confidence in these isolated cases. They were retained in the dataset due to their low frequency of occurrence and to enable a complete discussion over the full concentration range without biasing towards larger concentration (i.e. fitted peak areas) values.

**C. Additional Tables**

**Table C1.** Percentiles of relative standard deviation (*rows; corresponding to the box-whiskers plot in Figure D3e,f*) in percent from lognormal peak fits (bimodal deconvolution) for the resolved Aitken mode (a) and accumulation mode (b) particle concentration for diurnal size distributions at the HKUST supersite (2011/12), *columns* describe the data set, i.e. reconstructed size distributions from the 25th percentile, median, 75th percentile and mean of the size bins
*Sp=Spring, Su=Summer, Fa=Fall, Wi=Winter*

(a)

| Aitken mode | | 25th PC Distr. | | | | Median Distr. | | | | 75th PC Distr. | | | | Mean Distr. | | | | Range |
|---|---|---|---|---|---|---|---|---|---|---|---|---|---|---|---|---|---|---|
| % SD | | Sp | Su | Fa | Wi | Sp | Su | Fa | Wi | Sp | Su | Fa | Wi | Sp | Su | Fa | Wi | |
| **NO3** (UST) | PC-90 | 67 | 76 | 39 | 85 | 76 | 44 | 54 | 55 | 80 | 75 | 56 | 39 | 36 | 77 | 44 | 42 | 36-85 |
| | PC-75 | 52 | 42 | 28 | 57 | 66 | 32 | 39 | 40 | 61 | 59 | 44 | 29 | 22 | 38 | 35 | 34 | 22-66 |
| | **PC-50** | **36** | **33** | **22** | **46** | **44** | **26** | **31** | **22** | **40** | **43** | **34** | **23** | **18** | **28** | **24** | **30** | **18-46** |
| | PC-25 | 29 | 25 | 16 | 28 | 26 | 14 | 24 | 18 | 25 | 29 | 22 | 17 | 14 | 19 | 20 | 25 | 14-29 |
| | PC-10 | 21 | 20 | 15 | 24 | 19 | 12 | 19 | 15 | 17 | 19 | 17 | 10 | 13 | 13 | 15 | 21 | 10-24 |
| **SO4** (UST) | PC-90 | 38 | 38 | 42 | 74 | 19 | 36 | 39 | 43 | 22 | 40 | 40 | 42 | 19 | 36 | 37 | 81 | 19-81 |
| | PC-75 | 35 | 33 | 38 | 55 | 18 | 28 | 33 | 32 | 18 | 30 | 35 | 64 | 17 | 30 | 30 | 66 | 17-66 |
| | **PC-50** | **28** | **30** | **33** | **32** | **16** | **26** | **27** | **25** | **14** | **24** | **27** | **27** | **14** | **24** | **26** | **48** | **14-48** |
| | PC-25 | 21 | 25 | 26 | 24 | 13 | 21 | 22 | 21 | 11 | 18 | 24 | 22 | 12 | 21 | 23 | 40 | 11-40 |
| | PC-10 | 17 | 23 | 24 | 19 | 11 | 20 | 20 | 13 | 10 | 14 | 20 | 19 | 10 | 19 | 22 | 35 | 10-35 |
| **Org** (UST) | PC-90 | 52 | 23 | 41 | 44 | 42 | 28 | 32 | 27 | 47 | 48 | 45 | 53 | 46 | 29 | 24 | 32 | 23-53 |
| | PC-75 | 41 | 18 | 26 | 28 | 30 | 22 | 27 | 22 | 26 | 39 | 35 | 44 | 36 | 23 | 21 | 25 | 18-44 |
| | **PC-50** | **26** | **14** | **17** | **21** | **19** | **18** | **21** | **17** | **20** | **32** | **26** | **35** | **23** | **18** | **19** | **21** | **14-35** |
| | PC-25 | 16 | 11 | 13 | 18 | 17 | 16 | 19 | 15 | 18 | 28 | 20 | 29 | 18 | 14 | 16 | 17 | 11-29 |
| | PC-10 | 9 | 9 | 10 | 11 | 14 | 9 | 15 | 12 | 15 | 20 | 17 | 21 | 17 | 11 | 14 | 16 | 9-21 |

(b)

| Accum. mode | | 25th PC Distr. | | | | Median Distr. | | | | 75th PC Distr. | | | | Mean Distr. | | | | Range |
|---|---|---|---|---|---|---|---|---|---|---|---|---|---|---|---|---|---|---|
| % SD | | Sp | Su | Fa | Wi | Sp | Su | Fa | Wi | Sp | Su | Fa | Wi | Sp | Su | Fa | Wi | |
| **NO3** (UST) | PC-90 | 8 | 5 | 8 | 4 | 7 | 5 | 6 | 3 | 4 | 9 | 6 | 3 | 3 | 4 | 5 | 1 | 1-9 |
| | PC-75 | 6 | 4 | 7 | 3 | 3 | 4 | 5 | 2 | 4 | 7 | 5 | 2 | 2 | 3 | 5 | 1 | 1-7 |
| | **PC-50** | **4** | **3** | **5** | **2** | **3** | **3** | **5** | **2** | **3** | **4** | **5** | **2** | **2** | **3** | **4** | **1** | **1-5** |
| | PC-25 | 3 | 3 | 4 | 2 | 2 | 3 | 4 | 1 | 3 | 3 | 4 | 2 | 2 | 2 | 4 | 1 | 1-4 |
| | PC-10 | 2 | 2 | 4 | 1 | 2 | 2 | 4 | 1 | 2 | 2 | 3 | 1 | 2 | 2 | 3 | 1 | 1-4 |
| **SO4** (UST) | PC-90 | 2 | 3 | 3 | 5 | 2 | 2 | 2 | 3 | 3 | 2 | 3 | 4 | 2 | 2 | 2 | 2 | 2-5 |
| | PC-75 | 2 | 2 | 2 | 2 | 2 | 2 | 2 | 2 | 2 | 2 | 2 | 3 | 1 | 2 | 2 | 2 | 1-3 |
| | **PC-50** | **2** | **2** | **2** | **1** | **2** | **2** | **2** | **2** | **2** | **2** | **2** | **2** | **1** | **2** | **2** | **1** | **1-2** |
| | PC-25 | 2 | 2 | 2 | 1 | 1 | 2 | 2 | 1 | 1 | 2 | 2 | 2 | 1 | 2 | 2 | 1 | 1-2 |
| | PC-10 | 1 | 2 | 1 | 1 | 1 | 2 | 2 | 1 | 1 | 1 | 2 | 1 | 1 | 2 | 2 | 1 | 1-2 |
| **Org** (UST) | PC-90 | 29 | 12 | 16 | 9 | 18 | 8 | 9 | 5 | 10 | 6 | 7 | 3 | 18 | 7 | 5 | 5 | 3-29 |
| | PC-75 | 18 | 9 | 11 | 7 | 10 | 5 | 5 | 3 | 6 | 4 | 5 | 2 | 11 | 5 | 4 | 3 | 2-18 |
| | **PC-50** | **12** | **7** | **5** | **5** | **6** | **4** | **4** | **3** | **4** | **3** | **4** | **2** | **7** | **3** | **3** | **3** | **2-12** |
| | PC-25 | 7 | 4 | 3 | 3 | 5 | 3 | 3 | 2 | 3 | 2 | 3 | 2 | 5 | 2 | 2 | 2 | 2-7 |
| | PC-10 | 4 | 4 | 1 | 2 | 4 | 2 | 2 | 1 | 2 | 2 | 2 | 1 | 2 | 1 | 2 | 2 | 1-4 |

**Table C2.** Percentiles of relative standard deviation (*rows; corresponding to the box-whiskers plot in Figure D3e,f*) in percent from lognormal peak fits (bimodal deconvolution) for the resolved Aitken mode (a) and accumulation mode (b) particle concentration for diurnal size distributions at the urban MK site (2013), *columns* describe the data set, i.e. reconstructed size distributions from the 25th percentile, median, 75th percentile and mean of the size bins
*Sp=Spring, Su=Summer*

(a)

| Aitken mode % SD | | 25th PC Distr. | | Median Distr. | | 75th PC Distr. | | Mean Distr. | | Range |
|---|---|---|---|---|---|---|---|---|---|---|
| | | Sp | Su | Sp | Su | Sp | Su | Sp | Su | |
| NO3 (MK) | PC-90 | 38 | 46 | 40 | 34 | 40 | 41 | 22 | 26 | 22-46 |
| | PC-75 | 27 | 34 | 26 | 28 | 34 | 35 | 20 | 16 | 16-35 |
| | **PC-50** | **15** | **24** | **23** | **21** | **25** | **28** | **18** | **14** | **14-28** |
| | PC-25 | 9 | 20 | 19 | 17 | 18 | 22 | 15 | 12 | 9-22 |
| | PC-10 | 6 | 19 | 16 | 14 | 15 | 20 | 13 | 11 | 6-20 |
| SO4 (MK) | PC-90 | 63 | 46 | 35 | 38 | 24 | 31 | 23 | 21 | 21-63 |
| | PC-75 | 50 | 36 | 30 | 36 | 21 | 28 | 21 | 20 | 20-50 |
| | **PC-50** | **44** | **33** | **28** | **24** | **20** | **23** | **19** | **18** | **18-44** |
| | PC-25 | 37 | 27 | 24 | 21 | 17 | 20 | 15 | 17 | 15-37 |
| | PC-10 | 33 | 25 | 21 | 19 | 15 | 18 | 15 | 16 | 15-33 |
| Org (MK) | PC-90 | 22 | 12 | 22 | 19 | 30 | 21 | 15 | 14 | 12-30 |
| | PC-75 | 16 | 10 | 12 | 12 | 18 | 11 | 12 | 8 | 8-18 |
| | **PC-50** | **10** | **8** | **10** | **9** | **10** | **9** | **8** | **7** | **7-10** |
| | PC-25 | 8 | 7 | 8 | 7 | 7 | 8 | 6 | 6 | 6-8 |
| | PC-10 | 7 | 6 | 7 | 6 | 6 | 6 | 5 | 5 | 5-7 |

(b)

| Accum. mode % SD | | 25th PC Distr. | | Median Distr. | | 75th PC Distr. | | Mean Distr. | | Range |
|---|---|---|---|---|---|---|---|---|---|---|
| | | Sp | Su | Sp | Su | Sp | Su | Sp | Su | |
| NO3 (MK) | PC-90 | 9 | 9 | 6 | 7 | 3 | 5 | 2 | 5 | 2-9 |
| | PC-75 | 6 | 8 | 4 | 5 | 3 | 4 | 2 | 4 | 2-8 |
| | **PC-50** | **4** | **6** | **3** | **4** | **2** | **3** | **2** | **3** | **2-6** |
| | PC-25 | 2 | 5 | 2 | 4 | 2 | 3 | 1 | 3 | 1-5 |
| | PC-10 | 2 | 4 | 1 | 3 | 1 | 3 | 1 | 2 | 1-4 |
| SO4 (MK) | PC-90 | 4 | 3 | 2 | 3 | 2 | 5 | 2 | 3 | 2-5 |
| | PC-75 | 3 | 2 | 2 | 3 | 2 | 4 | 2 | 2 | 2-4 |
| | **PC-50** | **3** | **2** | **2** | **2** | **1** | **4** | **2** | **2** | **1-4** |
| | PC-25 | 2 | 2 | 2 | 2 | 1 | 3 | 1 | 2 | 1-3 |
| | PC-10 | 2 | 2 | 1 | 2 | 1 | 3 | 1 | 2 | 1-3 |
| Org (MK) | PC-90 | 9 | 8 | 8 | 7 | 9 | 5 | 6 | 6 | 5-9 |
| | PC-75 | 6 | 7 | 5 | 5 | 5 | 4 | 5 | 4 | 4-7 |
| | **PC-50** | **5** | **5** | **4** | **4** | **3** | **4** | **3** | **4** | **3-5** |
| | PC-25 | 4 | 4 | 3 | 3 | 3 | 3 | 3 | 3 | 3-4 |
| | PC-10 | 2 | 3 | 3 | 3 | 5 | 2 | 2 | 2 | 2-3 |

**Table C3.** Percentiles of percent standard deviation (*rows; corresponding to the box-whiskers plot in Figure D5e,f*) from lognormal peak fits (bimodal deconvolution) for the resolved Aitken mode and accumulation mode for 24h day-to-day size distributions at the suburban HKUST site (a) and the urban MK site (b) for all investigated species, *columns* describe the uncertainties in terms of quartiles of resolved peak area, where Q1 refers to the lowest 25% and Q4 the highest 25% of resolved peak area (see also Figure D4)

(a)

| HKUST '11/12 % SD | | NO3 | | | | | SO4 | | | | | Org | | | | |
|---|---|---|---|---|---|---|---|---|---|---|---|---|---|---|---|---|
| | | Q1 | Q2 | Q3 | Q4 | Range | Q1 | Q2 | Q3 | Q4 | Range | Q1 | Q2 | Q3 | Q4 | Range |
| Aitken mode | PC-90 | 95 | 64 | 50 | 27 | 27-95 | 97 | 78 | 49 | 30 | 30-97 | 62 | 37 | 28 | 26 | 26-62 |
| | PC-75 | 58 | 47 | 37 | 24 | 24-58 | 91 | 57 | 36 | 20 | 20-91 | 41 | 24 | 21 | 22 | 21-41 |
| | **PC-50** | **47** | **35** | **25** | **17** | **17-47** | **60** | **39** | **29** | **16** | **16-60** | **30** | **17** | **14** | **17** | **14-30** |
| | PC-25 | 30 | 25 | 20 | 13 | 13-30 | 32 | 26 | 21 | 13 | 13-32 | 25 | 14 | 10 | 12 | 10-25 |
| | PC-10 | 20 | 16 | 15 | 8 | 8-20 | 24 | 18 | 13 | 11 | 11-24 | 16 | 11 | 6 | 7 | 6-16 |
| Accum. mode | PC-90 | 31 | 9 | 7 | 4 | 4-31 | 4 | 3 | 3 | 3 | 3-4 | 18 | 11 | 9 | 7 | 7-18 |
| | PC-75 | 13 | 6 | 5 | 3 | 3-13 | 3 | 3 | 3 | 2 | 2-3 | 9 | 7 | 6 | 5 | 5-9 |
| | **PC-50** | **8** | **4** | **3** | **2** | **2-8** | **2** | **2** | **2** | **2** | **~2** | **6** | **4** | **5** | **3** | **3-6** |
| | PC-25 | 3 | 3 | 2 | 2 | 2-3 | 2 | 2 | 2 | 2 | ~2 | 4 | 3 | 3 | 2 | 2-4 |
| | PC-10 | 1 | 2 | 2 | 1 | 1-2 | 2 | 2 | 2 | 1 | 1-2 | 3 | 2 | 2 | 2 | 2-3 |

(b)

| MK '13 % SD | | NO3 | | | | | SO4 | | | | | Org | | | | |
|---|---|---|---|---|---|---|---|---|---|---|---|---|---|---|---|---|
| | | Q1 | Q2 | Q3 | Q4 | Range | Q1 | Q2 | Q3 | Q4 | Range | Q1 | Q2 | Q3 | Q4 | Range |
| Aitken mode | PC-90 | 62 | 52 | 47 | 30 | 30-62 | 94 | 50 | 29 | 24 | 24-94 | 23 | 17 | 13 | 16 | 13-23 |
| | PC-75 | 41 | 42 | 30 | 25 | 25-42 | 47 | 35 | 22 | 16 | 16-47 | 16 | 14 | 11 | 12 | 11-16 |
| | **PC-50** | **28** | **34** | **21** | **18** | **18-34** | **33** | **24** | **17** | **12** | **12-33** | **11** | **10** | **6** | **8** | **6-11** |
| | PC-25 | 21 | 22 | 19 | 12 | 12-22 | 26 | 17 | 13 | 9 | 9-26 | 8 | 7 | 5 | 6 | 5-8 |
| | PC-10 | 6 | 10 | 13 | 8 | 6-13 | 19 | 14 | 10 | 5 | 5-19 | 6 | 5 | 3 | 3 | 3-6 |
| Accum. mode | PC-90 | 22 | 21 | 6 | 6 | 6-22 | 7 | 6 | 6 | 4 | 4-7 | 13 | 8 | 9 | 8 | 8-13 |
| | PC-75 | 17 | 10 | 5 | 3 | 3-17 | 4 | 4 | 4 | 2 | 2-4 | 8 | 7 | 6 | 4 | 4-8 |
| | **PC-50** | **10** | **6** | **4** | **2** | **2-10** | **3** | **2** | **2** | **1** | **1-3** | **6** | **5** | **4** | **2** | **2-6** |
| | PC-25 | 6 | 3 | 2 | 2 | 2-6 | 2 | 2 | 1 | 1 | 1-2 | 4 | 4 | 3 | 2 | 2-4 |
| | PC-10 | 3 | 1 | 2 | 1 | 1-3 | 1 | 1 | 1 | 1 | ~1 | 2 | 3 | 2 | 1 | 1-3 |

**Table C4.** Median organic subcomponent concentrations in NR-PM$_1$ prior to and during meal hours at the urban MK site and their fractional contribution to total change in Organics in NR-PM$_1$ (Lee et al., 2015)

| Mass conc. μg m$^{-3}$ | Spring | | | | Summer | | | |
|---|---|---|---|---|---|---|---|---|
| | SOA | COA | HOA | Total Org | SOA | COA | HOA | Total Org |
| **Pre-lunch** 09:00 – 11:00 | 3.9 | 3.0 | 3.9 | 10.9 | 1.6 | 2.6 | 2.3 | 6.5 |
| **Lunch** 12:00 – 14:00 | 4.6 | 7.0 | 4.0 | 15.8 | 1.8 | 4.0 | 2.1 | 8.0 |
| **Contribution to ΔTotal Org** | +16.3% | +81.4% | +2.3% | --- | +18.3% | +94.1% | -8.6% | --- |
| **Pre-dinner** 15:00 – 17:00 | 4.4 | 4.0 | 3.8 | 12.1 | 1.7 | 3.3 | 2.3 | 7.3 |
| **Dinner** 19:00 – 21:00 | 4.6 | 9.2 | 4.1 | 17.9 | 1.6 | 6.3 | 2.3 | 10.1 |
| **Contribution to ΔTotal Org** | +4.7% | +89.6% | +5.7% | --- | -4.4% | +106.3% | -1.9% | --- |

**Table C5.** Ratio of 10[th] and 90[th] percentile mass concentration to median mass concentration of submicron (NR-PM$_1$) species at the urban MK site (Lee et al., 2015)

| Percentile ratio | Spring | | | Summer | | |
|---|---|---|---|---|---|---|
| | Org | SO4 | NO3 | Org | SO4 | NO3 |
| **10[th] / 50[th]** | 0.5 | 0.3 | 0.3 | 0.4 | 0.3 | 0.5 |
| **90[th] / 50[th]** | 2.1 | 1.7 | 2.8 | 2.3 | 1.8 | 2.2 |

**D.  Additional Figures**

[Figure]

**Figure D1.** Example of a log-normal peak fit *(Multipeak Fit V2, Igor Pro, Wavemetrics, Levenberg-Marquardt algorithm)* of an AMS organics size distribution

[Figure]

**Figure D2.** 24h average size distribution of sulfate (12/12/2011, suburban HKUST site) with (a) unimodal and (b) bimodal logN peak fitting applied; histograms of residuals from the unimodal (c) and bimodal (d) distributions with Gaussian fit (green); and cumulative probability density functions of uni- and bimodal fit residuals (e) with Kolmogorov-Smirnov D metric values at 95% confidence level

[Figure]

**Figure D3.** Standard deviation of peak area as a function of mode peak area (a,b), histogram of relative standard deviation i.e. the ratio of standard deviation to mode peak area (c,d) where the last bin also contains all values beyond the last bin range, and relative standard deviation as a function of mode peak area (e,f) for the fitted Aitken and accumulation mode with binned box-whiskers plot (25th to 75th PC box with horizontal median line and 10th to 90th PC whiskers where bins refer to quartiles of peak area from lowest Q1 to highest Q4); data for diurnal size distributions of NO3 at the urban Mong Kok site in summer 2013.

[Figure]

**Figure D4.** Plot of 75[th] percentile-bin diurnal variation with peak area fit variability and relative standard deviation (corresponding to green data and second to last box in Figure D3e,f)

[Figure]

**Figure D5.** Standard deviation of peak area as a function of mode peak area (a,b), histogram of relative standard deviation i.e. the ratio of standard deviation to mode peak area (c,d) where the last bin also contains all values beyond the last bin range, and relative standard deviation as a function of mode peak area (e,f) for the fitted Aitken and accumulation mode with binned box-whiskers plot (25th to 75th PC box with horizontal median line and 10th to 90th PC whiskers where bins refer to quartiles of peak area from lowest Q1 to highest Q4); data for day-to-day size distributions of NO3 at the HKUST site including all seasons.

[Figure]

**Figure D6.** Mode diameter (mass median diameter - MMD), integrated particle mass concentration and width (geometric standard deviation - GSD) of the Aitken mode and accumulation mode from bimodal diurnal peak fits of organic, nitrate and sulfate size distributions at the Mong Kok urban site in spring 2013 and summer 2013; the top panel depicts the diurnal variations of total measured submicron organic, nitrate and sulfate concentrations (AMS V-mode data)

[Figure]

**Figure D7.** Mode diameter (mass median diameter - MMD), integrated particle mass concentration and width (geometric standard deviation - GSD) of the Aitken mode and accumulation mode from bimodal diurnal peak fits of organic size distributions at the suburban HKUST site in four seasons (May 2011- Feb 2012); the top panel depicts the diurnal variations of total measured submicron organic concentrations (AMS V-mode data)

[Figure]

**Figure D8.** Mode diameter (mass median diameter - MMD), integrated particle mass concentration and width (geometric standard deviation - GSD) of the Aitken mode and accumulation mode from bimodal diurnal peak fits of nitrate size distributions at the suburban HKUST site in four seasons (May 2011- Feb 2012); the top panel depicts the diurnal variations of total measured submicron nitrate concentrations (AMS V-mode data)

[Figure]

**Figure D9.** Mode diameter (mass median diameter - MMD), integrated particle mass concentration and width (geometric standard deviation - GSD) of the Aitken mode and accumulation mode from bimodal diurnal peak fits of sulfate size distributions at the suburban HKUST site in four seasons (May 2011- Feb 2012); the top panel depicts the diurnal variations of total measured submicron sulfate concentrations (AMS V-mode data)

[Figure]

**Figure D10.** Diurnal variation of temperature (*orange*), RH (*blue*) and solar irradiance (*yellow*) in four seasons in 2011-2012 as well as spring and summer 2013; temperature and RH measurements from the HKUST supersite for the four seasons in 2011-2012, and from the Mong Kok urban site for spring and summer 2013; solar irradiance data always from the HKUST supersite.

[Figure]

**Figure D11.** Diurnal variation of fitted $C_xH_yN_zO$ and $C_xH_yN_zO_2$ ions (*raw W-mode mass concentrations*) at the urban Mong Kok site in 2013 in spring (a) and summer (b), meal hours highlighted in yellow

[Figure]

**Figure D12.** Means of clustered back trajectories (HYSPLIT4, 72h back trajectories) in each sampling season at the suburban HKUST site in (a) spring, (b) summer, (c) fall and (d) winter 2011-2012

[Figure]

**Figure D13.** Wind rose plots for observed surface wind frequency at the suburban HKUST site in spring 2011 (2011-05) for the whole sampling period and the nighttime period between 02:00 and 06:00 (a) and in summer 2011 (2011-09) for the whole sampling period and the morning period between 04:00 and 10:00

[Figure]

**Figure D14.** Diurnal variation of PMF-resolved organic aerosol factors at the suburban HKUST site in (a) spring, (b) summer, (c) fall and (d) winter 2011-2012, mean as colored solid line with standard deviations, median as open markers, $25^{th}$ and $75^{th}$ percentiles as hashed colored lines, for details see Li et al. (Li et al., 2015).

[revised manuscript text omitted]

---

## Author Response (AR2)

We thank the co-editor for her valuable input. Please find below our responses to the raised comments, questions and suggestions. In the following, raised **comments / suggestions are in red** and respective **responses in green**, while **alterations to the manuscript text are indicated in blue.**

**General Comment**

Thank you for your consideration of the referees' comments. I agree with Referee #1 that many of the comments have been addressed; however, there remain a few minor issues that should be addressed prior to acceptance. In addition to Referee #1's comment regarding transmission efficiency, please consider the comments below.

We have addressed the referee's comment and provide responses to the co-editor's individual comments in the specific comment section below.

**Specific Comments**

| | |
|---|---|
| **Comment** | 1. Lines 95-98: I encourage the rethink the wording regarding "finer time scales" and to use something that makes it more clear that hourly resolved diurnal patterns or 24-h average time-series data are being analyzed. This would better represent the results of this paper and may avoid confusion regarding the difference in time-scales achievable with AMS total loading vs. size-resolved measurements. |
| **Response** | We have revised the statement accordingly. |
| **Alteration** | […] In this work, we introduce a systematic approach of assessing temporal variations in AMS mass-based particle size distributions from hourly diurnal variations to 24h-average based day-to-day trends to utilize two key instrumental advantages, i.e. species segregation and high time resolution, to obtain a more detailed and quantitative understanding of the variabilities in ambient particle mass size distributions and to provide an additional dimension to standard AMS data analysis techniques. […] |
| **Comment** | 2. Lines 165-168: For completeness, the lower limit transmission efficiency should also be discussed. |
| **Response** | We have altered the paragraph to include both the upper and lower transmission limit. |
| **Alteration** | […] The transmission efficiency of the AMS aerodynamic lens is known to fall off below ~100nm and beyond ~550 nm of vacuum-aerodynamic diameter (Liu et al., 2007;Takegawa et al., 2009;Zhang et al., 2004;Bahreini et al., 2008;Williams et al., 2013;Knote et al., 2011) and may bias measured particle mass and mode diameters, particularly in the accumulation mode towards lower values if significant particle mass fractions fall in the size region of $D_{va} > 550$ nm. In the Aitken mode range, the effect of limited lens transmission is expected to be less substantial as particle volume (and hence particle mass) of Aitken mode particles are much smaller. We discuss the effects of lens transmission briefly in section 3.4. […] |
| **Comment** | 3. Lines 170-172: The possibility of slow vaporization and its influence on the results should be discussed. |
| **Response** | We have included a statement on the issue of slow vaporization and its possible impacts. |
| **Alteration** | [...]Delayed vaporization of particle components, e.g. under high mass loadings, can lead to small shifts towards larger mode diameters in AMS size distributions (Docherty et al., 2015) and enhanced tails in the size distributions (Cross et al., 2009), which may lead to larger fit residuals at the trailing edges. […] |
| **Comment** | 4. Line 189-190 and elsewhere: "attributable to traffic and cooking sources" These references to previous factor analysis of the data are numerous. It may be unclear to some readers that these results come from a previous analysis. Since much of the interpretation of the variability in the organic aerosol uses this past analysis, I suggest adding a few sentences in Sect. 2.1 to explicitly clarify where the factors come from (i.e. which publication) and a very brief overview of how they were derived. |

| | |
|---|---|
| **Response** | We have added an explanatory paragraph in section 2.1. |
| **Alteration** | […] AMS data were treated according to general AMS data treatment principles (DeCarlo et al., 2006;Jimenez et al., 2003) with standard software packages (SQUIRREL, PIKA). Analysis of the unit-mass resolution mass spectra yielded non-refractory submicron particle species concentrations of major inorganic constituents ($SO_4$, $NO_3$, $NH_4$, Chl) and total organics at a base time resolution of 10 min. Positive Matrix Factorization (PMF) was used to deconvolute high-resolution organic mass spectra acquired at 10 min time resolution following recommended PMF guidelines for AMS data (Zhang et al., 2011) with the AMS PMF analysis toolkit (Ulbrich et al., 2009). At the urban Mong Kok site, six organic aerosol (OA) factors were identified encompassing three secondary organic aerosol (SOA) and three primary organic aerosol (POA) factors of which one was attributed to traffic emissions and two to cooking activities (Lee et al., 2015). Similarly, four factors were obtained from analysis of the urban HKUST site dataset with two SOA factors and two POA factors, related to traffic and cooking respectively (Li et al., 2015). Further details on the treatment of AMS size distribution data from both sampling campaigns are provided in the following section. […] |
| **Comment** | 5. Please address the large discontinuity between the beginning and end of the day for O3 and NOx in Figure 2a and for NOx in Figure 2b. |
| **Response** | We apologize for the mistake. The first two hours (0:00 – 2:00) of the urban gas data were incorrectly averaged due to an issue with the time stamp of the original data time series. We have already addressed this issue earlier but overlooked to include it in the submitted revision. Also, we would like to note that the concentrations scale of CO should be in $mg/m^3$ instead of $\mu g/m^3$ (Figure 2 and 3) and erroneously contained a pre-exponent of $10^1$ (only Figure 2, i.e. the original axis label should have read "x 10 $\mu g/m^3$"). |
| **Alteration** | *See Figure 2 in the revised manuscript.* |
| **Comment** | 6. Figure 3: Please provide one larger legend rather than 4 legends. |
| **Response** | We have amended the figure accordingly. |
| **Alteration** | *See Figure 4 in the revised manuscript.* |
| **Comment** | 7. Please reorder the figures so that they are referenced in order (figure 3 is currently referenced after figure 4) |
| **Response** | We have reordered Figures 3 and 4 in the revised manuscript. |
| **Alteration** | *See Figures 3 and 4 in the revised manuscript.* |
| **Comment** | 8. Line 311: I suggest changing "points to" to "suggestive of" or something similar. Given the lack of supporting measurements, the interpretation should be cautious. |
| **Response** | We have changed the wording accordingly. |
| **Alteration** | "[…] and thus suggests the possibility of heterogeneous $SO_2$ oxidation […]" |
| **Comment** | 9. Sect 3.2: I suggest changing the title to "Day-to-day size distributions and seasonal averages" to better represent the content of the section. |
| **Response** | We have changed the section title according to the suggestion. |
| **Alteration** | 3.2. Day-to-day size distributions and seasonal averages |
| **Comment** | 10. Page 478: Suggest adding "from spring to summer" after "displayed a moderate decrease in mode diameter" to increase readability. |
| **Response** | The suggested clarification has been added. |
| **Alteration** | […] In the Aitken mode, organics and sulfate displayed a moderate decrease in mode diameter from spring to summer by 7-8% each, while nitrate saw a more significant decrease by 25% from spring to summer.[…] |
| **Comment** | 11. Figure 5: Only panels a and b are labeled but the legend mentions panels a-l. |
| **Response** | In the submitted revision, we opted to dispense on individual labelling of each panel but neglected to update the caption. This has now been rectified. |

| | |
|---|---|
| **Alteration** | *See Figure caption in the revised manuscript.* |
| **Comment** | 12. Figure 6: The caption does not match the figure with regards to the labeling of the panels. |
| **Response** | The caption has been corrected accordingly. |
| **Alteration** | *See Figure caption in the revised manuscript.* |
| **Comment** | 13. Figure D15 is much too small to read, particularly the legend. Please clarify in the caption the significance of the yellow shading. |
| **Response** | We have split the figure to allow for a larger image size. Yellow shading indicate episodic pollution events, but since we do not discuss these in the context of this paper, we have removed the shading from the figure in this revision. |
| **Alteration** | *See Figure D15 and D16 in the revised supplement.* |

We agree with the reviewer's suggestion and have amended the wording in the methodology section. We also have expanded the discussion on lens transmission (→subchapter 3.4 in the revised manuscript, also appended below).

[revised manuscript text omitted]

**Additional Figure in Supporting Material**

[Figure]

**Figure D18.** Examples of 24h size distributions from the fall season HKUST dataset for (a) organics, (b) nitrate, (c) sulfate; *original size distributions shaded and lens transmission corrected size distributions up to 2.0µm as hashed lines;*
Applied lens transmission curve (d) from Liu et al., 2007 with interpolation between data points and a linearly flattening tail between 1.1 and 2.0 *µm*;
Scatter plots of fit parameters from bimodal peak fits for the Aitken mode (e-g) and the accumulation mode (h-j); *original size distributions on x-axis and lens transmission corrected size distributions on y-axis.*

**References**

[revised manuscript text omitted]